# Polygenic prediction of occupational status GWAS elucidates genetic and environmental interplay in intergenerational transmission, careers and health in UK Biobank

Evelina T. Akimova [1,2,10] ✉, Tobias Wolfram [3,10] ✉, Xuejie Ding[2,4,5],
Felix C. Tropf [1,6,7,11] & Melinda C. Mills[2,8,9,11]

Socioeconomic status (SES) impacts health and life-course outcomes. This genome-wide association study (GWAS) of sociologically informed occupational status measures (ISEI, SIOPS, CAMSIS) using the UK Biobank ($N$ = 273,157) identified 106 independent single-nucleotide polymorphisms of which 8 are novel to the study of SES. Genetic correlations with educational attainment ($r_g$ = 0.96–0.97) and income ($r_g$ = 0.81–0.91) point to a common genetic factor for SES. We observed a 54–57% reduction in within-family predictions compared with population-based predictions, attributed to indirect parental effects (22–27% attenuation) and assortative mating (21–27%) following our calculations. Using polygenic scores from population predictions of 5–10% (incremental $R^2$ = 0.023–0.097 across different approaches and occupational status measures), we showed that (1) cognitive and non-cognitive traits, including scholastic and occupational motivation and aspiration, link polygenic scores to occupational status and (2) 62% of the intergenerational transmission of occupational status cannot be ascribed to genetic inheritance of common variants but other factors such as family environments. Finally, links between genetics, occupation, career trajectory and health are interrelated with parental occupational status.

Socioeconomic status (SES) stratifies society, with deep impacts on wealth[1], health[2], family and life course[3]. Various disciplines, including economics, demography, public health and sociology, have operationalized this multidimensional construct, focusing on the 'big three' indicators: educational attainment, income, earnings and wealth, and occupational status. Here we conduct a genome-wide association study (GWAS) on sociologically informed occupational status measures. We exploit our findings to advance understanding and quantitative modelling of status attainment processes across the life course and their complex relationship with health.

The deeply engrained intergenerational transmission of SES and inequalities across generations[4,5] has motivated social and medical scientists to consider whether genetics plays a role in SES[6–8] and, more recently, SES-related stratification and non-genetic inheritance, which biases genetic effects on a phenotype[9,10]. So far, the focus has primarily been on educational attainment[11,12] and income[13], with less attention to the heritability of occupational status. However, family studies indicate moderate heritability of occupational status comparable to other SES measures in the range of 0.30–0.40 (refs. 14–18). Molecular genetic research on SES proxies has focused on educational attainment[6,7,19–21]

## BOX 1

# Ethical considerations of this study

The study of genetics and its relationship with social status has a complex and fraught history, with some researchers using biological factors to discriminate and reinforce inequalities. Early nineteenth-century work[98] and other contemporary studies linked biology to the study of intelligence, criminality and status, which led to contentious debates on the motivation, validity and implications of their findings[99–101]. Later studies, such as on 'Social Mobility'[102], have been critiqued for assuming causality from correlations[103]. Furthermore, works such as 'The Bell Curve'[104] by Herrnstein and Murray revisited these earlier debates, suggesting a biological basis for societal stratification, inferring that social policy interventions would be futile.

It is important to recognize that these studies have contributed to an aversion, anger and even fear of studying genetics in social stratification research[99,105]. Our Frequently Asked Questions (FAQs) (Supplementary Information Section 1) offers an accessible explanation of what our study does and, importantly, does not, find and how it can be applied. Just as one would never use a single variable to predict a complex trait, it would be obviously incorrect to use the polygenic score alone to predict a complex outcome such as occupational status[106]. Our results show exactly the opposite and highlight the need for including family, environment and socioeconomic factors. We also explicitly distance our research from studies that are overtly classist and/or racist and reinforce inequalities, confuse structural inequality with biology or draw overly simplistic policy implications. We pursue a more complex biosocial understanding of occupational stratification, intergenerational transmission, gene–environment correlation and uncovering the role that socioeconomic status plays in genetic estimates.

In a 2023 consensus report from the Hastings Center[107], a group of bioethicists and researchers in the field of social and behavioural genomics emphasized the risks and need for responsible conduct in studies examining the genetics of social and behavioural phenotypes. The risks of introducing genomics in the study of occupational status for individuals could be self-fatalism or self-stigmatization (that is, believing their occupational status is fixed or inevitable or they are less capable). Or, if not managed or communicated properly, there are risks of potential discrimination against individuals (for example, in employment, insurance, criminal justice), stigmatization of others or against entire groups, with potential for harmful or inequitably distributed policies. Another risk is that genetics distracts and channels resources away from more effective ways of addressing social stratification. Despite clear messaging, an additional risk

is that some may not take the time to read our paper or careful communications. It could be misunderstood as genetic determinism by uninformed critics or conversely, falsely used to justify and reinforce existing inequalities as inevitable, incorrectly claiming that any social interventions are ineffective[105].

Although our analysis was conducted before the Hastings Center Report design guidelines, our work mostly adheres to relevant guidelines. We provide a comprehensive explanation of the definition and measurement of our key phenotypes, used an adequately powered sample, replicated out of sample, used within-family estimates and transparently discuss and even highlight observed reductions in effect sizes (responsible conduct). We did not follow the Hastings Center Report guidelines to engage with stakeholders because we deemed relevant stakeholders difficult to define given our large sample of employed persons. However, we developed an extensive FAQs along with a 'key-points' section and are transparent in our use of the 'genetic ancestry' term, consulting the NASEM report (responsible communications)[108]. We also neither attempt to nor endorse comparison of individuals across contentious socially constructed groups, such as by race or ethnicity, nor do we compare genetic ancestral groups that could be conflated with racial or ethnic categories. Our GWAS focuses exclusively on a population of British-European genetic ancestry, limiting the generalizability of the results to that population alone. We note that a high concentration of research in this ancestry group remains a general shortcoming of contemporary genetic research, with 72% of genetic discoveries covering just three countries (the United States, the United Kingdom and Iceland)[81,82]. We encourage inclusion of diverse populations into genetic research across multiple domains including ancestry, socioeconomic backgrounds, geography, age and beyond. We also note that the dataset we use is affected by participation bias[109,80] (see discussion, FAQ in Supplementary Information Section 1). Our encouragement for diversity should not be interpreted as an endorsement of studies aimed at comparing different ancestral groups, but rather understanding the unique genetic and environmental interplay within each group rather than drawing comparisons between them, in line with the NASEM framework[108].

Alongside the accurate scientific interpretation of our research, we advocate for open discussion on the importance of the role that socioeconomic status plays in patterning other genetic outcomes that engages scholars from a wide array of intellectual backgrounds and diversity in viewpoints.

---

and income[22,23], neglecting occupational status. SES measures are important since they introduce gene–environment correlations which affect GWAS results[24] and influence the patterns of genetic correlations of mental health traits[25]. This calls for a more nuanced and holistic understanding of SES that goes beyond educational attainment and income. While SES measures are intertwined, the dimensions are clearly analytically and empirically distinct[26], and individuals may, for example, trade off income for other types of status, in particular occupations. Educational attainment may therefore not necessarily translate into economic success.

We extend previous work of a GWAS on broadly skill-based occupational groups using the UK Biobank, which identified 30 independent single-nucleotide polymorphisms (SNPs) associated with 9 very

broad categories of the UK Standard Occupational Classification (SOC) and an SNP heritability of 0.085 (ref. 27). Since occupation in the UK Biobank is richly measured using 353 categories, we go beyond the existing GWAS by drawing from decades of sociological theory and measurement of occupational stratification. Sociological measures are preferable since purely skill-based measures suffer from inconsistent operationalization and lack theoretical and substantive thinking about the underlying mechanisms of status attainment, ignoring, for example, social prestige and other status factors[1] (Box 1).

Sociologists consider occupation as the primary social and economic role held by most adults outside their immediate family or household, often even as 'the single most important dimension in social interaction' (p. 203)[28]. It is a long-term stable indicator of an

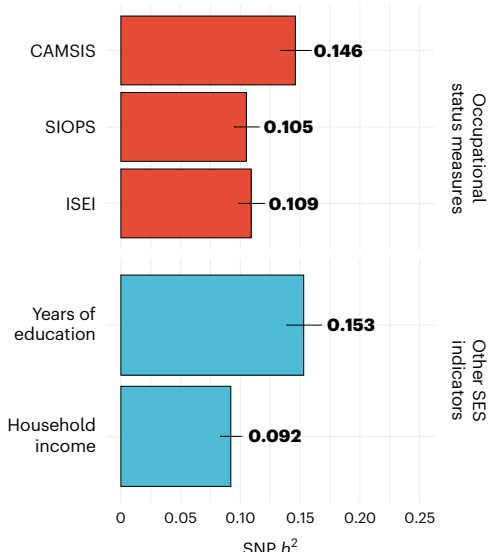

**Fig. 1 | Comparison of SNP-heritability estimates of occupational status measures vs income and education.** LD score-based SNP-heritability estimates of occupational status measures CAMSIS ($N$ = 273,157), SIOPS ($N$ = 271,769) and ISEI ($N$ = 271,769) compared to income ($N$ = 353,673) and education ($N$ = 404,420). Each bar is a single estimate of SNP heritability and each error bar indicates the s.e. of the estimate (95% confidence intervals (CIs) are presented).

individual's social position in society alongside income, consumption, division of labour and social reproduction[1]. Adequately measuring occupational status is complex, with generations of sociologists dedicated to mapping this complex qualitative trait on a continuous scale[29]. The three conceptual approaches to measuring occupational status consider either socioeconomic differences between occupations, inter-occupational social interaction, or ascribed prestige of different jobs[30].

In our analyses, we focus on three different measures of occupational status, championed by different theoretical traditions in sociology. First, the International Socioeconomic Index (ISEI)[31], is a status measure constructed from scaling weights that maximize the (indirect) influence of education on income through occupation. Second, the Standard International Occupational Prestige Scale (SIOPS)[28], is a prestige-based measure based on public opinion surveys where a representative population is tasked with ranking occupations by their relative social standing. Third, the Cambridge Social Interaction and Stratification Scale (CAMSIS)[1], measures the distance between occupations on the basis of the frequency of social interactions between them (operationalized as husband-and-wife combinations). This measure is based on the notion that differential association is a function of social stratification, with partners and friends more likely to be selected from within the same group. Although these measures are championed by different theoretical traditions in sociology, empirically they have substantial but not perfect correlations[32], alluding to an underlying latent factor of occupational status.

The current study investigates molecular genetic associations with ISEI, SIOPS and CAMSIS. Analyses were conducted on 273,157 (130,952 males and 142,205 females) individuals in the UK Biobank[33], identifying 106 independent SNPs, and replicated in the UK's National Child Development Study (NCDS; $N$ = 4,899; 2,525 females and 2,374 males). Genomic structural equation modelling (GSEM)[34] suggests a general genetic factor across all SES measures of occupational status, income and education. An overview of the study is provided in Extended Data Fig. 1.

The integration of molecular genetics into such a core topic of social science research promises a richer understanding of the role of the biological and social factors as well as the improvement of quantitative modelling and understanding of social processes of attainment status transmission. We thus utilize our GWAS discovery results for various sociogenomic investigations. While there is limited research that has identified a potential biological basis from GWAS findings for complex behavioural traits[19,35], there has been some progress towards understanding potential pathways. This is particularly in psychiatric and addiction-related phenotypes and type 2 diabetes[36–38]. Accordingly, we investigated how social and psychological mechanisms play a role in the genetics of occupational status, including childhood career aspirations, non-cognitive[39] and cognitive traits[27]. We then examined to what extent polygenic scores (PGSs) for occupational status predict the phenotype within and between families, their genetic penetrance of careers across the life course and the role common genetic variants play as a confounder of the intergenerational transmission of occupational status. Additional analyses explore the complex relationship between occupational status and health outcomes and how parental occupational status confounds the genetic prediction of general health. Our findings are conclusive that ignoring genetic data in parent–offspring SES transmission and quantitative stratification research in general leads to biased results in non-experimental studies, while the interplay between genes and the environment remains complex.

## Results

### Heritability, discovery and genetic links among SES measures

The main analyses were conducted on individuals from the UK Biobank on the three phenotypic measures of occupational status: CAMSIS ($N$ = 273,157), SIOPS ($N$ = 271,769) and ISEI ($N$ = 271,769; Methods). Linkage disequilibrium score regression (LDSC)-based SNP heritability ($h^2_{SNP}$)[40] was significantly different from zero for all occupational measures, and ~50% larger for CAMSIS ($h^2_{SNP}$ = 0.145, s.e. = 0.0066) compared with SIOPS ($h^2_{SNP}$ = 0.105, s.e. = 0.0052) and ISEI ($h^2_{SNP}$ = 0.109, s.e. = 0.0056, see Fig. 1). This is within the range of $h^2_{SNP}$ for other status measures estimated in the UK Biobank (Methods), such as education ($h^2_{SNP}$ = 0.153, s.e. = 0.0056) and income ($h^2_{SNP}$ = 0.092, s.e. = 0.0041), and for CAMSIS nearly twice as high as for previously reported occupational measures[27]. Genome-based restricted maximum likelihood (GREML) analyses confirmed these results (Supplementary Table 1).

The GWASs identified 106 independent SNPs for CAMSIS, including 56 also found for ISEI and 51 for SIOPS on the basis of an $R^2$ threshold of 0.1 and a window size of 1,000 kb (see Fig. 2 Manhattan plot), one of which (only significant for CAMSIS) was found on the X chromosome. We identified 11,206 SNPs in LD with our autosomal lead SNPs (Methods) and conducted an exhaustive phenome-wide association study (PheWAS) using the GWAS catalogue and the IEU OpenGWAS Project database. While we observe a substantial overlap with other socioeconomic status-related traits, 8 of our variants (rs12137794, rs17498867, rs10172968, rs7670291, rs26955, rs2279686, rs72744938, rs62058104) have not yet been linked to any status-related trait. For three variants (rs7670291, rs26955, rs72744938) not even suggestive associations ($P < 5 \times 10^{-6}$) with status traits are discernible. For two of these, we find strong links to platelet count. A full list of all implicated phenotypes is provided in Supplementary Table B8. The only non-autosomal hit (rs146852038) has previously been linked to the age of first sexual intercourse and educational attainment[41].

We then replicated these hits using the National Child Development Study (NCDS), an ongoing study of a British birth cohort born in 1958 (Methods). This dataset was chosen because it is a similar UK cohort, important since previous research demonstrated genetic variation by country and birth cohort for complex behavioural phenotypes[42]. Despite the notable disparity in sample size, with 4,899 individuals in the NCDS compared with ~273,157 in our discovery sample, our results surpassed the expected sign concordance and achieved a higher than anticipated number of significant hits at $P$ = 0.05 (Supplementary Information Section 7.4). This replication result underscores

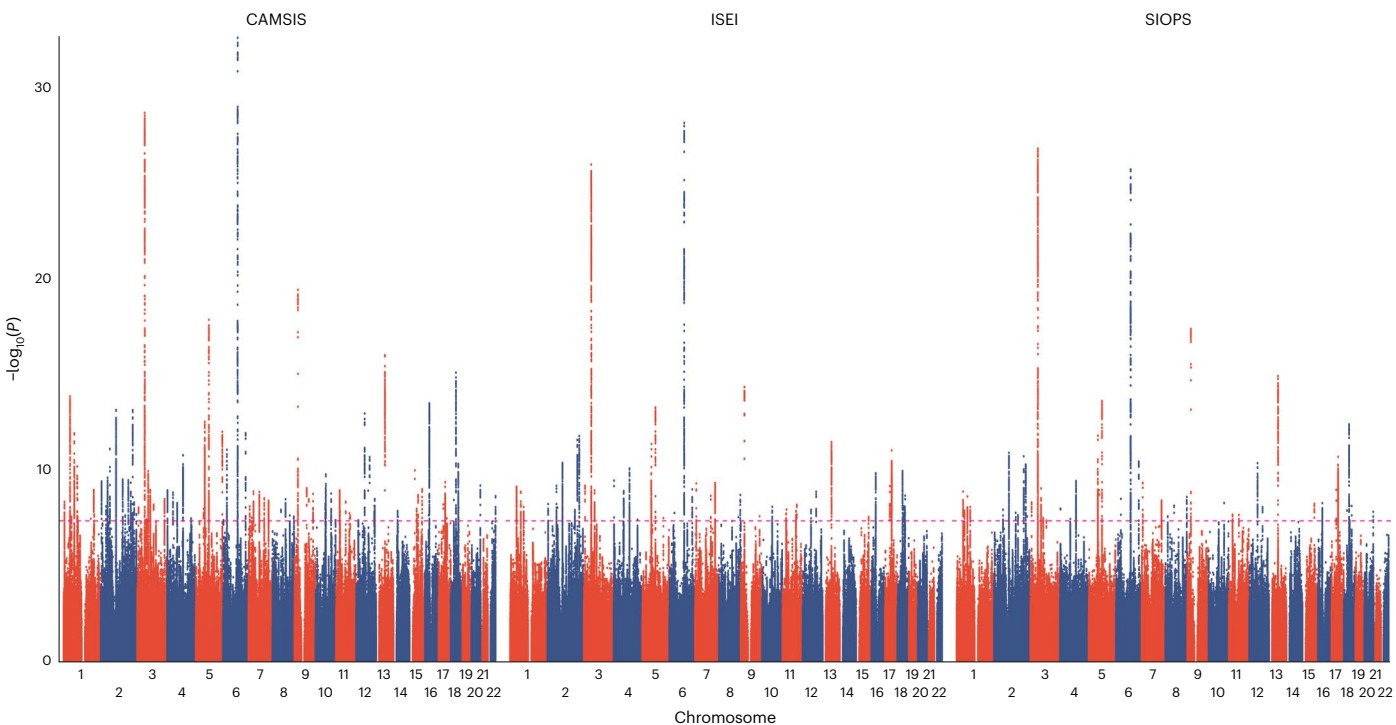

**Fig. 2 | Manhattan plot of the GWASs for occupational status measures.** Manhattan plot with autosomal SNP position on the *x* axis and the logarithm of the *P* value on the *y* axis of the GWASs for occupational status measures CAMSIS (*N* = 273,157), ISEI (*N* = 271,769) and SIOPS (*N* = 271,769).

the robustness of our findings, even when subjected to smaller-sample constraints.

To investigate the functional implications of the genetic variants associated with occupational status, we performed gene-based and gene-set analyses using multivariate analysis of genomic annotation (MAGMA; Methods)[43]. We observe that genes implicated by our SNPs are expressed in the brain, including the pituitary gland. No other tissue showed significant enrichment for gene expression.

We also jointly analysed the highly correlated occupational status measures together with income and education to increase statistical power using multitrait analysis of GWAS[44] (MTAG; Methods) resulting in 731, 646 and 653 variants passing the significance threshold for CAMSIS, ISEI and SIOPS, respectively.

Genetic correlations (Fig. 3, lower triangle) between the three measures were close to 1 and thus stronger than the phenotypic correlations (upper triangle), ranging between 0.80 and 0.90. The genetic correlations with educational attainment and household income were almost twice as high (0.81–0.97) as their phenotypic counterparts (0.32–0.44). Considering these high genetic correlations, it is unsurprising that we found strong evidence for a common genetic factor of occupational status using genomic structural equation modelling (GSEM)[34], with high loadings for all three measures (standardized path coefficients of 0.99, 0.99 and 0.99, for CAMSIS, ISEI and SIOPS, respectively; Supplementary Information Section 11). We furthermore provide evidence for a common genetic factor of SES including income and education (see Supplementary Fig. 6).

## Polygenic prediction

We assessed the out-of-sample predictive performance of the PGSs using two data sources. The first sample comprised a subset of siblings from the UK Biobank, for which we conducted an additional GWAS excluding individuals from the discovery analysis. The second sample consisted of the aforementioned NCDS.

MTAG-based out-of-sample predictions, which incorporate occupational status measures with household income and educational

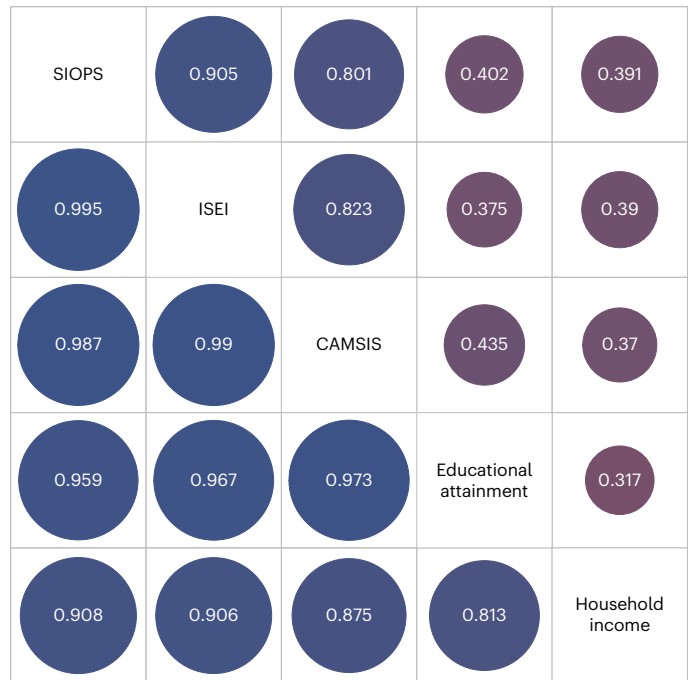

**Fig. 3 | Phenotypic and genetic correlations of occupational status measures and other SES indicators.** Upper triangle: phenotypic correlations. Lower triangle: genetic correlations. Correlations of occupational status measures and other SES indicators are based on LD score regression. *N* = 246,492 for phenotypic correlations. Darker blue circles indicate stronger positive correlations.

attainment, were slightly higher in the NCDS compared with the UK Biobank, with an incremental $R^2$ of 0.097 (s.e. = 0.0035) in NCDS across all observations (0.075, s.e. = 0.00287 in the UK Biobank) for CAMSIS,

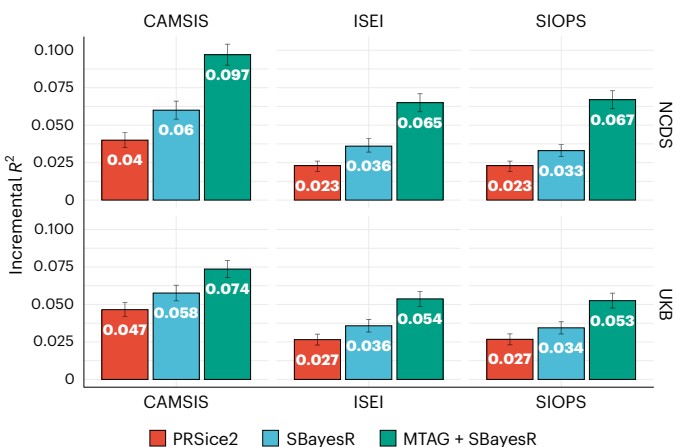

**Fig. 4 | Out-of-sample polygenic prediction performance within UK Biobank and NCDS.** Incremental $R^2$ compared to a baseline model consisting of 10 principal components, sex and age. Bars denote 95% CIs. $N = 24,579$ for CAMSIS and 24,472 for ISEI and SIOPS in the UK Biobank; for NCDS average performance over different ages, $N = 5,389; 5,312; 5,211; 4,902;$ and 4,263 for CAMSIS at ages 33, 42, 46, 50 and 55; and corresponding $N = 5,449; 5,293; 5,197; 4,892;$ and 4,252 for ISEI/SIOPS.

0.065 (s.e. = 0.0032; 0.054, s.e. = 0.0025 in the UK Biobank) for ISEI and 0.067 (s.e. = 0.0031; 0.053, s.e. = 0.00248 in the UK Biobank) for SIOPS (Fig. 4). As expected, polygenic scores based on PRSice2 and SBayesR weights have smaller but comparable incremental $R^2$ values in both UK data sets across all measures of occupational scores (Fig. 4).

The longitudinal data in the NCDS reveal changes in the PGS effects across the life course or career trajectories, respectively. First, we were able to examine PGS prediction of occupational status across the life course at ages 33, 42, 46, 50 and 55 (Supplementary Information Section 13).

By leveraging the NCDS activity calendar data, we delineated comprehensive career trajectories over 30 years, from the onset of participants' professional lives. When stratified by PGS quintiles, parental SES and sex, these trajectories revealed a notable interplay between polygenic scores and social factors (Fig. 5). Individuals who started their careers in the lower end of occupational status scores but ranked high in the PGS consistently advanced in their careers over the years. Conversely, those who initially held higher occupational status but had lower PGSs exhibited a steady decline in their professional trajectories, as measured by occupational status scores (Supplementary Fig. 12). While our focus is on CAMSIS, similar patterns were evident for SIOPS and ISEI, underscoring the consistency of our findings (Supplementary Figs. 13 and 14). These results further highlight the importance of understanding how and why societal structures and factors correlate with genotypes and jointly predict career trajectories.

### Disentangling direct, indirect and demographic effects
GWAS population estimates include a combination of direct effects (inherited genetic variation) and indirect effects or gene–environment correlations and can be further influenced by assortative mating. We conducted multiple analyses to better understand the relative importance of these dimensions in relation to our estimates (Supplementary Information Section 12)[45,46].

First, we investigated the predictive performance of our scores between more than 29,500 siblings in the UK Biobank, a common design to identify direct genetic effects. Notably, traits related to socioeconomic status or other non-clinical outcomes tend to exhibit considerable within-family effect reductions[45], potentially affecting

their practical utility[47]. Our analysis supports these previous studies, showing a reduction in effects for occupational status measures of more than 50% in total, with results for other SES measures (education and income) in a similar range (see Fig. 6 for the ratio of population and within-family models and Methods).

This discrepancy between the unrelated population and within-family estimate can be attributed to indirect family effects or assortative mating. Indirect effects include the (heritable) social transmission of economic resources, and cultural and social capital, as well as social-psychological factors such as parental expectations, which represent passive gene–environment correlation. To quantify the role of indirect effects, we use two research designs. First, we adjust the best-performing PGS prediction in the NCDS for parental SES (measured as parental occupational status at age 11). Second, we conduct an adoption prediction study. In an adoption design, children are raised by non-biological parents, thereby providing a unique opportunity to examine the influence of genetic factors while minimizing the effects of passive gene–environment correlation. We re-ran our GWAS for occupational status, while excluding the set of 3,414 respondents of British-European genetic ancestry in the UK Biobank that stated that they were adopted and for which occupational information was available. Results from both designs are remarkably similar, with the parental SES showing an effect attrition of 21% for all three measures, and the adoptee prediction resulting in an effect reduction of 23%, 22% and 27% for CAMSIS, ISEI and SIOPS, respectively. Notably, our results concur with ref. 48, where the extent of attenuation for cognitive and non-cognitive skills was considerably smaller in an adoption compared with a sibling design.

The observed remaining discrepancy between the population estimate controlling for indirect effects and within-family estimates could be attributed to attrition in the within-family design due to assortative mating, which attenuates the within-family effect. Recent findings by economic historians have demonstrated strong partner matching on occupational status within the United Kingdom dating back to at least the 1750s[49]. By employing a method first proposed by ref. 6, we demonstrate that, even in the absence of indirect effects, within-family effects are plausibly anticipated to be attenuated by 21–27% (Supplementary Information Section 12). We find further support for attenuation by directly analysing the spousal PGS correlation, which substantially exceed what could be expected from simple phenotypical assortment (Supplementary Information Section 12). Accordingly, it closes the observed gap between both estimates. Under the assumption of additive effect reduction due to assortative mating and indirect effects, all three methods consistently estimate the proportion of direct population effects to be within the range of 73–79%. This convergence of findings underscores the importance of accounting for biases related to partner matching when examining the role of genetic factors in occupational status. It furthermore motivates the inclusion of parental SES for robustness in the application of PGS analyses downstream of the population GWAS.

### Social mechanisms linking genetics and occupational status
A pertinent question to consider is which traits serve as mediators of the association between an individual's genome and occupational status. Evidence from twin studies indicates that both cognitive and non-cognitive traits play a mediating role in the relationship between genetic and social outcomes[50].

Building on previous behavioural phenotype GWASs and the literature[41], we identified five traits that are potential mediators of the general genetic factor of SES: cognitive performance[6], attention-deficit/hyperactivity disorder (ADHD; as a proxy for behavioural disinhibition)[51], openness to experience[52], risk tolerance[53] and neuroticism[54]. In a multivariate genetic regression model (Supplementary Information Section 11.3), overall, we can explain 70% of the genetic association with occupational status. Among these mediators,

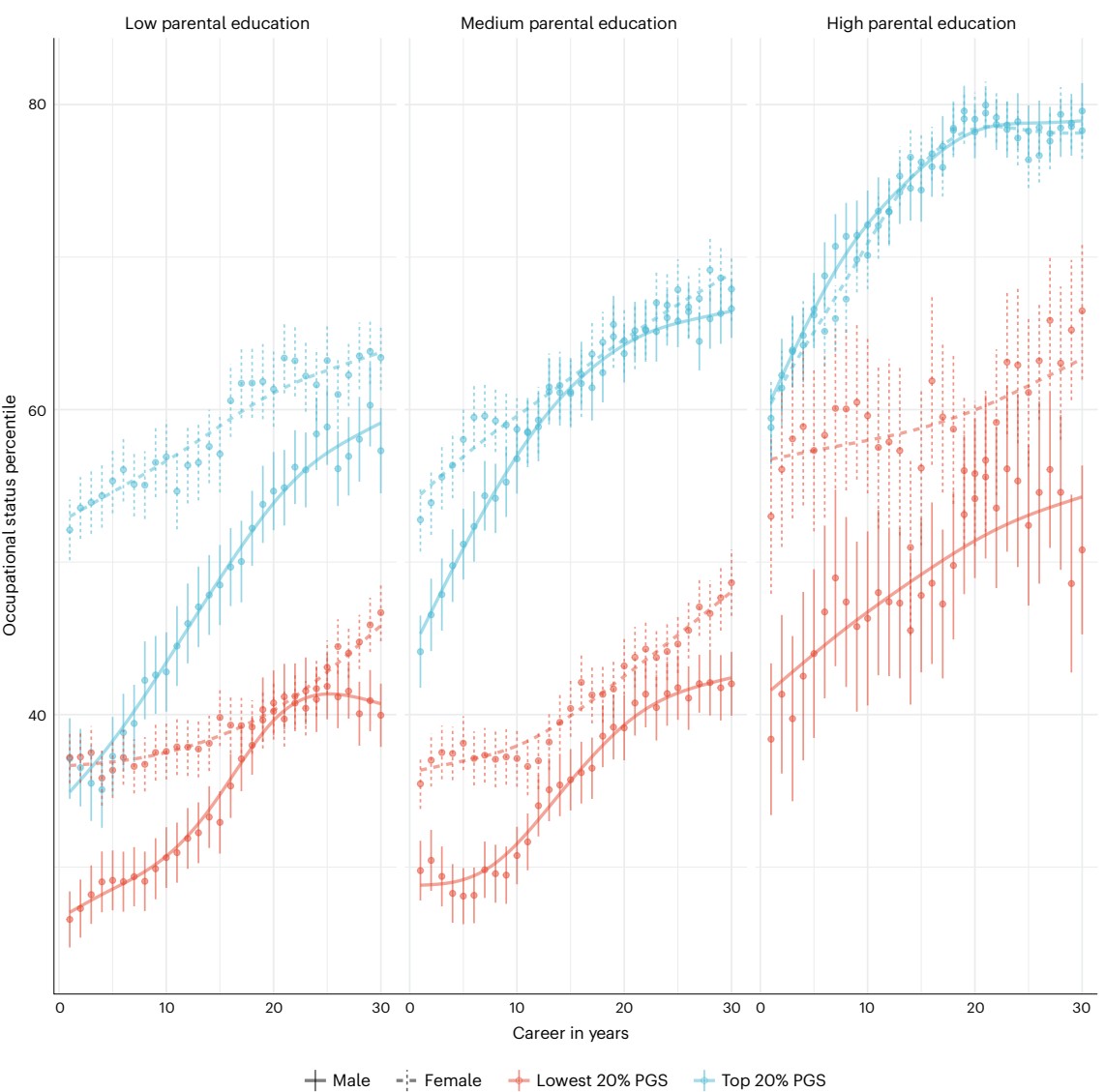

**Fig. 5 | Mean percentile of the CAMSIS occupational status distribution across career stratified by sex, parental education and the CAMSIS PGS.** *N* = 201,939 time points from 5,475 individuals. Parental education measured as Low = no qualifications, Medium = lower secondary and High = upper secondary/degree. Bars denote 95% CIs.

the associations are generally similar for all three measures of occupational status. Of these, the strongest effects are observed for cognitive performance. However, when we introduce ADHD and openness to experience into the models, these associations are slightly reduced. The importance of ADHD is increased by the introduction of risk tolerance. In contrast to ADHD and neuroticism, risk tolerance positively correlates with the SES factor, when controlling for the other potential mediators (see Supplementary Tables 3–5).

In the NCDS data, we tested the mediating effects of adolescent phenotypic measures of cognitive ability, externalizing behaviour, internalizing behaviour, scholastic motivation, occupational aspiration and subjective health for the occupational status PGS (Fig. 7, Methods and Supplementary Information Section 15). Depending on the career stage of the respondents indicated by NCDS waves, these variables explained 56–74% of the link between our PGSs and occupational status (Fig. 7). As expected, cognitive ability was the main mediator, explaining 33–51% of the association depending on respondents' age. Scholastic motivation explained between 8–11%, occupational aspiration 7–11% and other non-cognitive traits up to 5%. The overall mediation by subjective health was minimal. Effect reductions

are proportional when adjusting for parental SES to control for passive gene–environment correlation or indirect effects, respectively (Supplementary Information Section 11.3).

## Intergenerational transmission

Given that parental status is strongly associated with their offspring's status, the study of intergenerational status transmission has a long tradition, often focusing on educational attainment[55,56]. In the NCDS data, the phenotypic correlation between paternal occupation at age 11 and offspring occupational status at various ages for all three measures was substantial (~0.30). Including a PGS to control for genetic inheritance and identify social effects reduced the intergenerational correlation of occupational status by 11%. However, this is probably an underestimation given the power limitations of GWAS in capturing full SNP heritability. Rescaling the results to estimated SNP heritability[57], up to 38% of the intergenerational correlation is due to common genetic inheritance; 62% is due to other factors, which include social inheritance and possibly the effects of rare genetic variants[58] not captured by SNP-heritability estimates (see also Fig. 8 and Supplementary Information Section 16 for estimates by age).

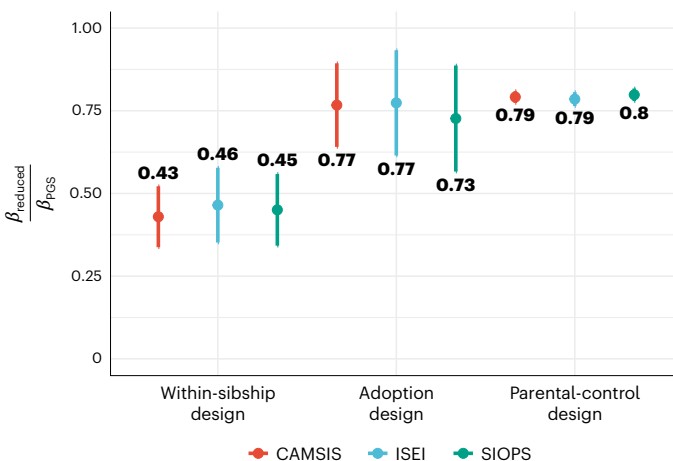

**Fig. 6 | Ratio of standardized beta coefficients for the effect of the respective PGS on the phenotype based on within-sibship, adoption and parental control models to the population estimate for CAMSIS, SIOPS and ISEI.** Ratios based on within-sibship, adoption and parental-control models. $N = 24,579$ for CAMSIS, 24,472 for ISEI and SIOPS (within-sibship); $N = 3,398$ for CAMSIS and 3,414 for ISEI and SIOPS (adoption); $N = 13,972$ for CAMSIS and 13,973 for ISEI and SIOPS (parental control). Each estimate is the ratio of standardized beta coefficient of within-sibship, adoption or parental-control model PGS, $\beta_{reduced}$, to the beta coefficient of population-based PGS, $\beta_{PGS}$. The error bars represent 95% CIs calculated with the bootstrap method (1,000 repetitions).

## Genetic confounding between occupational status and health

Occupational status is correlated with various health outcomes and higher-SES individuals typically live longer and are in better health[2]. It is essential to understand to what extent this association between occupational status and health is a causal one to, for example, design effective health intervention strategies. The observed association could partly be driven by endogeneity since individuals with better health also potentially secure better jobs or have higher performance at work. Controlling for genetic associations reduces biases arising from genetic endogeneity also in regard to potential direct pleiotropic effects[59]. We therefore investigate to what extent the occupational status PGS confounds the observed relationship between occupational status and general health as well as mental health in the NCDS data (see Supplementary Tables 14 and 16 for regression estimates). Similar to the intergenerational transmission of status, we find significant genetic confounding in the observed relationship.

To better understand the degree to which the genotypic effect of occupational status on general and mental health might incorporate indirect effects, we analysed the health of the respondents on the basis of their occupational status PGSs with and without parental occupational status at age 11 as a control variable. In accordance with previous results, we found that taking parental occupational status into account reduced the PGS prediction of general health on average across ages and outcomes by 19.5% and of mental health by 23.7%, demonstrating the importance of considering parental SES indicators for the genetic study of offspring's health outcomes (see Supplementary Tables 15 and 17).

## Discussion

Analysing data from 273,157 individuals from the UK Biobank, we identified 106 independent SNPs associated with occupational status measures, 8 of which have not been previously reported in related SES GWASs. Our study provides PGSs that are associated with occupational status in two samples of individuals of European ancestry in the United Kingdom, with an out-of-sample prediction of 5–8% depending on the status measure and up to 9% depending on career stage. Genetic

associations derived from CAMSIS were ~50% larger than for SIOPS and ISEI and twice as high as measures applied previously[27]. This is likely since SIOPS and ISEI are based on multicountry data from the 1970s and 1980s, and CAMSIS was constructed within the United Kingdom where our sample is located. CAMSIS conceptually focuses on social interactions, in contrast to, for example, purely skill-based measures. A potential reason for this observation may be genetic selection into interaction networks of friends[60,61]. A particular feature of CAMSIS is the inclusion of spousal networks. As Fisher stated, referring to past historical periods and particular contexts: "[P]revailing opinion, mutual interest and the opportunities for social intercourse, have proved themselves sufficient, in all civilized societies, to lay on the great majority of marriages the restriction that the parties shall be of approximately equal social class"[62]. Evidence for genetic assortative mating has been demonstrated for political views[63] and educational attainment[64,65], supporting strong phenotypic evidence of assortative mating by SES, race/ethnicity and religion, also showing that this has evolved with demographic change[66]. The heritability of CAMSIS might partly capture effects of assortative mating on the phenotype of the individual. However, high genetic correlations between CAMSIS, SIOPS and ISEI may point to the benefits of a more granular and exact measure of the same latent phenotype in CAMSIS and construction of measures within similar populations[42]. We also provide the R package 'ukbjobs' to equip researchers using the UK Biobank to employ more-standard, well-defined sociologically informed measures[67].

Our study not only demonstrates the genetic interdependence of occupational status measures, but also reveals a strong genetic correlation between educational attainment, income and occupational status, identifying a common genetic factor of SES. Notably, genetic correlations among SES indicators surpass phenotypic correlations by a factor of two to three. This outcome represents an outlier from the conjecture of ref. 68, which states that phenotypic correlations can serve as proxies for genetic correlations—a notion that finds empirical support in both animals and humans[69,70].

The deviation might have several reasons, including trade-offs between investments into different dimensions of SES. Higher education does not always guarantee high income or occupational status, since labour market conditions, personal networks, ethnicity and gender can influence career trajectories[5]. Higher occupational status does not always bring a high income or demand high education, and may vary across cultures and social contexts[71]. Certain genetic traits may be associated with individuals achieving higher levels in particular areas through a mechanism known as vertical pleiotropy (that is, mediated pleiotropy)[23]. For instance, genetic factors correlate with cognitive abilities, personality traits and mental health, which may, in turn, impact educational attainment, income and occupational status. Environmental factors such as family background, social norms, cultural expectations and chance also shape SES. Environmental differences in individual cases can lead to more heterogeneity and thus weaker phenotypic correlations, and subsequently have a completely different causal pathway in influencing health and behavioural outcomes.

We have shown that the prediction attrition within families is in part due to indirect genetic effects or genetic nurture, respectively, which also consistently contribute to the latent factor for constructed SES measures. Moreover, a mounting body of evidence suggests that strong assortative mating on this latent factor has been present for multiple generations[49,72]. Notably, a higher spousal correlation has been observed for the genetic predictor of educational attainment than for the actual phenotype[64,73]. This phenomenon may partially account for why genetic variants display a stronger predictive power for occupational status between families, as opposed to within families where the variation in these variants is more limited.

We integrated the polygenic signal for occupational status into occupational mobility and social stratification research and vice versa, with crucial implications on both sides. First, intergenerational

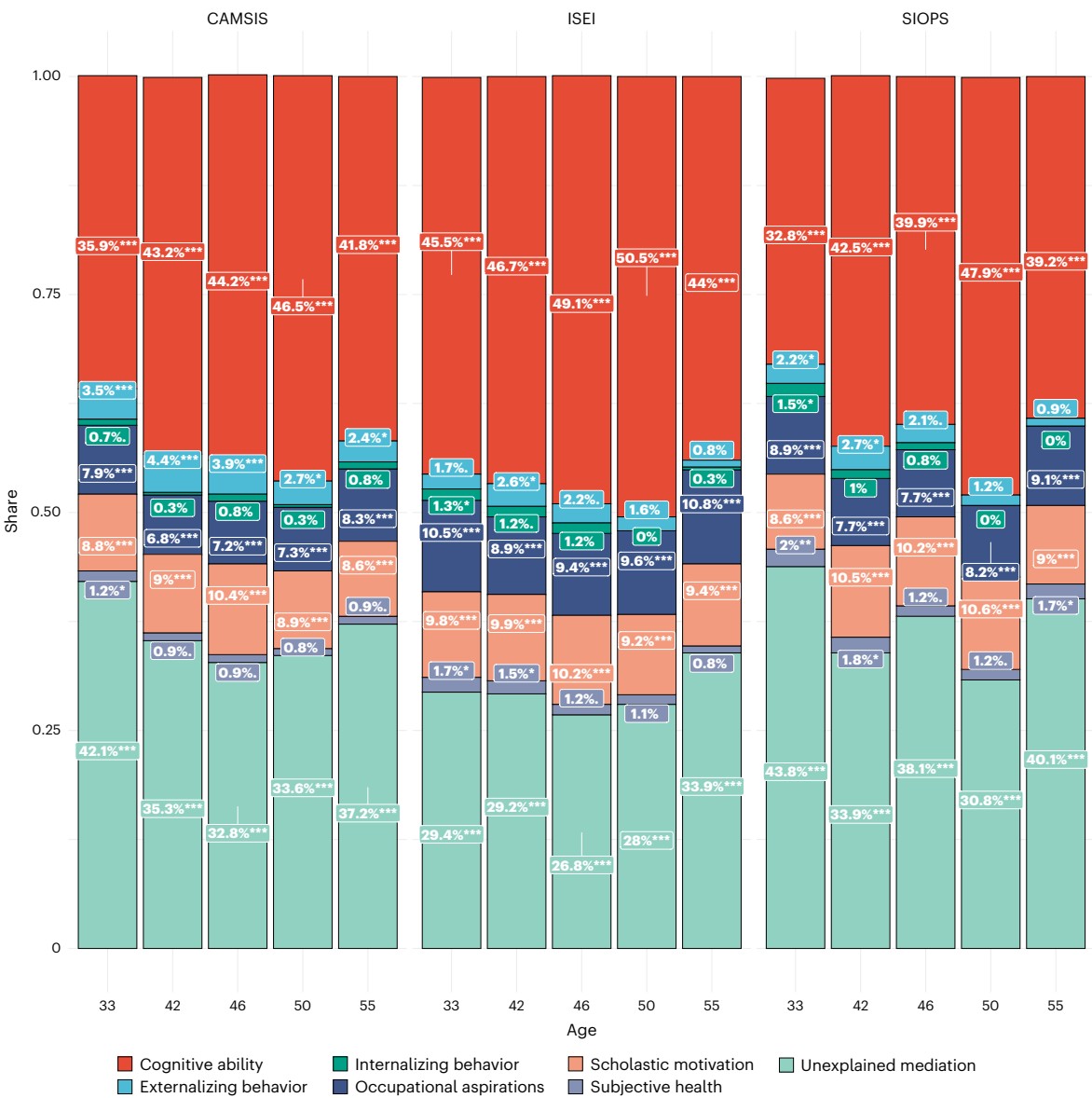

**Fig. 7 | Mediation model results of polygenic prediction of occupational status in NCDS through the life course.** $N$ = 3,169; 3,111; 3,075; 2,881; and 2,499 for CAMSIS at ages 33, 42, 46, 50 and 55; and corresponding $N$ = 3,196; 3,100; 3,068; 2,878; and 2,494 for SIOPS and ISEI. Separate linear regression models with two-sided tests. Stars indicate the significance level based on $P$ values: no star, $P > 0.05$; *$0.01 \leq P < 0.05$; **$0.001 \leq P < 0.01$; ***$P < 0.001$.

mobility in social status is of great interest, not only for social scientists, but also policymakers, public health and epidemiology and is related to questions of equality of opportunity and societies' degree of openness[5,18,74]. Next to cognitive skills, we showed that also scholastic motivation, occupational aspiration, personality traits and behavioural disinhibition (proxied by ADHD) mediate the association between genetics and occupational status. It is also vital to note that around one-third of the polygenic signal remains unexplained in each of our approaches, although it is likely that this is at least in part a result of the incomplete overlap of mediating variables between both analyses. We need further investigations to better understand the role of genetics in status inheritance and evaluate the interpretation of heritability as a pure merit measure in the context of questions addressing equality of opportunity.

Second, there are important considerations related to the intergenerational transmission of SES. It has been a common assumption in heritability studies of educational attainment that genetic influences are stable in absolute terms, while environmentally driven inequalities

tend to reduce with lower intergenerational correlations[8]. Extrapolating results from PGSs, we show that the intergenerational correlation for occupational status is up to 38% due to genetic inheritance—this is even stronger than for educational attainment[55,56]. This suggests that social stratification researchers need to adjust their sole focus on intergenerational correlations to also explicitly consider gene–environment correlation in statistical modelling. We note that the applied extrapolation assumes SNP-heritability levels but could still represent an underestimation since PGSs have a lower prediction compared with SNP heritability. However, the latter is still smaller than the heritability estimated from twin models; hence, SNP heritability as measured here remains conservative compared with previous studies[55]. The discrepancy between SNP and twin heritability might be due to rare genetic variants, higher environmental homogeneity within families and nonlinear genetic effects[42,75].

Third, we highlight questions about the causality of the relationship between health and occupational status and SES in general[2]. It is plausible to assume that higher status causally leads to better health,

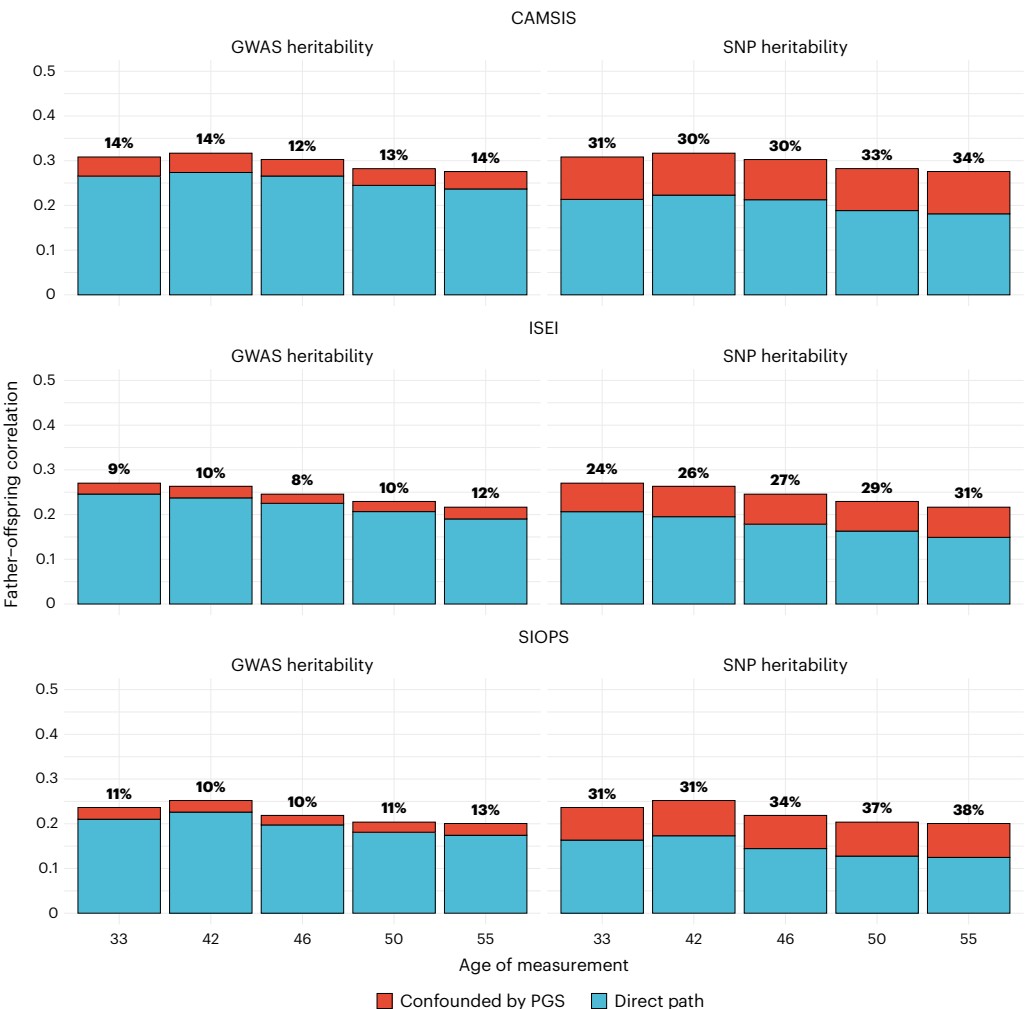

**Fig. 8 | Percentage of genetic confounding in the intergenerational transmission of occupational status in NCDS through the life course.** Percentages based on the predictive validity of polygenic scores (GWAS heritability) and an extrapolation of their effect to the variance explained by common SNPs (SNP heritability). $N$ = 3,875; 3,835; 3,747; 3,550; and 3,079 for CAMSIS at ages 33, 42, 46, 50 and 55; and corresponding $N$ = 3,902; 3,797; 3,718; 3,522; and 3,053 for SIOPS/ISEI.

for example, due to a higher living standard, nutrition and better knowledge about and access to health care systems, among others[2]. At the same time (heritable) poor health might force an individual into a lower-status occupation, or genetics might have direct pleiotropic effects on education and health or related factors, leading to an overestimation of a direct, phenotypic causal effect. The question of causality, however, is paramount for designing targeted policy interventions, and genetic confounding needs to be considered. It is also relevant to quantify their potential impact and clarify claims in social mobility research regarding genetically driven, health-related confounders. We show that the association between occupational status and health is up to 25% confounded by common genetic effects and therefore substantially upward biased when genetic factors are not considered.

Fourth, combining theoretical, measurement and modelling perspectives of the social sciences and genetics is not only important for the interpretation of status in social science theory and modelling, but also for genetic research[75]. First, the discovery of indirect parental effects unravelled the importance of social influences correlated with genotypes in the discovery of genetic effects on education[9]. We show that controlling for parental occupational status strongly reduces genetic prediction of the occupational status PGS with general and mental health. While genetic prediction based on our PGSs on health is comparably small (1%) and confounding effects may not

entirely generalize to other regions of the genome important for health outcomes, further investigation is required to understand whether and how parental SES measures should be integrated in population GWAS studies. Second, the continued use of the measures that have a strong theoretical, conceptual and measurement basis, such as occupational status in social stratification research, underlines the importance of precision phenotypes. Contrary to a previous GWAS that relied solely on a skill-based minimal occupational classification[27], our occupational status phenotypes, which have been developed by sociologists over decades, doubled the heritability using CAMSIS, increased SNP discovery by more than threefold and also provides a consistently meaningful interpretation of the outcome variable. This also emphasizes the genetic relevance of socially theorized measures and of social factors included in them, such as potential interaction or social prestige.

Finally, our findings embrace an interdisciplinary perspective when studying social stratification, mobility and status transmission. By further studying the underlying latent factor of individual socioeconomic status indicators, we can foster a better understanding of the genetic correlates of socioeconomic status and its broader implications for society. It is imperative to comprehend the role of indirect effects and passive gene–environment correlations in this puzzle, as well as the causes and consequences of assortative mating on these relationships. The dynamic nature of the

intergenerational transmission of socioeconomic status would be best served by a more comprehensive and rigorous social, historical and genetic approach.

Our study also has its limitations. The UK Biobank represents only 5.5% of the approached target population and over-represents individuals with lower mental health problems, BMI, non-smokers, with higher education and from less-economically deprived areas[45,76–79]. Consequently, participation bias affects the UK Biobank, limiting its generalizability and introducing the potential that observed genetic associations may be influenced by the characteristics of the subset of individuals who chose to participate in the UK Biobank[80]. We do show how our measures of occupational status differ from UK census data (Supplementary Information Section 6). We can expect environmental heterogeneity across different populations to challenge our findings. Accordingly, we use the NCDS sample, another UK dataset with different potential selection biases, to strengthen our analytic approach. While PGS predictions are nearly identical in our two British-European genetic ancestry populations, previous research has demonstrated that for educational attainment, only 50% of genetic effects are universal across seven Western populations[42]. Population genetic heterogeneity also limits the scope of this study beyond UK residents, since similar to the majority of GWAS so far, we focus on European-ancestry individuals in a Western country[81,82]. The integration of other ancestries, temporal, geographical and more diverse socioeconomic contexts is the future. The reduction of PGS prediction within families also emphasizes the relevance of recent initiatives for discovery designs using family data and to further study the role of assortative mating for within-family effect reduction[45]. It is particularly important since parental genetic factors could influence their children's occupational status through non-genetic mechanisms, and these effects might not be adequately captured when considering only the child's PGS. Therefore, we recognize the importance of including both parents' PGSs as control variables to estimate genetic confounding effects, but this was not possible using the current data. This underscores the need for additional research with multigenerational genetic and social survey data. Despite these limitations, the current study offers many new insights into the interplay between genetics and occupational status scores along with socioeconomic status.

## Methods

This Article has Supplementary Information with details about data and methods and additional detailed analyses. Extended Data Fig. 1 also provides an overview. We have also built the R package 'ukbjobs' that allows researchers to construct CAMSIS, ISEI and SIOPS occupational scores directly from the UK Biobank data (https://github.com/tobiaswolfram/ukbjobs).

### Ethics approval

This research was conducted using the UK Biobank under application 32696 and NCDS under application GDAC_2021_16_TROPF, with ethics approval from the University of Oxford under application SOC_R2_001_C1A_21_60. Both the UK Biobank and NCDS applications were specific to the scope of this paper. For the UK Biobank approval, we received approval for a scope extension to ensure transparency, allowing us to expand from our focus on non-standard occupations to also include occupational status. Here we specified that our plan was "to perform GWAS analysis using employment histories from the UK Biobank to construct sociologically informed measures of occupational status". We specified that we would construct sociologically informed measures of occupational status (CAMSIS, SIOPS and ISEI) for our GWAS and noted that the analysis would be accompanied by NCDS genetic and phenotypic data. For the NCDS application, we specified not only the information mentioned above but also the set of polygenic prediction analyses. We also preregistered our analysis plan (https://osf.io/djbr2/), which was updated for replication (https://osf.io/x6va5).

### UK Biobank

For both the discovery and prediction of occupational status measures, education and income, we used data from the UK Biobank. The UK Biobank is a large-scale biomedical database and research resource containing in-depth genetic and health information from 502,655 individuals recruited between 2006 and 2010. The database is globally accessible to approved researchers. Details of the UK Biobank genotyping procedure can be found elsewhere[83]. After phenotype selection and genetic quality control (performed in PLINK v.1.9, v.2), we conducted our analyses on 273,157 individuals (130,952 males, 142,205 females).

### NCDS

As a second, longitudinal UK prediction sample, we used the NCDS following 17,000 children born in Great Britain in 1 week in 1958. NCDS has been designed to examine the social and obstetric factors associated with stillbirth and death in early infancy. Overall, there were ten waves available (birth: 1958, age 7: 1965, age 11: 1969, age 16: 1974, age 23: 1981, age 33: 1991, age 42: 2000, age 46: 2004, age 50: 2008 and age 55: 2013).

### Phenotyping

'Socioeconomic differences'-based indices measure the 'attributes of occupations that convert a person's main resource (education) into a person's main reward (income)'[31]. The most commonly used measure is occupational prestige, termed the International Socioeconomic Index (ISEI)[31], which is constructed from scaling weights that maximize the (indirect) influence of education on income through occupation.

Other prestige-based measures are the result of public opinion surveys in which representative samples of the population are tasked with ranking occupations by their relative social standing. Emerging at a similar time as socioeconomic indices[84], Treiman[85] demonstrated that prestige-based measures were surprisingly constant over time and cultures, consolidating their use in social scientific research. The Standard International Occupational Prestige Scale (SIOPS or Treiman-prestige)[28] remains another commonly used metric in this tradition.

Lastly, occupational status indicators derived from 'social interaction' focus on the heterogeneity of associations between occupants of different jobs, following the tradition of refs. [86,87]. They are based on the idea that differential association is a function of social stratification since members of a group are more likely to interact within that group than with out-group members. Thus, acquaintances, friends and spouses are much more likely to be selected from within the same group than from outside. A group of Cambridge sociologists reversed this approach to measure social structure on the basis of interactions. The Cambridge Social Interaction and Stratification Scale (CAMSIS) measures the distance between occupations on the basis of the frequency of social interactions (operationalized as husband-and-wife combinations) between them[1].

Information on occupational status scales was merged to the occupational classification scheme utilized in the UK Biobank (the Standard Occupational Code version 2000 (SOC2000))[88]. CAMSIS-based status could be directly merged using the data available in ref. [89]. ISEI and SIOPS (as provided by the R package 'strat', R software v.4.2.0, v.4.1.2)[90], however, use the less granular ISCO-88 scale, so a mapping from ISCO to SOC was employed[91]. If multiple job codes for a respondent were available, the most recent job was used.

'Income' was measured similarly as in ref. [23] using a coarse, 5-level ordinal household income variable. Educational attainment was defined as years of education and coded according to the scheme provided in ref. [6].

The prestige of 'current' or 'most recent occupation' is treated as a continuous measure. In the initial discovery analysis using the UK Biobank, respondents were asked to provide job titles for the current or the most recent job held. The job information was coded using the four-digit UK SOC2000. We built a procedure to link the UK SOC2000

to ISCO-88(COM) and then derive ISEI and SIOPS from ISCO-88(COM). All phenotypes were inverse-normal rank transformed before analysis. In the NCDS, the SOC2000 code of the respondent's occupation (as well as their father's when they were 11 years old), is also available, thus the same procedure was applied.

In the NCDS, we looked at 'health' measured at ages 23, 33, 42, 46, 50 and 55. Participants were asked to rate their general health on a scale from 1 (excellent) to 4 (poor) (age 23 and 33), 1 (excellent) to 5 (very poor) (age 42) and 1 (excellent) to 5 (poor) (age 46, 50 and 55).

For each time point, the outcome was treated as metric and standardized to have a mean of zero and a standard deviation of 1. We then regressed it on the CAMSIS, ISEI and SIOPS PGSs, respectively, while controlling for sex and 10 principal components to correct for population stratification.

### Discovery

An analysis plan was preregistered and uploaded in February 2021 (https://osf.io/329pr/) and updated in February 2023 (https://osf.io/x6va5). All calculations were based on mixed-model association tests as implemented in the programme FastGWA[92], with association testing based on v.3 imputed data. Following the preposted open science analysis plan in each regression, the following covariates were included: the first 10 genomic principal components, age at assessment and age[2], UK Biobank (UKB) assessment centre at recruitment, sex and genotyping array (BiLEVE or Axiom) on the sample of British-European genetic ancestry. Chromosomes were analysed separately. To speed up the calculation of summary statistics, a minimum minor allele frequency (MAF) filter of 0.01 was imposed, leaving 10.2 million SNPs for the analysis. We supplemented our autosomal analyses with association analyses of SNPs on the X chromosome in a joint association analysis of both sexes.

### PheWAS

All 1,000 genome SNPs in linkage disequilibrium ($R^2 > 0.6$ for European ancestry) with the 106 independent SNPs were identified using FUMA[93]. For these 11,206 SNPs, 1,005,470 phenotypical associations reaching at least suggestive significance ($P < 5 \times 10^6$) in the GWAS catalogue and the IEU OpenGWAS project were collected[94]. All variants with at least one genome-wide significant link to a trait associated with education, income or any other socioeconomic outcome were removed, leaving 8 hits (rs12137794, rs17498867, rs10172968, rs7670291, rs26955, rs2279686, rs72744938, rs62058104) that have not yet been linked to any SES-related trait. For three variants (rs7670291, rs26955, rs72744938), not even suggestive associations ($P > 5 \times 10^{-6}$) were found.

### Univariate LDSC

Univariate LDSC regression was performed on the summary statistics from the GWAS to quantify the degree to which population stratification influenced the results and to estimate heritability (performed in Python v.3.8.4, v.3.9.15). For this, GWA test statistics were regressed onto the LD score of each SNP. LD scores were used with European genetic ancestry individuals and weights were downloaded from https://utexas.app.box.com/s/vkd36n197m8klbaio3yzoxsee6sxo11v. SNPs were included if they had a MAF of >0.01 and an imputation quality score of >0.9 and were available in the LD score file. Intercepts for all three occupational status measures were close to 1 (CAMSIS: 1.1193, s.e. = 0.013; SIOPS: 1.0845, s.e. = 0.011; ISEI: 1.0993, s.e. = 0.0125).

### MAGMA

To investigate the functional implications of the genetic variants associated with occupational status, we performed gene-based and gene-set analyses using MAGMA[43]. We used FUMA[93] to annotate, prioritize, visualize and interpret GWAS results, to run MAGMA on our summary statistics and to map SNPs to genes. We tested whether the genes prioritized by FUMA were enriched for expression in 30 general tissue types (GTEx v.8) and 53 specific tissue types (GTEx v.8) using MAGMA's

gene-set analysis. We observed a strong expression in all brain tissues compared with other tissues. No other tissue showed significant enrichment for gene expression.

### MTAG

MTAG (in Python v.2.7)[44] was used to meta-analyse all three occupational status measures with a secondary GWAS on household income in UKB and a secondary GWAS on educational attainment in UKB (for the validation subsample of siblings in UKB) or the third GWAS meta-analysis for education[6] excluding 23andMe participants as well as the NCDS cohort (for the validation using the NCDS data). This allowed us to leverage the high genetic correlations between the occupational status measures and income/education (see above) to increase power and detect variants, and improve prediction as outlined above and in Supplementary Information Section 10.

### GSEM

We used the infrastructure provided by the GenomicSEM package[34] to compute LDSC-based genetic covariances and correlations between our occupational status measures and education and income. SNPs were included using similar criteria as specified for univariate LDSC. Covariance structures between the three measures of occupational status were used as input in a genomic structural equation model to analyse their loading on a joint factor of occupational status (Supplementary Information Section 11.1) We furthermore applied a multivariate genetic regression model to the genetic covariance matrix of each of our occupational status measures and cognitive performance, ADHD, openness to experience, risk tolerance and neuroticism (Supplementary Information Section 11.3).

### Prediction analyses

Overall, we constructed three types of polygenic scores for each phenotype (Supplementary Information Section 10): (1) Pruning and thresholding using PRSice[95], (2) SBayesR[96] and (3) MTAG + SBayesR. In our prediction analyses, we residualized for sex, age (only in UKB) and 10 principal components before calculating the $R^2$. For the within-family analysis in UK Biobank, we identified a sample of siblings and computed family-fixed-effects regressions with both polygenic scores as well as phenotypes standardized beforehand, and interpreted the change in coefficients (Supplementary Information Section 12).

### Mediation analyses

NCDS respondents were asked at age 11 about the type of job they would like to do in the future. We coded these jobs to SOC2000, constructed their occupational status and ran mediation models in lavaan in R (v.4.2.0, v.4.1.2)[97] to quantify the share of the association between PGS and occupational status that can be attributed to occupational aspirations. We tested a comprehensive multiple mediation model, introducing cognitive ability, internalizing behaviour, scholastic motivation and externalizing behaviour as additional mediators (Supplementary Fig. 8).

### Confounding analyses

Within NCDS, information on the paternal occupation at age 12 was used to estimate the correlation between paternal and offspring occupational status at various ages for all three measures. We combined the approach of scaling the variance explained by polygenic scores, outlined in ref. 57, and integrated it into a mediation model to test which share of the intergenerational correlation for each of the three metrics was confounded by the corresponding polygenic score if we assumed that it only explained the amount of variance in our prediction analysis or the full SNP heritability (Supplementary Information Section 14).

### Reporting summary

Further information on research design is available in the Nature Portfolio Reporting Summary linked to this article.

## Data availability

The GWAS summary statistics generated in this study are available on the GWAS catalogue website (https://www.ebi.ac.uk/gwas/) under accession codes GCST90446160, GCST90446162 and GCST90446163. Access to the UK Biobank is available through: http://www.ukbiobank.ac.uk. Access to The National Child Development Study (NCDS) is available through: https://cls.ucl.ac.uk/data-access-training/. PheWAS analysis was performed using the IEU OpenGWAS project data available at: https://gwas.mrcieu.ac.uk. LDSC regression analysis was performed by using LD scores and weights available at: https://utexas.app.box.com/s/vkd36n197m8klbaio3yzoxsee6sxo11v. Analysis of the representativity of the UK Biobank with the Office of National Statistics (ONS) data was performed using publicly available ONS data which can be accessed at: https://www.ons.gov.uk/employmentandlabourmarket/peopleinwork/employmentandemployeetypes/datasets/employmentbyoccupationemp04. Source data are provided with this paper.

## Code availability

The R package 'ukbjobs' is available at https://github.com/tobiaswolfram/ukbjobs, https://doi.org/10.5281/zenodo.10061205. The package allows researchers to construct CAMSIS, ISEI and SIOPS occupational scores directly from the UK Biobank data. No other custom code was used; all analyses and modelling were performed using standard software as described in Methods and in Supplementary Information.

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

## Acknowledgements

This research was conducted using the UK Biobank under application 32696 and NCDS under application GDAC_2021_16_TROPF, with ethics approval from the University of Oxford under application SOC_R2_001_C1A_21_60. Both the UK Biobank and NCDS applications were specific to the scope of this paper. For the UK Biobank approval, we received approval for a scope extension to ensure transparency, allowing us to expand from our focus on non-standard occupations to also occupational status. Here we specified that our plan was: 'to perform GWAS analysis using employment histories from the UK Biobank to construct sociologically informed measures of occupational status'. We specified that we would construct sociologically informed measures of occupational status (CAMSIS, SIOPS and ISEI) for our GWAS and noted that the analysis would be accompanied by NCDS genetic and phenotypic data. For the NCDS application, we specified not only the information mentioned above but also the set of polygenic prediction analyses. We also preregistered our analysis plan (https://osf.io/djbr2/), which was updated for replication (https://osf.io/x6va5). Funding for this project for M.C.M. and E.T.A. was from the European Research Council ERC Advanced Grant CHRONO (835079), and the Leverhulme Trust (RC-2018–003) Leverhulme Centre for Demographic Science; for M.C.M., Economic and Social Research Council, United Kingdom Science and Innovation (UKRI) Connecting Generations Grant (ES/W002116/1), MapIneq Project, European Union's Horizon Europe research and innovation programme (No. 101061645); and for F.C.T., UKRI FINDME (EP/Y023080/1) and AnalytiXIN, which is primarily funded through the Lilly Endowment, IU Health and Eli Lilly and Company. The funders had no role in study design, data collection and analysis, decision to publish or preparation of the manuscript. The authors thank D. M. Brazel for his important contribution and comments at the initial stage of the discovery.

## Author contributions

M.C.M. and F.C.T. supervised the study and M.C.M. led in devising and preregistering the study in 2021. T.W. and F.C.T. wrote the paper, with extensive revisions and comments from M.C.M. and E.T.A. For the Supplementary Information, M.C.M. wrote 'Frequently Asked Questions' (jointly revised with E.T.A.) and 'Background', X.D. wrote 'Representativity of the UK Biobank with the Office of National Statistics', and E.T.A. and T.W. wrote the remainder, with comments from all authors. T.W. and E.T.A. conducted statistical analyses, with input from X.D. on representativeness of occupations. M.C.M. devised and prepared Extended Data Fig. 1. All authors reviewed and approved the final version of the paper.

## Competing interests

M.C.M. is a Trustee of the UK Biobank, is on the Scientific Advisory Board of Our Future Health and Lifelines Biobank and is on the Data Management Advisory Board of the Health and Retirement Survey. The remaining authors declare no competing interests. F.C.T. is a research fellow at AnalytiXIN, which is a consortium of health-data organizations, industry partners and university partners in Indiana primarily funded through the Lilly Endowment, IU Health and Eli Lilly and Company.

## Additional information

**Extended data** is available for this paper at https://doi.org/10.1038/s41562-024-02076-3.

**Correspondence and requests for materials** should be addressed to Evelina T. Akimova or Tobias Wolfram.

[1]Department of Sociology, Purdue University, West Lafayette, IN, USA. [2]Leverhulme Centre for Demographic Science, Nuffield Department of Population Health and Nuffield College, University of Oxford, Oxford, UK. [3]Department of Sociology, University of Bielefeld, Bielefeld, Germany. [4]WZB Berlin Social Science Center, Berlin, Germany. [5]Einstein Center Population Diversity, Berlin, Germany. [6]Centre for Longitudinal Studies, University College London, London, UK. [7]AnalytiXIN, Indianapolis, IN, USA. [8]Department of Genetics, University Medical Centre Groningen, Groningen, the Netherlands. [9]Department of Economics, Econometrics and Finance, University of Groningen, Groningen, the Netherlands. [10]These authors contributed equally: Evelina T. Akimova, Tobias Wolfram. [11]These authors jointly supervised this work: Felix C. Tropf, Melinda C. Mills. ✉e-mail: eakimova@purdue.edu; twolfram.eisenach@gmail.com

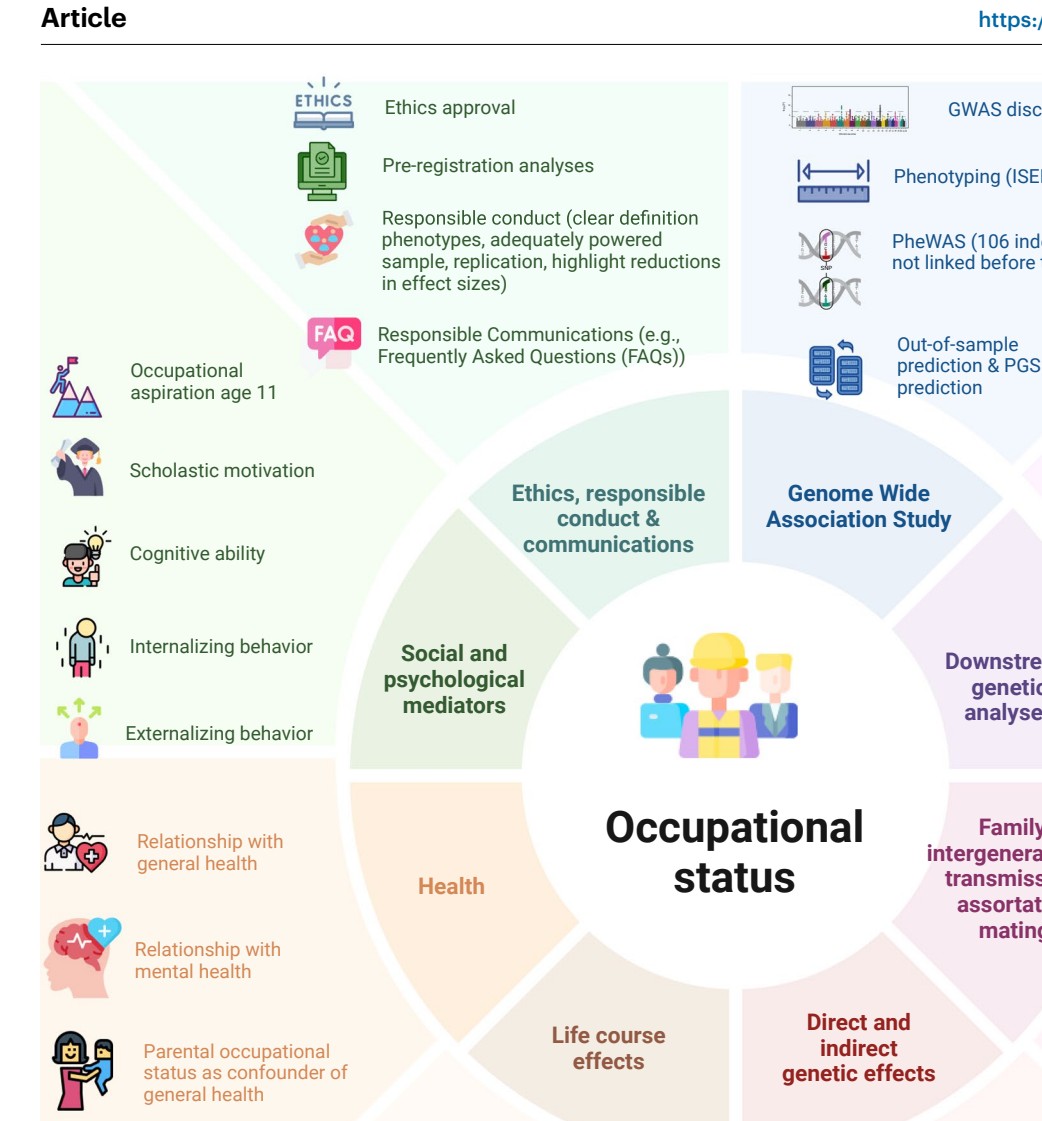
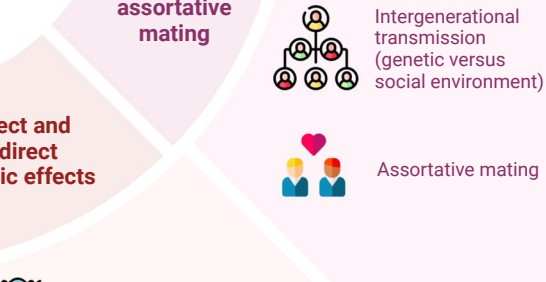

**Extended Data Fig. 1 | Study Summary Diagram.**

Tobias Wolfram

# Reporting Summary

## Statistics

For all statistical analyses, confirm that the following items are present in the figure legend, table legend, main text, or Methods section.

| n/a | Confirmed | |
|---|---|---|
| ☐ | ☒ | The exact sample size (*n*) for each experimental group/condition, given as a discrete number and unit of measurement |
| ☐ | ☒ | A statement on whether measurements were taken from distinct samples or whether the same sample was measured repeatedly |
| ☐ | ☒ | The statistical test(s) used AND whether they are one- or two-sided<br>*Only common tests should be described solely by name; describe more complex techniques in the Methods section.* |
| ☐ | ☒ | A description of all covariates tested |
| ☐ | ☒ | A description of any assumptions or corrections, such as tests of normality and adjustment for multiple comparisons |
| ☐ | ☒ | A full description of the statistical parameters including central tendency (e.g. means) or other basic estimates (e.g. regression coefficient) AND variation (e.g. standard deviation) or associated estimates of uncertainty (e.g. confidence intervals) |
| ☐ | ☒ | For null hypothesis testing, the test statistic (e.g. *F*, *t*, *r*) with confidence intervals, effect sizes, degrees of freedom and *P* value noted<br>*Give P values as exact values whenever suitable.* |
| ☐ | ☒ | For Bayesian analysis, information on the choice of priors and Markov chain Monte Carlo settings |
| ☒ | ☐ | For hierarchical and complex designs, identification of the appropriate level for tests and full reporting of outcomes |
| ☐ | ☒ | Estimates of effect sizes (e.g. Cohen's *d*, Pearson's *r*), indicating how they were calculated |

*Our web collection on statistics for biologists contains articles on many of the points above.*

## Software and code

Policy information about availability of computer code

| Data collection | No software used for data collection. |
|---|---|

| Data analysis | Analytical methods and software are described in the Methods section and in the Supplementary Information. Software used:<br>PLINK v1.9 (https://www.cog-genomics.org/plink/1.9/)<br>PLINK v2 (https://www.cog-genomics.org/plink/2.0/)<br>LDSC v1.0.1 (https://github.com/bulik/ldsc)<br>R versions v4.2.0, v4.1.2 (https://www.r-project.org)<br>Python v2.7, v3.8.4, v3.9.15 (https:www.anaconda.org)<br>FastGWA (https://yanglab.westlake.edu.cn/software/gcta/#fastGWA)<br>FUMA v1.5.2 (https://fuma.ctglab.nl)<br>MAGMA v1.10 (https://cncr.nl/research/magma/)<br>MTAG v0.9.0 (https:www.github.com/JonJala/mtag)<br>GenomicSEM v0.0.5 (https://github.com/GenomicSEM/GenomicSEM)<br>SBayesR v2.03 (https://cnsgenomics.com/software/gctb/#Overview)<br>PRSice v2 (https://choishingwan.github.io/PRSice/)<br>R-package "lavaan" v0.6-18 (https://cran.r-project.org/web/packages/lavaan/index.html)<br>R-package "strat" v0.1 (https://cran.r-project.org/web/packages/strat/index.html)<br><br>R-package "ukbjobs" available at https://github.com/tobiaswolfram/ukbjobs, https://doi.org/10.5281/zenodo.10061205. The package allows researchers to construct CAMSIS, ISEI, and SIOPS occupational scores directly from the UK Biobank data. |
|---|---|

For manuscripts utilizing custom algorithms or software that are central to the research but not yet described in published literature, software must be made available to editors and reviewers. We strongly encourage code deposition in a community repository (e.g. GitHub). See the Nature Portfolio guidelines for submitting code & software for further information.

# Data

Policy information about availability of data

All manuscripts must include a data availability statement. This statement should provide the following information, where applicable:
- Accession codes, unique identifiers, or web links for publicly available datasets
- A description of any restrictions on data availability
- For clinical datasets or third party data, please ensure that the statement adheres to our policy

The GWAS summary statistics generated in this study are available on the GWAS catalogue website (https://www.ebi.ac.uk/gwas/) under accession codes GCST90446160, GCST90446162, GCST90446163. Access to the UK Biobank is available through: http://www.ukbiobank.ac.uk. Access to The National Child Development Study (NCDS) is available through: https://cls.ucl.ac.uk/data-access-training/. PheWAS analyses was performed using the IEU OpenGWAS project data available at: https://gwas.mrcieu.ac.uk. LDSC regression analysis was performed by using LD scores and weights available at: https://utexas.app.box.com/s/vkd36n197m8klbaio3yzoxsee6sxo11v. Analysis of the representativity of the UK Biobank with the Office of National Statistics (ONS) data was performed using publicly available ONS data which can be accessed at: https://www.ons.gov.uk/employmentandlabourmarket/peopleinwork/employmentandemployeetypes/datasets/employmentbyoccupationemp04. Source data are provided with this paper.

# Research involving human participants, their data, or biological material

Policy information about studies with human participants or human data. See also policy information about sex, gender (identity/presentation), and sexual orientation and race, ethnicity and racism.

| Reporting on sex and gender | All analyses were performed in sex combined models. We do not directly measure gender and accordingly do not report on it. |
|---|---|
| Reporting on race, ethnicity, or other socially relevant groupings | Our research sample contains British-European genetic ancestry only; we reflect on this in Box 1 and Discussion. We also developed an extensive FAQs along with a "key-points" section and are transparent in our use of genetic ancestry term. Genetic ancestry was determined based on principal components (PC) analysis of the genetic data. We focus on individuals of British-European genetic ancestry in order to decrease the risk of confounding due to population stratification. |
| Population characteristics | Population characteristics for both samples are described in the "Behavioural & social sciences study design" section below. |
| Recruitment | Recruitment was performed independently by UK Biobank and National Child Development Study (NCDS). |
| Ethics oversight | This research was conducted using the UK Biobank under application 32696 and NCDS under application GDAC_2021_16_TROPF, with ethical approval from the University of Oxford under application SOC_R2_001_C1A_21_60. Both the UK Biobank and NCDS applications were specific to the scope of this paper. For the UK Biobank approval, we received approval for a scope extension to ensure transparency, allowing us to expand from our focus on non-standard occupations to also include occupational status. Here we specified that our plan was: "to perform GWAS analysis using employment histories from the UK Biobank to construct sociologically informed measures of occupational status." We specified that we would construct sociologically informed measures of occupational status (CAMSIS, SIOPS, and ISEA) for our GWAS and noted that the analysis would be accompanied by NCDS genetic and phenotypic data. For the NCDS application, we specified not only the information mentioned above but also the set of polygenic prediction analyses. We also preregistered our analysis plan (https://osf.io/djbr2/) which was updated for replication (https://osf.io/x6va5). |

Note that full information on the approval of the study protocol must also be provided in the manuscript.

# Field-specific reporting

Please select the one below that is the best fit for your research. If you are not sure, read the appropriate sections before making your selection.

☐ Life sciences      ☒ Behavioural & social sciences      ☐ Ecological, evolutionary & environmental sciences

For a reference copy of the document with all sections, see nature.com/documents/nr-reporting-summary-flat.pdf

# Behavioural & social sciences study design

All studies must disclose on these points even when the disclosure is negative.

| | |
|---|---|
| Study description | This is a genome-wide association study (GWAS) on sociologically informed occupational status measures (ISEI, SIOPS, and CAMSIS) using the UKBiobank with multiple analytic approaches employed including genomic structural equation models (GSEM), multi-trait analysis (MTAG), sibling and adoption models (the full list of the approaches and the motivation can be found in Supplementary Information Section 1). |
| Research sample | The research sample includes two data sources. First, the UK Biobank is a large-scale biomedical database and research resource, containing in-depth genetic and health information from 502,655 individuals recruited between 2006 and 2010. More information is available at: http://www.ukbiobank.ac.uk. Second, the National Child Development Study (NCDS) follows 17,000 children born in Great Britain in one week in 1958. More information is available at: https://ncds.info.<br><br>Overall, the research sample in total consists of 273,157 (130,952 males, 142,205 females) and 271,769 (130,129 males, 141,640 females) individuals for the occupational status phenotypes (CAMSIS and SIOPS/ISEI) and 353,673 (169,201 males, 184,472 females) and 404,420 (185,632 males, 218,788 females) individuals for the secondary analyses (household income and education), respectively. To validate our findings, we replicated our top hits using the genotyped subsample of the NCDS, including approximately 6,500 individuals with both genetic and phenotypic information. UK Biobank participants were between 40 and 69 years of age at the time of their recruitment between 2006 and 2010. For the NCDS, since it is a longitudinal study, observations for current occupations were pooled over all waves starting at age 33 (N = 5,389; 5,312; 5,211; 4,902; 4,263 for CAMSIS at age 33, 42, 46, 50, and 55, N = 5,449; 5,293; 5,197; 4,892; 4,252 for ISEI/SIOPS).<br><br>The rationale for using these samples is the following: the UK Biobank has the required large sample size and detailed occupational codes. We then replicated our results using the NCDS sample. This dataset was chosen because it is a similar UK cohort, which is important since previous research has demonstrated genetic variation by country and birth cohort for complex behavioral phenotypes.<br><br>The UK Biobank is not a nationally representative study; NCDS is a cohort study and representative for its respective birth cohort in the UK (born in 1958). |
| Sampling strategy | To obtain the largest samples possible for both discovery and replication that would cover participants of roughly the same age and with detailed occupation information from the United Kingdom. |
| Data collection | Data collection was performed independently by the UK Biobank and the NCDS. Both are observational studies used for secondary data analysis. Since this is not a controlled randomized study, there was no step involved equivalent to blinding. |
| Timing | UK Biobank and NCDS have variable data collection time-periods. UK Biobank recruited individuals between 2006 and 2010; NCDS is a cohort study and includes those born in one week in 1958. |
| Data exclusions | Any observations without SOC2000 occupational information were excluded; genetic and phenotypic quality controls were implemented as well (Supplementary Information 7.2 describes them in detail). We also restricted our analytic sample to British-European genetic ancestry only. Overall, we excluded 229,462 individuals for CAMSIS analyses, 230,851 individuals for ISEI/SIOPS analyses; 148,947 and 98,200 individuals for the secondary analyses - household income and education respectively. |
| Non-participation | The UK Biobank response rate was 5.5%. In the NCDS, the response rate at the first sweep was 98.7%. Additionally, 25.8% have participated in all 11 sweeps, and 60.5% have taken part in 7 or more sweeps. Participants were able to select 'Prefer not to answer' options through the questionnaires in the UK Biobank and 'Refusal' options in the NCDS. |
| Randomization | Participants were not allocated into experimental groups. |

# Reporting for specific materials, systems and methods

We require information from authors about some types of materials, experimental systems and methods used in many studies. Here, indicate whether each material, system or method listed is relevant to your study. If you are not sure if a list item applies to your research, read the appropriate section before selecting a response.

## Materials & experimental systems

| n/a | Involved in the study |
|---|---|
| ☒ ☐ | Antibodies |
| ☒ ☐ | Eukaryotic cell lines |
| ☒ ☐ | Palaeontology and archaeology |
| ☒ ☐ | Animals and other organisms |
| ☒ ☐ | Clinical data |
| ☒ ☐ | Dual use research of concern |
| ☒ ☐ | Plants |

## Methods

| n/a | Involved in the study |
|---|---|
| ☒ ☐ | ChIP-seq |
| ☒ ☐ | Flow cytometry |
| ☒ ☐ | MRI-based neuroimaging |

## Plants

| | |
|---|---|
| Seed stocks | *Report on the source of all seed stocks or other plant material used. If applicable, state the seed stock centre and catalogue number. If plant specimens were collected from the field, describe the collection location, date and sampling procedures.* |
| Novel plant genotypes | *Describe the methods by which all novel plant genotypes were produced. This includes those generated by transgenic approaches, gene editing, chemical/radiation-based mutagenesis and hybridization. For transgenic lines, describe the transformation method, the number of independent lines analyzed and the generation upon which experiments were performed. For gene-edited lines, describe the editor used, the endogenous sequence targeted for editing, the targeting guide RNA sequence (if applicable) and how the editor was applied.* |
| Authentication | *Describe any authentication procedures for each seed stock used or novel genotype generated. Describe any experiments used to assess the effect of a mutation and, where applicable, how potential secondary effects (e.g. second site T-DNA insertions, mosiacism, off-target gene editing) were examined.* |

