## [Peer Review File · Nature Human Behaviour]

Polygenic prediction of occupational status GWAS elucidates genetic & environmental interplay in intergenerational transmission, careers & health in UK Biobank

Corresponding Author: Dr Evelina Akimova

Version 0:

Decision Letter:

5th April 2023

Dear Dr Akimova

Thank you for submitting your manuscript entitled, "GWAS of occupational status and prestige identifies 106 genetic variants and defines their role for intergenerational transmission and life course", and for your patience while awaiting an initial decision.

While we appreciate your work, we are unable to consider GWAS studies without replication in a separate dataset (do let me know if I missed this though). As such, I regret that we cannot offer to publish this work in Nature Human Behaviour.

Please be assured that this editorial decision does not represent a criticism of the quality of your work, nor are we questioning its value to others working in this area. We hope that you will rapidly receive a more favourable response elsewhere.

I am sorry that we cannot respond more positively on this occasion, and hope that the negative outcome in this instance will not deter you from submitting future work to Nature Human Behaviour.

Sincerely,

[REDACTED]

I suggest that you consider Communications Biology as a suitable venue for your work. To transfer your manuscript there, please use our manuscript transfer portal. You will not have to re-supply manuscript metadata and files, unless you wish to make modifications, but please note that this link can only be used once and remains active until used. For more information, please see our [manuscript transfer FAQ](http://www.nature.com/authors/author_resources/transfer_manuscripts.html?WT.mc_id=EMI_NPG_1511_AUTHORTRANSF&WT.ec_id=AUTHOR) page.

Note that any decision to opt in to In Review at the original journal is not sent to the receiving journal on transfer. You can opt in to [In Review](https://www.nature.com/nature-portfolio/for-authors/in-review) at receiving journals that support this service by choosing to modify your manuscript on transfer. In Review is available for primary research manuscript types only.

Version 1:

Decision Letter:

24th July 2023

Dear Dr Akimova,

Thank you once again for your manuscript, entitled "Polygenic predictions of occupational status GWAS elucidate genetic and environmental interplay for status transmission, careers, and health", and for your patience during the peer review process.

Your Article has now been evaluated by 4 referees. You will see from their comments copied below that, although they find your work of [considerable] potential interest, they have raised quite substantial concerns. In light of these comments, we cannot accept the manuscript for publication, but would be interested in considering a revised version if you are willing and able to fully address reviewer and editorial concerns.

We hope you will find the referees' comments useful as you decide how to proceed. If you wish to submit a substantially revised manuscript, please bear in mind that we will be reluctant to approach the referees again in the absence of major revisions. We are committed to providing a fair and constructive peer-review process. Do not hesitate to contact us if there are specific requests from the reviewers that you believe are technically impossible or unlikely to yield a meaningful outcome.

As you will see, the reviewers suggest a number of additional analyses to increase the robustness of your results. We ask that you perform all of them and include results in your revision. Could you also provide information on how you accessed the NCDS data and the associated approval.

If you wish to submit a suitably revised manuscript, we would hope to receive it within 3 months. I would be grateful if you could contact us as soon as possible if you foresee difficulties with meeting this target resubmission date.

- Include a "Response to the editors and reviewers" document detailing, point-by-point, how you addressed each editor and referee comment. If no action was taken to address a point, you must provide a compelling argument. When formatting this document, please respond to each reviewer comment individually, including the full text of the reviewer comment verbatim followed by your response to the individual point. This response will be used by the editors to evaluate your revision and sent back to the reviewers along with the revised manuscript.
- Highlight all changes made to your manuscript or provide us with a version that tracks changes.

Link Redacted

Thank you for the opportunity to review your work. Please do not hesitate to contact me if you have any questions or would like to discuss the required revisions further.

Sincerely,

[REDACTED]

Reviewer expertise:

Reviewer #1: sociology, genetics

Reviewer #2: behavioral genetics, cognitive development

Reviewer #3: stat genetics, gwas,

Reviewer #4: behav genetics, econ

REVIEWER COMMENTS:

Reviewer #1:

Remarks to the Author:

This paper investigates genetic associations with occupational status and gene-environment interplay for intergenerational status transmission. The research questions are interesting and the findings are important for multiple disciplines, including human genetics and social science research in general. I have several comments that hopefully useful to improve the current manuscript.

First, it is uncertain how some of the key variables were measured in the study (e.g., cognitive performance, scholastic motivation, externalizing behavior etc.). It'll be great if a table can be added to the supplemental materials to describe the definitions of the key variables used in the analysis.

Second, the results on three occupational status variables (S Figure 7) consistently show non-linear patterns of the genetic association with occupational status (i.e., decreases after age 33 but increases after age 42). I'd suggest the authors give more discussions on these findings. What are possible explanations for this pattern and to what extent it's attributable to measurement issues (e.g., measures were not available in a 10-year window between 33 and 42)?

Third, related to the second point, another interesting finding is from the trajectory analysis. The results show that occupational status of individuals at the lower end of the PGS distribution experienced an early increase but then stabilized (or even decreased), whereas those at the higher end had an opposite pattern. I'd like to see more investigations on this result. For example, who are those at the lower and higher ends of the PGS distribution and why they show divergent trajectories in their occupational status? That'll help to understand the results better.

Fourth, in terms of the social mechanisms linking genetics and occupational status, in addition to the factors investigated in the analysis, health status and delinquency at early life are two other possible mediators. The authors have shown that the association between PGS and general health is reduced when parental occupational status is included. Another possibility is that health status at early life mediates the association between genetics and occupational status. Moreover, there is evidence that delinquency and early-life criminal justice involvement mediate the genetic association with educational attainments. If appropriate measures are available in the data, the authors may explore those additional social mechanisms.

Fifth, another key finding is that 38% of the intergenerational transmission of occupational status can be attributed to genetic inheritance and 62% is to non-genetic inheritance. Based on the description, these estimates seem to be based on the reduction of the parent-child correlation in occupational status after adjusting for the child's PGS rescaled based on the SNP heritability. Yet only including the child's PGS does not fully take into account all possibilities of genetic confounding. Parental genetic factors can be linked to children's occupational

status via non-genetic mechanisms (i.e., genetic nurturing effects) and these effects are not captured by child's PGS. Therefore, to estimate genetic confounding effects, both parents' PGSs are better options as control variables. I'd encourage the authors to perform this analysis or at least to discuss the limitations if data (i.e., parents' genotypes) are not available.

Reviewer #2:

Remarks to the Author:

The authors report the results of a GWA analysis of occupational status (OS) measures using UKB data. The GWA analysis identified 106 associations of which 8 are novel. The OS measures yield a SNP heritability of 10-15% and polygenic scores derived from their GWA summary statistics predicted 5-8% of the OS variance in an independent sample.

Although several GWA analyses of SES variables, especially educational attainment, have been reported, this GWA report of occupational status per se follows up a previous GWA report of occupational status in UKB (Ko et al., 2022). The present paper analyzes three indices of occupational status and shows that they are highly correlated genetically and could be aggregated in a single index of occupational status (OS).

This is a thorough analysis of the UKB data, with replication of key findings in other data sets, as well as a within-family analysis using sib pairs in UKB.

I have only one issue, which is the relationship of the present analysis to the previous UKB GWA analysis of OS by Ko et al. The authors say Ko et al. used 'skill-based occupational groups' and say that 'skill-based measures have been criticized' and contrasted Ko et al.'s measure with their use of '353 categories'. However, this seems a bit unfair to Ko et al., who analysed a measure based on the UK Standard Occupational Classification (OSC) scheme which was devised by the UK Office for National Statistics and begins with the 353-unit groups mentioned by the current authors. Importantly, the OSC makes it possible to treat OS as a continuous variable, which provides more robust analyses than the analysis of categories. The OSC has been used in thousands of studies, so it is an important measure to analyse – and to analyse in relation to the three OS measures used in the present paper. One of the present study's findings is that their three OS indices yielded very high genetic intercorrelations. So, I bet that the present paper's composite measure of OS correlates very highly with the Ko et al. OSC measure. Could the authors calculate the phenotypic and genetic correlations between their OS index and the OSC index used by Ko et al.? Regardless of the overlap with Ko et al., as the authors point out, their OS index doubled the heritability using CAMSIS and increased SNP discovery by more than threefold. But their paper should be seen as an extension of the Ko et al. analysis.

The paper is well written and reports other interesting analyses that replicate and extend other publications. For example, the authors report that intergenerational transmission of OS is more than half due to shared family environment.

Another finding that replicates findings in the literature is that the between-family association between polygenic scores and OS is halved when the association is analyzed within families. The authors suggest that assortative mating is responsible for much of this reduction, which indicates that so-called indirect effects need to take assortative mating into account. Could the authors test this hypothesis using spouses in UKB? What is the assortative mating coefficient for the authors' three indices of occupational status?

A novel and intriguing finding is the compression of genetic prediction in mid-career, although I wasn't sure what to make of this. Could the authors unpack this some more? I note that they did not include this finding in their FAQ ('What are the big take-away findings'), where they might have explained it in simpler terms.

The Discussion is well written and raises intriguing issues. The Discussion, Box 1 and FAQ in the Supplement will attenuate some of the blowback that might come from showing the importance of genetics in the traditional sociological domain of occupational status.

One very minor quibble: The Discussion includes this sentence: 'As often noted, however, twin studies likely also overestimate heritability due to the violation of the assumption that identical and fraternal twins share environmental influences to the same extent.' However, this is unreferenced and, to my knowledge, many tests of the equal environment assumption using different designs show that the assumption holds up quite well.

Reviewer #3:

Remarks to the Author:

This is an exceptional well thought through and conducted study. It should be regarded as the gold standard as to what questions regarding SES can be addressed using genetic data as well as the best way to address them. My comments are minor but I hope contribute in some way to polishing a fine piece of scientific literature.

In the abstract and throughout the manuscript 106 genetic variants are described as being identified. Either loci or independent SNPs would be preferable here as there are far more than 106 associated variants. Furthermore it is stated that eight of these variants has not been previously associated with differences in SES. Could the authors clarify if these eight loci harbour variants that have previously been associated with differences in SES?

Additionally, it is stated that FUMA was used to identify these 106 SNPs. However, in the results it is stated that the R2 threshold was 0.1. The documentation of FUMA is poor here but FUMA uses two rounds of clumping. The first is conducted in exactly the same manner as PLINK and is used to define independent significant SNPs as well as the border of the loci that they are in. A second round of clumping (by default at a lower R2) is then applied on these independent significant SNPs to identify what FUMA calls lead SNPs.

My question is was the second R2 equal to the first? If so the independent significant SNPs are the same as the lead SNPs. Also why does the R2 change between defining independent SNPs/loci and examining if these same SNPs are associated with other traits (In the PHE was the R2 = 0.6)?

The use of MTAG seems somewhat questionable here. I do see the increase in the number of genome-wide significant loci but I don't understand how this has occurred. Specifically the authors have near (genetically) identical traits (the r_g between them is >0.8) and the GWAS are performed on the same samples (If the education data set used is from Lee et al. there would be new participants but it seems the data used are all from UKBB). Could the authors provide the mean χ^2 and effective N_s pre and post MTAG? It would help confirm if MTAG is of benefit.

Also regarding the mean χ^2 . I do see that the authors have used fast-GWA-GLMM (again minor but I believe the title of the paper is incorrect) to account for relatedness and reported the LDSC intercepts which all seem close enough to 1. However, by including the mean

χ^2 it would be possible to observe the degree to which non-polygenic effects have contributed to the signal. In the discussion it is stated "Certain genetic traits may predispose individuals to achieve higher levels in these areas through biological pleiotropy (Hill et al. (2019)". I think you mean vertical pleiotropy here. Hill et al. (2019) write "more biologically proximal/distal phenotypes" but the schematic in Figure 1 of Hill et al. (2019) clearly describes vertical (mediated) pleiotropy.

Reviewer #4:

Remarks to the Author:

This study will be of great interest to behavioural geneticists and social scientists in general, and I am sure it will become highly-cited. The discovery of 106 variants (8 of which are novel), significantly expands our knowledge of genetic influence on behaviour, and the development of (relatively) highly-predictive polygenic scores will find many applications in future studies. The outstanding feature of the study is its meticulous phenotyping, using reliable measures that have been traditionally confined to sociological work. Consequently, the paper is also a great example of interdisciplinary work, combining behavioural genetics and sociology. The detailed phenotyping is also the critical improvement compared to the previous GWAS of social status (Ko et al., 2022), an evidence-based advance of scale and rigour, as defined by the standards of Nature Human Behaviour. A few comments:

- 1) The second sentence in Box 1 implies that the linking of biology to intelligence, criminality and status, is invalid. Given the recent advances in sociogenomics (including the current study), this position appears tenuous and is being contradicted by the rest of the paper. I suggest re-writing the sentence in a more neutral manner.
- 2) It is interesting that all three measures of occupational status showed similar heritabilities. Have the authors considered validating the LDSC heritability estimates using another method, such as GREML?
- 3) The reported incremental R^2 of 7.7% is that of the MTAG predictor. Is it fair to report this as the main result of the PGS analysis, given that MTAG also makes use of summary statistic from previous GWAS (education and income)? At the very least, it should be made clear from early in the paper, that this estimate comes from MTAG using education and income.
- 4) The authors use a clever method by Lee et al. to indicate that the reduction in PGS within-family prediction can be entirely due to assortative mating. While this may well be true, work by Demange et al. (2022) shows that adoptee prediction should not be attenuated by population stratification or assortative mating (see supplementary note 6 of the former study). In the current study, the adoptee design attenuation is very close to that of the sibling design, suggesting a role for indirect parental effects (I would urge the authors to use this term instead of "genetic nurture", which is no longer favoured), rather than AM or stratification. This would make sense, given that the phenotype in question (occupation) is probably greatly influenced by family background. The Demange study was on cognitive and non-cognitive skills, and it showed the opposite result, namely much less attenuation in the adoption design compared to the other designs (suggesting that indirect effects were mostly due to AM/stratification). Perhaps it would be useful to compare and contrast these two findings.
- 5) In the GenomicSEM analysis, it is shown that much of the genetic effect on social status is mediated by other factors, including risk tolerance. It is stated in the supplement that risk tolerance, perhaps surprisingly, has a positive effect on social status. I believe this is worth reporting in the main paper.
- 6) The measure that is defined by interaction networks (CAMSIS) is the one with the strongest genetic signal. According to the authors: "A potential reason for this observation may be genetic selection into interaction networks". I would rather argue that these interaction networks eventually become genetic networks, since by definition they include spouses. R.A. Fisher operationally defined social class as a spousal network (see a recent PNAS commentary by James Lee which provides the full quote). The authors should discuss this process more.
- 7) The sentence "Second, when examining the genetics related to the intergenerational transmission of SES, mostly heritability studies on educational attainment assumed that genetic influences are stable in absolute terms and environmentally driven inequality reduces with lower intergenerational correlations" is poorly written and hard to comprehend. Please rephrase.
- 8) "As often noted, however, twin studies likely also overestimate heritability due to the violation of the assumption that identical and fraternal twins share environmental influences to the same extent." I do not think there is enough evidence to render the violation of the equal environments assumption "likely". Also, one of the authors has a preprint on how heritability may also be underestimated in twin studies (Wolfram & Morris). Please either remove or rewrite the sentence.
- 9) Minor point, there is a reference to "reproductive behaviour" in page 8 of the supplement, which should probably be replaced by "occupational status".

Version 2:

Decision Letter:

Our ref: NATHUMBEHAV-23031006B

10th May 2024

Dear Dr. Akimova,

Thank you for submitting your revised manuscript "Polygenic prediction of occupational status GWAS elucidate genetic and environmental interplay for intergenerational transmission, careers, and health" (NATHUMBEHAV-23031006B). It has now been seen by the original referees and an additional bioethics reviewer ((Reviewer 5), and their comments are below. As you can see, the reviewers find that the paper has improved in revision. We will therefore be happy in principle to publish it in Nature Human Behaviour, pending revisions to satisfy the referees' final requests (in particular those raised by our ethical reviewer, Reviewer 5) and to comply with our editorial and formatting guidelines.

We are now performing detailed checks on your paper and will send you a checklist detailing our editorial and formatting requirements within two weeks. Please do not upload the final materials and make any revisions until you receive this additional information from us.

Sincerely,

[REDACTED]

Reviewer #1 (Remarks to the Author):

Thanks for addressing all my comments! The manuscript has been much improved after revision. This will be a significant contribution to the literature.

Reviewer #2 (Remarks to the Author):

The authors have addressed my comments well. They reframed their paper as an extension of Ko et al. and calculated phenotypic and genetic correlations with the measures used by Ko et al. The genetic correlations are > 0.95 , which might make readers wonder how the authors' SNP heritability can be twice as large as Ko et al.'s estimate when the measures are nearly identical genetically. The authors might forestall this concern by noting that genetic correlations are independent of heritability – that is, one measure could be highly heritable and the other measure modestly heritable, but their genetic correlation could be 1.0.

The new analyses of assortative mating using spousal data from UK Biobank confirm their hypothesis that assortative mating accounts in part for the reduction in effects from between families to within families. Spousal phenotypic correlations are from 0.22 to 0.33 and spousal polygenic score correlations are about 0.10.

The authors have also helpfully unpacked their intriguing finding about the compression of genetic prediction in mid-career.

Reviewer #3 (Remarks to the Author):

Reviewer #3

Remarks to the Author:

This is an exceptional well thought through and conducted study. It should be regarded as the gold standard as to what questions regarding SES can be addressed using genetic data as well as the best way to address them. My comments are minor but I hope contribute in some way to polishing a fine piece of scientific literature.

Answer:

We want to express our sincere gratitude for such kind words! Your acknowledgment of our work is nothing but truly inspiring. We also appreciate

your comments, which, though minor, undoubtedly contributed to refining our manuscript – we were able to perform additional tests to address your concerns and further clarified methods and statistical tools we used.

Reviewer #3

In the abstract and throughout the manuscript 106 genetic variants are described as being identified. Either loci or independent SNPs would be preferable here as there are far more than 106 associated variants. Furthermore, it is stated that eight of these variants has not been previously associated with differences in SES. Could the authors clarify if these eight loci harbour variants that have previously been associated with differences in SES?

Answer:

We thank the reviewer for this thoughtful comment and have now made the necessary amendments to address these concerns. To provide clarity and precision, we have updated the terminology in the main text and the appendix, replacing "genetic variants" with "independent SNPs".

With respect to the eight SNPs not previously associated with SES differences, we accordingly conducted a comprehensive assessment. Utilizing FUMA, we identified all 1000 Genomes SNPs in linkage disequilibrium (with $R^2 > 0.6$ for European ancestry) related to the 106 independent SNPs. This yielded a total of 11,206 SNPs. Subsequent analysis, referencing the GWAS catalogue and the IEU OpenGWAS project, revealed 1,005,470 phenotypical associations that reached at least suggestive significance ($p < 5 \times 10^{-8}$). We then excluded any variants having a genome-wide significant association with traits related to education, income, or other socioeconomic outcomes. This filtering process culminated in the identification of 8 unique hits (rs12137794, rs17498867, rs10172968, rs7670291, rs26955, rs2279686, rs72744938, rs62058104). Overall, none of these eight SNPs is in LD (at $R^2 > 0.6$) with a SNP that is present in 1KG and has been noted in the GWAS catalogue or IEU OpenGWAS to harbour associations with income, years of schooling, qualifications or similar socioeconomic status related traits at $p < 5 \times 10^{-8}$. These findings are in the Table B8 of the Supplementary Tables file (Supplementary Tables | B8. PheWAS of Top Hits).

Reviewer #3

Perfect.

Reviewer #3

Additionally, it is stated that FUMA was used to identify these 106 SNPs. However, in the results it is stated that the R2 threshold was 0.1. The documentation of FUMA is poor here but FUMA uses two rounds of clumping. The first is conducted in exactly the same manner as PLINK and is used to define independent significant SNPs as well as the border of the loci that they are in. A second round of clumping (by default at a lower R2) is then applied on these independent significant SNPs to identify what FUMA calls lead SNPs.

My question is was the second R2 equal to the first? If so the independent significant SNPs are the same as the lead SNPs. Also why does the R2 change between defining independent SNPs/loci and examining if these same SNPs are associated with other traits (In the PHE was the R2 = 0.6)?

Answer:

We thank the reviewer for raising this point and for providing us with the opportunity to clarify the methods we used. To eliminate any confusion regarding the FUMA method and its application, we have now updated Section 6.3 of the Supplementary Information (Supplementary Information| P.18):

"7.3 Findings

After inflating the standard errors by the square root of their respective intercepts from LD Score regressions, our GWASs identified 106 independent SNPs for CAMSIS including 56 found also for ISEI and 51 for SIOPS based on clumping all genome-wide significant SNPs using an R^2 -threshold of 0.1 and a window-size of 1000kb, one of which (only significant for CAMSIS) was found on the X-chromosome (see Figure 2 in the main text for the Manhattan plot of the autosome). In exploratory sex-specific GWAS, no separate hits emerged, with genetic correlations being very close to and not significantly different from one."

As the reviewer can see now, FUMA was not employed to identify the independent SNPs. Instead, for this task, we utilized the `--clump` command in PLINK. We set an R^2 -threshold of 0.1 and a window-size of 1000kb, targeting all genome-wide significant hits ($p < 5 \times 10^{-8}$). We then turned to FUMA, but solely with the objective of identifying all SNPs present in the 1000 Genomes dataset that are in linkage disequilibrium (LD) with our previously identified 106 independent SNPs, as detailed in our response to the comment above.

Reviewer #3

Comparing the documentation of FUMA to the documentation of PLINK it seems that what the authors have done will result in a greater number of loci. Using PLINK clumping the distance between two independent loci will be the distance between the two genome-wide significant SNPs that define those loci. However, using FUMA the distance between two loci will be the distance between the closest SNPs that are in LD with the genome-wide significant SNPs that define those loci. So, what clumping would call two separate loci FUMA would call a single locus. Have the authors compared the difference between these two methods? If so how did they select clumping by PLINK rather than by FUMA?

Reviewer #3

The use of MTAG seems somewhat questionable here. I do see the increase in the number of genome-wide significant loci but I don't understand how this has occurred. Specifically, the authors have near (genetically) identical traits (the r_G between them is >0.8) and the GWAS are performed on the same samples (If the education data set used is from Lee et al. there would be new participants, but it seems the data used are all from UKBB). Could the authors provide the mean χ^2 and effective N_s pre and post MTAG? It would help confirm if MTAG is of benefit.

Answer:

These are important points, and we would like to clarify our approach. In the case of the NCDS, we utilized the EA3 GWAS dataset, excluding data from 23andMe and NCDS itself. When conducting validation within the UKB dataset, we exclusively employed the UKB GWAS dataset and ensured the exclusion of siblings or adoptees for specific analyses. The observed improvement in our results can be attributed to variations in sample size and heritability. The sample sizes for income, and particularly for educational attainment (EA), are substantially larger. Regarding heritability, EA exhibited the highest values, followed by CAMSIS. This hierarchical pattern of heritability is also reflected in the relative increases in effective sample size, which are largest for SIOPS/ISEI and smallest for EA, as displayed in the following table below.

Response Table 1. MTAG: χ^2 and sample sizes

Trait GWAS_mean_ χ^2 MTAG_mean_ χ^2 GWAS_

MTAG_eff_N

CAMSIS 1.738 2.508 273157 697176

ISEI 1.568 2.418 271769 817666

SIOPS 1.534 2.435 271769 863640

Income 1.639 1.972 353673 721973

Years of Education (EA3) 2.074 2.584 763541 886012

Reviewer #3

I understand the authors point but my question is if there is any benefit from having CAMSIS, ISEI & SIOPS included in the same MTAG analysis. These data sets overlap considerably and the magnitude of the genetic correlation between them ($r_g > 0.95$) would indicate that there is very little additional information added.

Reviewer #3

Also, regarding the mean χ^2 . I do see that the authors have used fast-GWAGLMM (again minor but I believe the title of the paper is incorrect) to account for relatedness and reported the LDSC intercepts which all seem close enough to 1. However, by including the mean χ^2 it would be possible to observe the degree to which non-polygenic effects have contributed to the signal.

Answer: We apologize for any confusion and would like to clarify our methodology. We indeed employed fastGWA for our analyses; however, since all our phenotypes are continuous, we did not use fastGWA-GLMM, which is an extension tailored for binary phenotypes.

Regarding the title, if you are referring to our original title - 'Genome-wide association study of occupational status and prestige identifies 106 genetic variants and defines their role for intergenerational status transmission and the life course,' we have made revisions. The title is now 'Polygenic prediction of occupational status GWAS elucidate genetic and environmental interplay for intergenerational transmission, careers, and health.' If your concern revolves around the term 'polygenic,' we are open to further adjustments for even greater clarity.

In response to the concern raised about the mean χ^2 , we agree with the assessment. Below, we provide the values for the mean χ^2 . The observed inflation, which ranges between 5% and 7.5%, suggests that potential confounders such as population stratification have a limited impact. This finding further supports our interpretation that the predominant portion of the observed signal is polygenic. We accordingly added a new section to our supplementary materials (Supplementary Information| P.20):

"9. Population stratification test

We applied LD Score intercept method⁸ to assess whether population stratification influenced our findings or potentially resulted in false positives. In doing so, we used the LDSC software⁹ to calculate LD Score regressions for each occupational score separately. LD Scores were computed utilizing genotypic data from individuals of European ancestry within the 1000 Genomes Project, specifically focusing on HapMap3 SNPs. Inclusion in the LD Score regression analysis was limited to HapMap3 SNPs with a minor allele frequency (MAF) exceeding 0.01 only.

Supplementary Table 2 below demonstrates the results of our analyses. It is noticeable that LD Score intercepts deviate statistically significantly from 1, although the deviation is not substantial. The χ^2 statistics fall within the range of 1.53 to 1.74 for each occupational score. Consequently, we find evidence supporting the notion that some of the identified SNPs are linked to our phenotypes, and approximately 5.2% to 7.4% of the observed inflation in χ^2 can be attributed to potential biases arising from factors such as population stratification and other confounders. Nevertheless, this influence appears limited, and our overall findings indicate evidence that the predominant portion of the observed signal is attributable to polygenic factors.

Outcome Mean χ^2 Intercept (SE) Inflation

CAMSIS	1.738	1.1193	(0.0142)	0.0740622
ISEI	1.568	1.0993	(0.0134)	0.0616461
SIOPS	1.534	1.0845	(0.0123)	0.0524581

Supplementary Table 2. χ^2 statistics for occupational status scores"

Reviewer #3

Supplementary table 2 is a great inclusion. However, I'm confused as to how inflation was derived. Using LDSC it is typically $(\text{intercept} - 1) / (\text{mean}(\chi^2) - 1)$ and using this equation I derive inflation of 15-17%.

In addition, apologies for my lack of clarity. The title to which I was referring to the title of reference 99

Yang, J., Jiang, L. & Zheng, Z. FastGWA-GLMM: a generalized linear mixed model association tool for biobank-scale data. (2021).

I believe the correct reference is

Jiang, L., Zheng, Z., Fang, H. et al. A generalized linear mixed model association tool for biobank-scale data. Nat Genet 53, 1616–1621 (2021).

Reviewer #3

In the discussion it is stated "Certain genetic traits may predispose individuals to

achieve higher levels in these areas through biological pleiotropy (Hill et al. (2019)).

I think you mean vertical pleiotropy here. Hill et al. (2019) write "more biologically proximal/distal phenotypes" but the schematic in Figure 1 of Hill et al. (2019) clearly describes vertical (mediated) pleiotropy.

Answer:

We thank the reviewer for this excellent suggestion, and we corrected it accordingly. We, indeed, rather referred here to the instances of vertical pleiotropy. The sentence is changed to the following (Manuscript | P.13): "Certain genetic traits may predispose individuals to achieve higher levels in particular areas through a mechanism known as vertical pleiotropy (i.e., mediated pleiotropy).23"

Reviewer #3

Perfect.

Reviewer #4 (Remarks to the Author):

Congratulations to the authors for addressing our comments in a thorough and exhaustive way. This is truly professional work. I have no further comments.

Reviewer #5 (Remarks to the Author):

Thank you for the opportunity to review the manuscript Polygenic prediction of occupational status GWAS elucidate genetic and environmental interplay for intergenerational transmission, careers, and health. This paper describes the results of a GWAS on sociologically-informed occupational status measures using the UKBiobank My understanding is that the study aims to examine the complex interplay between biological inheritance and social processes. I have specifically been asked to provide an ethical review of this manuscript and so my review is focused on that.

In general

I appreciated the Box 1 Ethical Considerations. I reminded me of the 2023 Mignogna et al Nature Human Behavior paper's Box 1. I actually went back to read the Mignogna Box 1 to compare, and it raised some questions for me about this study that should perhaps be covered in Box 1 or in their Methods (I know the authors briefly touch on some of what I am about to ask in their supplementary materials). Did the authors of this study seek specific approval from the datasets they used to conduct this study? Why or why not? What sort of IRB or ethics review process was there for this study? The supplemental materials state: "This research was conducted using the UK Biobank resource under application 32696 396 and NCDS under application GDAC_2021_16_TROPF, with ethical approval from the University of Oxford." It is not clear from that statement whether the UK Biobank application or NCDS application were specific to the scope of this study.

I also appreciate the authors putting together an FAQ. These are time and labor intensive and not always appreciated by colleagues. Have the authors considered submitting their FAQ to the FoGS repository? <https://www.thehastingscenter.org/genomic-findings-on-social-and-behavioral-outcomes-faqs/> (to learn more: <https://doi.org/10.1038/s41588-021-00929-5>)

I was also hoping for more of a reflection on the ethical considerations of studying occupational status specifically. Why is studying the genetics of occupational status so sensitive? One reason is that it challenges intuitions about free-will and human agency. Another is that it might lead to self-fatalism, policy fatalism, stigmatization of self or others, etc. The authors reference a 2023 consensus report (note that it is from the Hastings Center not the Hastings Institute) and Figure 3 of that report lists some risks of sociobehavioral genomics research. I felt the points raised in Figure 3 of that Hastings Center report, that is, a discussion of the risks this specific research poses, were missing from both Box 1 and the FAQ. It will be hard to combat against those risks without mentioning them.

I thought these might be some additional helpful cites for the authors as they think about the ethical considerations of their work:

These are older, but still relevant:

Berryessa, C. M., & Cho, M. K. (2013). Ethical, legal, social, and policy implications of behavioral genetics. *Annual Review of Genomics and Human Genetics*, 14, 515–534. <https://doi.org/10.1146/annurev-genom-090711-163743>

Panofsky, A. (2014). *Misbehaving science: Controversy and the development of behavior genetics* (pp. xi, 321). University of Chicago Press. <https://doi.org/10.7208/chicago/9780226058597.001.0001>

Parens, E. (2004). Genetic differences and human identities: On why talking about behavioral genetics is important and difficult. *The Hastings Center Report*, 34(1), S1–S1.

More recent:

M.N. Meyer, T. Gjorgjieva, C.F. Chabris, "Laypeople Overestimate the Predictive Power of Polygenic Scores But Do Not View Them as Any More Anxiety-Producing than Other Scores for the Same Trait," *Behavior Genetics* 52, no. 6 (2022): 378. In <https://dornsife.usc.edu/cesr/wp-content/uploads/sites/54/2023/11/Finch-et-al-2022.pdf>

De Hemptinne, M. C., & Posthuma, D. (2023). Addressing the ethical and societal challenges posed by genome-wide association studies of behavioral and brain-related traits. *Nature Neuroscience*, 26(6), 932–941. <https://doi.org/10.1038/s41593-023-01333-4>

Box 1.

Box 1 says: "It is important to recognize that these past studies have contributed to an aversion, anger and even fear of studying genetics in these areas of research." Could the authors be more direct and name 'the areas of research'?

I think it is important that Box 1 talked about how this study does not attempt “to compare individuals across contentious social divisions in contemporary societies, such as by race or ethnicities...” I want to make sure, however, that when the authors later state, “Although extending our study to other ancestral groups is beyond the scope of our current discovery study, we continue to encourage further research in this area”, that it is not interpreted as the authors endorsing group comparisons research (unless they are?). It might be helpful to say why you encourage further research in this area. I also am not sure why this sentence does not come directly after the sentence “we note that a high concentration of research...”. It seems that the two sentences preceding “Although extending our study...” are unrelated to this sentence but that the sentence “we note that a high concentration of research...” is.

Related to the above: “Our investigation neither attempts to compare individuals across contentious social divisions in contemporary societies, such as by race or ethnicity, nor does it involve comparing genetic ancestral groups that could be conflated with racial or ethnic categories.” This is a response to the Hastings Center report’s categorization of SBG research of “greatest concern” not “heightened concern.” I think the authors know this distinction but to avoid confusing readers it may make sense to first explicitly state that the Hastings Center report defined SBG research of “greatest concern” as “research on sensitive phenotypes that compares two or more groups defined by race, ethnicity, or genetic ancestry, where—due to similarities in how races, ethnicities, and genetic ancestral populations are typically identified—genetic ancestry could easily be misunderstood as race or ethnicity” (this is a direct quote from the report). This way it is clearer that the authors are not mistakenly presenting group-comparisons research as SBG research of “heightened concern.”

“It is important to recognize that these past studies have contributed to an aversion, anger and even fear of studying genetics in these areas of research.^{29,34}” I wondered what the authors would say about more current studies published by individuals like Noah Carl and Bo Winegard in journals like *Evolutionary Psychological Science*? The way the sentence is framed makes it sound as though studies making group comparisons or concluding that there is an immutable biological basis for societal stratification are relics of the past. Part of what contributes to the “aversion, anger, and even fear of studying” the genetics of human behavior and social outcomes are the continued ways in which, as Alondra Nelson phrases it: “bigotry claims science as its ally” (<http://www.nature.com/articles/d41586-020-02546-4>). The Buffalo Shooting is one example of what Nelson is talking about. I think it important that the authors recognize the present moment and not just the past.

I find it problematic to frame one side of the Hastings Center project as “critical voices” and the other as “researchers.” Doing so makes it sound like the “critical voices” were not also researchers. It also frames the ‘critique’ as only coming from one side.

The authors write “our work already adhered to the Hastings Institute Report design guidelines” – can they specify which design guidelines they followed briefly here? They go into some of those guidelines in later in the Box. But, for example, one of the Hastings Center guidelines for responsible conduct is “engaging with stakeholders,” which I do not believe the research team did. I’m not saying they should have, but the way it is written makes it sound as if the authors followed all the Hastings Center recommendations rather than some. As another example, the Hastings Center report recommends “justifying the use and definition of ‘populations.’” This paper uses the term “British-European ancestry” but does not justify the use or the definition of the term. Doing so would also be in accordance with the NASEM report on population descriptors: <https://www.nap.edu/catalog/26902> I noticed that the NASEM report was not referenced by the authors, so perhaps they have not seen it. Furthermore, the Hastings Center Report had guidelines for both responsible conduct and responsible communication. The authors primarily seemed to have followed a number of the “responsible conduct” recommendations, but there are responsible communication recommendations from that report that could be followed in addition to the FAQ the authors already put together:

1. Developing a “key-points” box that includes how results should(n’t) be (mis)interpreted or (mis)used
2. Diverting misinterpretations or misuse via FAQs (which the authors have done), videos, and careful press release
3. Reporting effect sizes in the abstract and avoiding exaggerating them, including in graphs
4. Embedding caveats and context in graphs and tables
5. Defining and justifying the use of “populations”
6. Moving away from population language that is easily conflated with race or ethnicity

I think in particular, developing a “key-points” box that includes how results should(n’t) be interpreted or used could enhance the accessibility of the FAQ. I also think that briefly describing how results should(n’t) be interpreted or used in Box 1 would be useful. I also think the authors could report effect sizes in the abstract.

“Our GWAS focuses solely on a population of British-European ancestry, which means that results can only be generalized to that population.” There is evidence that participation bias has impacted the UKBB (e.g., Schoeler, T., Speed, D., Porcu, E., Pirastu, N., Pingault, J.-B., and Kutalik, Z. (2023). Participation bias in the UK Biobank distorts genetic associations and downstream analyses. *Nat. Hum. Behav.* 7, 1216–1227. [10.1038/s41562-023-01579-9](https://doi.org/10.1038/s41562-023-01579-9)). As a result, the UKBB database has several caveats, as participation bias creates the possibility that any genetic trends identified are not related to the trait but to the subset of individuals who enroll in the database. It therefore seems that there is some nuance required before being able to state that the results can be generalized to British-European ancestries. The authors do describe the limitations of UK BB later in the manuscript on page 16 (though they do not necessarily go so far as to state the possibility that any genetic trends identified are not related to the trait but to the subset of individuals who enroll in the database). However, given these limitations how appropriate is it to say the findings can be generalized to a population of British-European ancestry in Box 1 when readers have not been given the appropriate nuance?

FAQ.

I appreciated that the FAQ explicitly stated that it was intended for broader audience who are “new to the scientific terminology and methods.” I therefore read the FAQ with this in mind and have several suggestions as I felt it often was not written in language that would be accessible to a broader audience.

First, the FAQ document could do with some re-organization. For instance, it introduces concepts like GWAS and DNA variants 4 pages in when the terms have already been used. It would make more sense to define key concepts first.

Second, there are moments when the FAQ reads in more accessible language but then within the same sentence or set of sentence switches to more technical language. It is unclear to me how the authors decide which terms to define more accessibly and which to leave as-is. This is another reason why I think some definitions of key terms at the very start would be helpful (e.g., GWAS, polygenic score, SNP, genetic association, occupational status).

Just as an example, on page 8 the authors discuss “individual genetic scores” called “polygenic scores.” I see this as a way to help make the term ‘polygenic score’ more digestible for someone who may not be familiar with the scientific terminology. But then the authors go on to say “which we used to control for genetic associations” and I am left wondering why they do not explain what a ‘genetic association’ is. I also wonder whether referring to a ‘polygenic score’ as an ‘individual genetic score’ might lead people to think this information is predictive at the level of the individual rather than as a population-level tool.

Third, I would urge the authors to take a look at the “Responsible Research Lifecycle” in a 2023 Gordon et al paper on the genetics of musicality: <https://onlinelibrary.wiley.com/doi/abs/10.1111/nyas.14972> Part of reflecting on the ethical considerations of this study is not just to list those considerations but to spend some time thinking about how to respond to them. Box 1 provides a brief description of a fraught history, it also states what the study is not (i.e., not doing group comparisons), and then it tells people to go look at the FAQ. I understand that there are constraints on what can go in Box 1. In the spirit of the “responsible research lifecycle,” the FAQ could be a place to explicitly challenge determinist interpretations and address potential for misuse while providing guidelines for appropriate use (e.g., an FAQ question like: “did you find the genetic basis for occupational status” and “are people biologically predetermined to an occupational status” or to take a question from the Mignogna FAQ: “Should public officials, policy makers, insurers, or health care professionals use the results of this study to make decisions?”)

The FAQ says: “For socioeconomic status measures such as educational attainment, nearly 4,000 genetic variants have been associated with the outcomes in previous studies. Genetic variants associated with income have also been established. Previous research has also shown that the third measure of occupational status, is not only driven by socio-environmental factors, but it also potentially has a biological and genetic basis. Twin studies suggest a heritability of occupational status as between 0.30 to 0.40. The current study goes substantially beyond what we know about the genetics and biological factors associated with occupational status.” I was struck by how little was discussed regarding what the socio-environmental factors known to impact occupational status. Earlier the FAQ says: “The majority of social stratification research has focused on the social determinants and social aspects of intergenerational transmission (i.e., from parent to offspring) of SES, often neglecting any role of biology or genetics,” but again does not share what that literature has revealed. Given the fact that the study is presented as examining a complex interplay between genes and environments, I worried that not enough attention is being given to the environments side when describing the prior literature.

I am struck by what I think is a very important point in the abstract of the manuscript that did not seem to be in the FAQ: 62% of the intergenerational transmission of occupational status can be ascribed to non-genetic inheritance such as family environment and possibly the effects of rare genetic variants.” There are two very important points that are highlighted in the FAQ that are related: “The intergenerational transmission of occupational status only partly explain children’s occupational status. Occupational status is only partly associated with genetic variants, and partly with the family background; the rest is neither family nor genes but unique circumstances and factors that are still not measured. “ “Social environmental factors, such as the family that someone grows up in, has a strong impact via what is called gene-environment correlations... gene-environment correlations likely account for approximately 25% of the discovered estimates.” However, I could not find mention of this 62% anywhere in the FAQ and wondered why. It looks like it may be broken down into different components.

Version 3:

Decision Letter:

Dear Dr Akimova,

We are pleased to inform you that your Article "Polygenic prediction of occupational status GWAS elucidates genetic & environmental interplay in Intergenerational Transmission, Careers & Health in UK Biobank", has now been accepted for publication in *Nature Human Behaviour*.

Please note that *Nature Human Behaviour* is a Transformative Journal (TJ). Authors may publish their research with us through the traditional subscription access route or make their paper immediately open access through payment of an article-processing charge (APC). Authors will not be required to make a final decision about access to their article until it has been accepted. [Find out more about Transformative Journals](https://www.springernature.com/gp/open-research/transformative-journals)

Authors may need to take specific actions to achieve [compliance](https://www.springernature.com/gp/open-research/funding/policy-compliance-faq) with funder and institutional open access mandates. If your research is supported by a funder that requires immediate open access (e.g. according to [Plan S principles](https://www.springernature.com/gp/open-research/plan-s-compliance)) then you should select the gold OA route, and we will direct you to the compliant route where possible. For authors selecting the subscription publication route, the journal’s standard licensing terms will need to be accepted, including [self-archiving policies](https://www.springernature.com/gp/open-research/policies/journal-policies). Those licensing terms will supersede any other terms that the author or any third party may assert apply to any version of the manuscript.

With best regards,

[REDACTED]

P.S. Click on the following link if you would like to recommend Nature Human Behaviour to your librarian
<http://www.nature.com/subscriptions/recommend.html#forms>

** Visit the Springer Nature Editorial and Publishing website at http://editorial-jobs.springernature.com?utm_source=ejP_NHumB_email&utm_medium=ejP_NHumB_email&utm_campaign=ejp_NHumB for more information about our career opportunities. If you have any questions please click [here](mailto:editorial.publishing.jobs@springernature.com). **

Reviewer #1 (Remarks to the Author):

Thanks for addressing all my comments! The manuscript has been much improved after revision. This will be a significant contribution to the literature.

Reviewer #2 (Remarks to the Author):

The authors have addressed my comments well. They reframed their paper as an extension of Ko et al. and calculated phenotypic and genetic correlations with the measures used by Ko et al. The genetic correlations are > 0.95 , which might make readers wonder how the authors' SNP heritability can be twice as large as Ko et al.'s estimate when the measures are nearly identical genetically. The authors might forestall this concern by noting that genetic correlations are independent of heritability – that is, one measure could be highly heritable and the other measure modestly heritable, but their genetic correlation could be 1.0.

The new analyses of assortative mating using spousal data from UK Biobank confirm their hypothesis that assortative mating accounts in part for the reduction in effects from between families to within families. Spousal phenotypic correlations are from 0.22 to 0.33 and spousal polygenic score correlations are about 0.10.

The authors have also helpfully unpacked their intriguing finding about the compression of genetic prediction in mid-career.

Answer: We are grateful for the thoughtful feedback and suggestions and glad that we addressed them all. Thank you also for a helpful note with regards to genetic correlations and heritability estimates of the measures we used in this study and the ones previously used by Ko et al. We now provide an additional note that states:

We observed strong correlations between all three occupational prestige and status measures. Supplementary Figure 3 below illustrates an extremely high genetic correlation between CAMSIS, ISEI, and SIOPS (lower triangle), much stronger than implied by their phenotypic correlations (upper triangle). We also observe a substantial genetic correlation with the SOC2000 index used by Ko et. al.36, while phenotypic correlations are of a lesser magnitude.[1]

[1] We would like to point out that although our genetic correlations with the measures previously used by Ko et al. are high, our SNP heritability estimates

are almost twice as high, especially for CAMSIS. This similarity in genetic correlations, despite the substantial difference in LDSC-based SNP heritability, is somewhat expected. These are independent measures, and high genetic correlations can occur between outcomes independent of heritability. Moreover, we also expect variation in validity among different occupational scores (i.e., different measures capture the intended concepts to varying extents), which likely plays a role in the observed high genetic correlations and substantial differences in SNP heritabilities.

(Supplementary Material | P.25)

Reviewer #3 (Remarks to the Author):

Reviewer #3

Remarks to the Author:

This is an exceptional well thought through and conducted study. It should be regarded as the gold standard as to what questions regarding SES can be addressed using genetic data as well as the best way to address them. My comments are minor but I hope contribute in some way to polishing a fine piece of scientific literature.

Answer:

We want to express our sincere gratitude for such kind words! Your acknowledgment of our work is nothing but truly inspiring. We also appreciate your comments, which, though minor, undoubtedly contributed to refining our manuscript. We can confirm that we were able to perform additional tests to address your concerns and further clarified the methods and statistical tools we used.

Reviewer #3

In the abstract and throughout the manuscript 106 genetic variants are described as being identified. Either loci or independent SNPs would be preferable here as there are far more than 106 associated variants. Furthermore, it is stated that eight of these variants has not been previously associated with differences in SES. Could the authors clarify if these eight loci harbour variants that have previously been associated with differences in SES?

Answer:

We thank the reviewer for this thoughtful comment and have now made the necessary amendments to address these concerns. To provide clarity and precision, we have updated the terminology in the main text and the appendix, replacing “genetic variants” with “independent

SNPs”.

With respect to the eight SNPs not previously associated with SES differences, we arrived at this number after conducting a comprehensive assessment. Utilizing FUMA, we identified all 1000 Genomes SNPs in linkage disequilibrium (with $R^2 > 0.6$ for European ancestry) related to the 106 independent SNPs. This yielded a total of 11,206 SNPs. Subsequent analysis, referencing the GWAS Catalog and the IEU OpenGWAS project, revealed 1,005,470 phenotypic associations that reached at least suggestive significance ($p < 5 \times 10^{-6}$). We then excluded any variants having a genome-wide significant association with traits related to education, income, or other socioeconomic outcomes. This filtering process culminated in the identification of 8 unique hits (rs12137794, rs17498867, rs10172968, rs7670291, rs26955, rs2279686, rs72744938, rs62058104). Overall, none of these eight SNPs is in LD (at $R^2 > 0.6$) with a SNP that is present in 1KG and has been noted in the GWAS Catalog or IEU OpenGWAS to harbour associations with income, years of schooling, qualifications or similar socioeconomic status related traits at $p < 5 \times 10^{-6}$.

These findings are in the Table B8 of the Supplementary Tables file (Supplementary Tables | B8. PheWAS of Top Hits).

Reviewer #3

Perfect.

Reviewer #3

Additionally, it is stated that FUMA was used to identify these 106 SNPs. However, in the results it is stated that the R^2 threshold was 0.1. The documentation of FUMA is poor here but FUMA uses two rounds of clumping. The first is conducted in exactly the same manner as PLINK and is used to define independent significant SNPs as well as the border of the loci that they are in. A second round of clumping (by default at a lower R^2) is then applied on these independent significant SNPs to identify what FUMA calls lead SNPs.

My question is was the second R^2 equal to the first? If so the independent significant SNPs are the same as the lead SNPs. Also why does the R^2 change between defining independent SNPs/loci and examining if these same SNPs are associated with other traits (In the PHEwas the $R^2 = 0.6$)?

Answer:

We thank the reviewer for raising this point and for providing us with the opportunity to clarify the methods we used. To eliminate any confusion regarding the FUMA method and its application, we have now updated Section 6.3 of the Supplementary Information (Supplementary Information| P.18):

“7.3 Findings

After inflating the standard errors by the square root of their respective intercepts from LD Score regressions, our GWASs identified 106 independent SNPs for CAMSIS including 56 found also for ISEI and 51 for SIOPS based on clumping all genome-wide significant SNPs using an R^2 -threshold of 0.1 and a window-size of 1000kb, one of which (only significant for CAMSIS) was found on the X-chromosome (see Figure 2 in the main text for the Manhattan plot of the autosome). In an exploratory sex-specific GWAS, no separate hits emerged, with genetic correlations being very close to and not significantly different from one.”

As the reviewer can see now, FUMA was not employed to identify the independent SNPs. Instead, for this task, we utilized the `--clump` command in PLINK. We set an R^2 -threshold of 0.1 and a window-size of 1000kb, targeting all genome-wide significant hits ($p < 5 \times 10^{-6}$). We then turned to FUMA, but solely with the objective of identifying all SNPs present in the 1000 Genomes dataset that are in linkage disequilibrium (LD) with our previously identified 106 independent SNPs, as detailed in our response to the comment above.

Reviewer #3

Comparing the documentation of FUMA to the documentation of PLINK it seems that what the authors have done will result in a greater number of loci. Using PLINK clumping the distance between two independent loci will be the distance between the two genome-wide significant SNPs that define those loci. However, using FUMA the distance between two loci will be the distance between the closest SNPs that are in LD with the genome-wide significant SNPs that define those loci. So, what clumping would call two separate loci FUMA would call a single locus. Have the authors compared the difference between these two methods? If so how did they select clumping by PLINK rather than by FUMA?

Answer: Thank you for your detailed feedback, which motivated us to investigate the differences between FUMA and PLINK’s `--clump` command. We would like to note that we are using PLINK’s `--clump` command given this is standard practice in GWAS procedures. PLINK’s `--clump` command is the prevailing software for this task in existing GWAS, and we adopted specifications that should not lead to overestimations of the number of independent SNPs.

On the other hand, FUMA is designed for the functional annotation of GWAS results. We used it to identify which SNPs we discovered have not yet been linked to any socioeconomic status-related traits, alongside the MAGMA analysis we conducted.

There is very limited literature and evidence comparing these two methods and preference of one over the other. In FUMA’s documentation, it is not clearly stated

whether the FUMA algorithm systematically provides a larger or smaller number of independent SNPs compared to PLINK. Our understanding is that PLINK builds clumps of SNPs that are not only a certain distance from the index SNP but also in LD with the index SNP, so there should not be a significant difference between clumping with PLINK or FUMA.

For further transparency, we have added the following footnote in our Supplementary Material to ensure readers are aware of the procedure we are following:

After inflating the standard errors by the square root of their respective intercepts from LD Score regressions, our GWASs identified 106 independent SNPs for CAMSIS including 56 found also for ISEI and 51 for SIOPS based on clumping all genome-wide significant SNPs using a threshold of 0.1 and a window-size of 1000kb, one of which (only significant for CAMSIS) was found on the X-chromosome (see Figure 2 in the main text for the Manhattan plot of the autosome).^[1]

^[1] We are using the PLINK --clump command with the specifications we described above.

(Supplementary Material | P.21)

Reviewer #3

The use of MTAG seems somewhat questionable here. I do see the increase in the number of genome-wide significant loci but I don't understand how this has occurred. Specifically, the authors have near (genetically) identical traits (the rG between them is >0.8) and the GWAS are performed on the same samples (If the education data set used is from Lee et al. there would be new participants, but it seems the data used are all from UKBB). Could the authors provide the mean χ^2 and effective N_s pre and post MTAG? It would help confirm if MTAG is of benefit.

Answer:

These are important points, and we would like to clarify our approach. In the case of the NCDS, we utilized the EA3 GWAS dataset, excluding data from 23andMe and NCDS itself. When conducting validation within the UKB dataset, we exclusively employed the UKB GWAS dataset and ensured the exclusion of siblings or adoptees for specific analyses. The observed improvement in our results can be attributed to variations in sample size and heritability. The sample sizes for income, and particularly for educational attainment (EA), are substantially larger. Regarding heritability, EA exhibited the highest values, followed by CAMSIS. This hierarchical pattern of heritability is also reflected in the relative increases in effective sample

size, which are largest for SIOPS/ISEI and smallest for EA, as displayed in the following table below.

Response Table 1. MTAG: χ^2 and sample sizes

Trait	GWAS_mean_chi2	MTAG_mean_chi2	GWAS_N	MTAG_eff_N
CAMSIS	1.738	2.508	273157	697176
ISEI	1.568	2.418	271769	817666
SIOPS	1.534	2.435	271769	863640
Income	1.639	1.972	353673	721973
Years of Education (EA3)	2.074	2.584	763541	886012

Reviewer #3

I understand the authors point but my question is if there is any benefit from having CAMSIS, ISEI & SIOPS included in the same MTAG analysis. These data sets overlap considerably and the magnitude of the genetic correlation between them ($rg > 0.95$) would indicate that there is very little additional information added.

Answer: Thank you for your question and we hope we clarify our approach in our response here. While we acknowledge that these measures overlap considerably and exhibit high genetic correlations ($rg > 0.95$), our approach here is also motivated by theoretical and substantive considerations, given that these are the most widely used status measures. Importantly, although correlated, each of these occupational measures was developed to capture slightly different dimensions of occupational status and it is thus an empirical question as well to test whether they do. As we describe on p.6 in our main text, ISEI is a status measure that aims to maximize the (indirect) influence of education on income through occupation, SIOPS is a prestige-based measure based on public opinion ranking occupations by their relative social standing, and CAMSIS measures the distance between occupations based on the frequency of social interactions between them. Accordingly, including all three measures in MTAG allows us to take into account a broader spectrum of occupational characteristics and ensures that we do not overlook subtle yet potentially meaningful nuances presented in each measure. This is important as some occupations may score differently across these measures, reflecting distinct

aspects of occupations. We have also included the following footnote to clarify our motivation of using MTAG:

The use of all occupational status scores GWAS for MTAG also has a theoretical and substantive motivation. While ISEI, SIOPS, and CAMSIS have genetic correlations > 0.95 (Supplementary Figure 3), each of these occupational measures captures slightly different dimensions of occupational status. Accordingly, we use MTAG to take into account a broader spectrum of occupational characteristics and to ensure that we do not overlook subtle, yet potentially meaningful nuances within each measure.

(Supplementary Material | P.24)

Reviewer #3

Also, regarding the mean χ^2 . I do see that the authors have used fast-GWAGLMM (again minor but I believe the title of the paper is incorrect) to account for relatedness and reported the LDSC intercepts which all seem close enough to 1. However, by including the mean χ^2 it would be possible to observe the degree to which non-polygenic effects have contributed to the signal.

Answer: We apologize for any confusion and would like to clarify our methodology. We indeed employed fastGWA for our analyses; however, since all our phenotypes are continuous, we did not use fastGWA-GLMM, which is an extension tailored for binary phenotypes.

Regarding the title, if you are referring to our original title - 'Genome-wide association study of occupational status and prestige identifies 106 genetic variants and defines their role for intergenerational status transmission and the life course,' we have made revisions. The title is now 'Polygenic prediction of occupational status GWAS elucidate genetic and environmental interplay for intergenerational transmission, careers, and health.' If your concern revolves around the term 'polygenic,' we are open to further adjustments for even greater clarity. In response to the concern raised about the mean χ^2 , we agree with the assessment. Below, we provide the values for the mean χ^2 . The observed inflation, which ranges between 5% and 7.5%, suggests that potential confounders such as population stratification have a limited impact. This finding further supports our interpretation that the predominant portion of the observed signal is polygenic. We accordingly added a new section to our supplementary materials (Supplementary Information| P.20):

"9. Population stratification test

We applied the LD Score intercept method⁸ to assess whether population stratification influenced our findings or potentially resulted in false positives. In doing so, we used the LDSC software⁹ to calculate LD Score regressions for each occupational score separately. LD Scores were computed utilizing genotypic data from individuals of European ancestry within the 1000

Genomes Project, specifically focusing on HapMap3 SNPs. Inclusion in the LD Score regression analysis was limited to HapMap3 SNPs with a minor allele frequency (MAF) exceeding 0.01 only.

Supplementary Table 2 below demonstrates the results of our analyses. It is notable that LD Score intercepts deviate statistically significantly from 1, although the deviation is not substantial. The chi2 statistics fall within the range of 1.53 to 1.74 for each occupational score. We thus find evidence supporting the notion that some of the identified SNPs are linked to our phenotypes, and approximately 5.2% to 7.4% of the observed inflation in chi2 can be attributed to potential biases arising from factors such as population stratification and other confounders. Nevertheless, this influence appears limited, and our overall findings indicate evidence that the predominant portion of the observed signal is attributable to polygenic factors.

Outcome	Mean chi2	Intercept (SE)	Inflation
CAMSIS	1.738	1.1193 (0.0142)	0.0740622
ISEI	1.568	1.0993 (0.0134)	0.0616461
SIOPS	1.534	1.0845 (0.0123)	0.0524581

Supplementary Table 2. Chi2 statistics for occupational status scores”

Reviewer #3

Supplementary table 2 is a great inclusion. However, I'm confused as to how inflation was derived. Using LDSC it is typically $(\text{intercept}-1)/(\text{mean}(\text{chi}^2)-1)$ and using this equation I derive inflation of 15-17%.

In addition, apologies for my lack of clarity. The title to which I was referring to the title of reference 99

Yang, J., Jiang, L. & Zheng, Z. FastGWA-GLMM: a generalized linear mixed model association tool for biobank-scale data. (2021).

I believe the correct reference is

Jiang, L., Zheng, Z., Fang, H. et al. A generalized linear mixed model association tool for biobank-scale data. Nat Genet 53, 1616–1621 (2021).

Answer: Thank you for providing the correct information about the reference which we have updated. Thank you also for pointing out this oversight - we have now updated both Supplementary Table 2 and the text accordingly:

Supplementary Table 2 below demonstrates the results of our analyses. It is noticeable that LD Score intercepts deviate statistically significantly from 1, although the deviation is not substantial. The χ^2 statistics fall within the range of 1.53 to 1.74 for each occupational score. We thus find evidence supporting the notion that a portion of the identified SNPs are linked to our phenotypes, and approximately 15.8% to 17.5% of the observed inflation in χ^2 can be attributed to potential biases arising from factors such as population stratification and other confounders. Hence, our overall findings indicate evidence that the predominant portion of the observed signal is attributable to polygenic factors but the biases due to stratification or relatedness explain on average 16.5% of the inflation of χ^2 .

Outcome	Mean χ^2	Intercept (SE)	Inflation
CAMSIS	1.738	1.1193 (0.0142)	0.1616531
ISEI	1.568	1.0993 (0.0134)	0.1748239
SIOPS	1.534	1.0845 (0.0123)	0.1582397

Supplementary Table 2. χ^2 statistics for occupational status scores

(Supplementary Material | P.23)

We have also updated the reference accordingly.

Reviewer #3

In the discussion it is stated “Certain genetic traits may predispose individuals to achieve higher levels in these areas through biological pleiotropy (Hill et al. (2019)”. I think you mean vertical pleiotropy here. Hill et al. (2019) write “more biologically proximal/distal phenotypes” but the schematic in Figure 1 of Hill et al. (2019) clearly describes vertical (mediated) pleiotropy.

Answer:

We thank the reviewer for this excellent suggestion, and we corrected it accordingly. We indeed referred here to the instances of vertical pleiotropy. The sentence is changed to the following (Manuscript | P.13):

“Certain genetic traits may predispose individuals to achieve higher levels in particular areas through a mechanism known as vertical pleiotropy (i.e., mediated pleiotropy).²³”

Reviewer #3

Perfect.

Reviewer #4 (Remarks to the Author):

Congratulations to the authors for addressing our comments in a thorough and exhaustive way. This is truly professional work. I have no further comments.

Reviewer #5 (Remarks to the Author):

Thank you for the opportunity to review the manuscript Polygenic prediction of occupational status GWAS elucidate genetic and environmental interplay for intergenerational transmission, careers, and health. This paper describes the results of a GWAS on sociologically-informed occupational status measures using the UKBiobank. My understanding is that the study aims to examine the complex interplay between biological inheritance and social processes. I have specifically been asked to provide an ethical review of this manuscript and so my review is focused on that.

Answer: We are grateful for the thoughtful and constructive feedback. We have thoroughly revised our manuscript and the FAQs accordingly — adding the additional information that was required, clarifying the use of terms, and extensively revising Box 1. We are confident that we were able to address all of the points and thank the reviewers for ensuring the paper has become even stronger with the granular and transparent revisions to Box 1 and the FAQs. Below, we provide our detailed response.

In general I appreciated the Box 1 Ethical Considerations. It reminded me of the 2023 Mignogna et al Nature Human Behavior paper's Box 1. I actually went back to read the Mignogna Box 1 to compare, and it raised some questions for me about this study that should perhaps be covered in Box 1 or in their Methods (I know the authors briefly touch on some of what I am about to ask in their supplementary materials). Did the authors of this study seek specific approval from the datasets they used to conduct this study? Why or why not? What sort of IRB or ethics review process was there for this study? The supplemental materials state: "This research was conducted using the UK Biobank resource under application 32696 396 and NCDS under application GDAC_2021_16_TROPF, with ethical approval from the University of Oxford." It is

not clear from that statement whether the UK Biobank application or NCDS application were specific to the scope of this study.

Answer: We thank you for pointing us to the study by Mignogna et al. (2023) which has been an important and extremely helpful resource. We agree we should have specified in the earlier versions that our data applications were specific to the scope of our study and included a dedicated ethics section within our manuscript. We have now updated the FAQ accordingly and added the 'Ethics approval' section in our Methods:

Ethics approval

This research was conducted using the UK Biobank under application 32696 and NCDS under application GDAC_2021_16_TROPF, with ethical approval from the University of Oxford under application SOC_R2_001_C1A_21_60. Both the UK Biobank and NCDS applications were specific to the scope of this paper. For the UK Biobank approval, we received approval for a scope extension to ensure transparency, allowing us to expand from our focus on non-standard occupations to also include occupational status. Here we specified that our plan was: "to perform GWAS analysis using employment histories from the UK Biobank to construct sociologically informed measures of occupational status." We specified that we would construct sociologically informed measures of occupational status (CAMSIS, SIOPS, and ISEA) for our GWAS and noted that the analysis would be accompanied by NCDS genetic and phenotypic data. For the NCDS application, we specified not only the information mentioned above but also the set of polygenic prediction analyses. We also preregistered our analysis plan (<https://osf.io/djbr2/>) which was updated for replication (<https://osf.io/x6va5>).

(Manuscript | P.17-18) (Supplementary Material | P.13)

I also appreciate the authors putting together an FAQ. These are time and labor intensive and not always appreciated by colleagues. Have the authors considered submitting their FAQ to the FoGS repository? <https://www.thehastingscenter.org/genomic-findings-on-social-and-behavioral-outcomes-faqs/> (to learn more: <https://doi.org/10.1038/s41588-021-00929-5>)

Answer: Thank you very much for the excellent suggestion. We will contact the FoGS repository if and once the paper is accepted for publication. In anticipation of this, we have also developed a "key messages" section, which we have now included in our FAQ and will provide for the FoGS submission, as it is one of the main platform's requirements.

I was also hoping for more of a reflection on the ethical considerations of studying occupational status specifically. Why is studying the genetics of occupational status so sensitive? One reason is that it challenges intuitions about free-will and human agency. Another is that it might lead to self-fatalism, policy fatalism, stigmatization of self or others, etc. The authors reference a 2023 consensus report (note that it is from the Hastings Center not the Hastings Institute) and Figure 3 of that report lists some risks of sociobehavioral genomics research. I felt the points raised in Figure 3 of that Hastings Center report, that is, a discussion of the risks this specific research poses, were missing from both Box 1 and the FAQ. It will be hard to combat against those risks without mentioning them.

Answer. These are excellent points and we have now added in these very important points from Figure 3 in the Hasting Center Report to both Box 1 and throughout our FAQs (especially the section ‘What are the potential risks of studying occupational status and genetics?’).

I thought these might be some additional helpful cites for the authors as they think about the ethical considerations of their work:

These are older, but still relevant:

Berryessa, C. M., & Cho, M. K. (2013). Ethical, legal, social, and policy implications of behavioral genetics. *Annual Review of Genomics and Human Genetics*, 14, 515–534.

<https://doi.org/10.1146/annurev-genom-090711-163743>

Panofsky, A. (2014). *Misbehaving science: Controversy and the development of behavior genetics* (pp. xi, 321). University of Chicago Press.

<https://doi.org/10.7208/chicago/9780226058597.001.0001>

Parens, E. (2004). Genetic differences and human identities: On why talking about behavioral genetics is important and difficult. *The Hastings Center Report*, 34(1), S1–S1.

More recent:

M.N. Meyer, T. Gjorgjieva, C.F. Chabris, “Laypeople Overestimate the Predictive Power of Polygenic Scores But Do Not View Them as Any More Anxiety-Producing than Other Scores for the Same Trait,” *Behavior Genetics* 52, no. 6 (2022): 378. In <https://dornsife.usc.edu/cesr/wp-content/uploads/sites/54/2023/11/Finch-et-al-2022.pdf>

De Hemptinne, M. C., & Posthuma, D. (2023). Addressing the ethical and societal challenges posed by genome-wide association studies of behavioral and brain-related traits. *Nature Neuroscience*, 26(6), 932–941. <https://doi.org/10.1038/s41593-023-01333-4>

Answer: We are grateful for these excellent suggestions and integrated what was possible and relevant into the main body of the text and supplementary material.

Box 1.

Box 1 says: "It is important to recognize that these past studies have contributed to an aversion, anger and even fear of studying genetics in these areas of research." Could the authors be more direct and name 'the areas of research'?

Answer: We clarified our text accordingly and now Box 1 reads as:

It is important to recognize that these studies have contributed to an aversion, anger and even fear of studying genetics in social stratification research.

(Manuscript | P.4)

I think it is important that Box 1 talked about how this study does not attempt "to compare individuals across contentious social divisions in contemporary societies, such as by race or ethnicities..." I want to make sure, however, that when the authors later state, "Although extending our study to other ancestral groups is beyond the scope of our current discovery study, we continue to encourage further research in this area", that it is not interpreted as the authors endorsing group comparisons research (unless they are?). It might be helpful to say why you encourage further research in this area. I also am not sure why this sentence does not come directly after the sentence "we note that a high concentration of research...". It seems that the two sentences preceding "Although extending our study..." are unrelated to this sentence but that the sentence "we note that a high concentration of research..." is.

Answer: Thank you for pointing it out and there was indeed room for confusion, which we have rectified. These are important considerations and we amended our text accordingly, also going beyond only the need for ancestral diversity but also socioeconomic status and geography:

We also neither attempt to nor endorse comparison of individuals across contentious socially-constructed groups, such as by race or ethnicity, nor do we compare genetic ancestral groups that could be conflated with racial or ethnic categories. Our GWAS focuses exclusively on a population of British-European genetic ancestry, limiting the generalizability of the results to that population alone. We note that a high concentration of research in this ancestry group remains a general shortcoming of contemporary genetic research, with 72% of genetic discoveries from just three countries (US, UK, Iceland).^{38,39} We encourage inclusion of diverse populations into genetic research across multiple domains including ancestry, socioeconomic backgrounds, geography, age and beyond. We also note that the dataset we use is affected by participation bias^{40,41} (see discussion, FAQ in Supp Mat). Our encouragement for diversity should not be interpreted as an endorsement of studies aimed at comparing different

*ancestral groups, but rather understanding the unique genetic and environmental interplay within each group rather than drawing comparisons between them, line with NASEM framework.*³⁷

(Manuscript | P.4)

Related to the above: “Our investigation neither attempts to compare individuals across contentious social divisions in contemporary societies, such as by race or ethnicity, nor does it involve comparing genetic ancestral groups that could be conflated with racial or ethnic categories.” This is a response to the Hasting’s Center report’s categorization of SBG research of “greatest concern” not “heightened concern.” I think the authors know this distinction but to avoid confusing readers it may make sense to first explicitly state that the Hastings Center report defined SBG research of “greatest concern” as “research on sensitive phenotypes that compares two or more groups defined by race, ethnicity, or genetic ancestry, where—due to similarities in how races, ethnicities, and genetic ancestral populations are typically identified—genetic ancestry could easily be misunderstood as race or ethnicity” (this is a direct quote from the report). This way it is clearer that the authors are not mistakenly presenting group-comparisons research as SBG research of “heightened concern.”

Answer: We agree that our initial version lacked necessary clarification. Since we extensively revised Box 1, and hope this eliminates any confusion.

“It is important to recognize that these past studies have contributed to an aversion, anger and even fear of studying genetics in these areas of research.^{29,34}” I wondered what the authors would say about more current studies published by individuals like Noah Carl and Bo Winegard in journals like *Evolutionary Psychological Science*? The way the sentence is framed makes it sound as though studies making group comparisons or concluding that there is an immutable biological basis for societal stratification are relics of the past. Part of what contributes to the “aversion, anger, and even fear of studying” the genetics of human behavior and social outcomes are the continued ways in which, as Alondra Nelson phrases it: “bigotry claims science as its ally” (<http://www.nature.com/articles/d41586-020-02546-4>). The Buffalo Shooting is one example of what Nelson is talking about. I think it important that the authors recognize the present moment and not just the past.

Answer: This is an excellent point, and we have now changed the language throughout Box 1 to clarify the tense that this type of research is not only in the past.

I find it problematic to frame one side of the Hastings Center project as “critical voices” and the other as “researchers.” Doing so makes it sound like the “critical voices” were not also researchers. It also frames the ‘critique’ as only coming from one side.

Answer: It was not our intention to suggest that these were mutually exclusive groups. We agree that it might sound misleading and corrected the text accordingly:

In a 2023 consensus report from the Hastings Center,³⁶ a group of bioethicists and researchers in the field of social and behavioral genomics emphasized the risks and need for responsible conduct in studies examining the genetics of social and behavioral phenotypes.

(Manuscript | P.4)

The authors write “our work already adhered to the Hastings Institute Report design guidelines” – can they specify which design guidelines they followed briefly here? They go into some of those guidelines in later in the Box. But, for example, one of the Hastings Center guidelines for responsible conduct is “engaging with stakeholders,” which I do not believe the research team did. I’m not saying they should have, but the way it is written makes it sound as if the authors followed all the Hastings Center recommendations rather than some.

Answer: Although we understand that you also note that this is not a requirement, given that we are examining a large sample of employed persons (in which we can derive occupational status measures), it would be outside of the scope of the study to engage with stakeholders, given that they would be hard to define and manifold. We agree that we should have clarified that we followed most of the guidelines we followed and have done so accordingly in Box 1:

Although our analysis was conducted prior to the Hastings Center Report design guidelines, our work mostly adheres to relevant guidelines. We provide a comprehensive explanation of the definition and measurement of our key phenotypes, used an adequately powered sample, replicated out of sample, used within-family estimates and transparently discuss and even highlight observed reductions in effect sizes (responsible conduct). We also developed an extensive FAQs along with a “key-points” section and are transparent in our use of genetic ancestry term, consulting the NASEM report (responsible communications).³⁷ We also neither attempt to nor endorse comparison of individuals across contentious socially-constructed groups, such as by race or ethnicity, nor do we compare genetic ancestral groups that could be conflated with racial or ethnic categories. Our GWAS focuses exclusively on a population of British-European genetic ancestry, limiting the generalizability of

the results to that population alone. We note that a high concentration of research in this ancestry group remains a general shortcoming of contemporary genetic research, with 72% of genetic discoveries from just three countries (US, UK, Iceland).^{38,39} We encourage inclusion of diverse populations into genetic research across multiple domains including ancestry, socioeconomic backgrounds, geography, age and beyond. We also note that the dataset we use is affected by participation bias^{40,41} (see discussion, FAQ in Supp Mat). Our encouragement for diversity should not be interpreted as an endorsement of studies aimed at comparing different ancestral groups, but rather understanding the unique genetic and environmental interplay within each group rather than drawing comparisons between them, line with NASEM framework.³⁷

(Manuscript | P.4)

As another example, the Hastings Center report recommends “justifying the use and definition of “populations.”” This paper uses the term “British-European ancestry” but does not justify the use or the definition of the term. Doing so would also be in accordance with the NASEM report on population descriptors: <https://www.nap.edu/catalog/26902> I noticed that the NASEM report was not referenced by the authors, so perhaps they have not seen it.

Answer: We would like to thank you once again for pointing us towards another excellent and important resource. We have now closely consulted the NASEM report along with the one from the Hastings Center and have implemented the following changes to address this concern:

1. We have defined the term “genetic ancestry” in our Glossary section for transparency and to decrease the possibility of harmful misuse:

Genetic ancestry. *In the context of GWAS, genetic ancestry refers to the measure of genetic similarity among individuals to eliminate biases that are due to historical human migration. It should not be confused with race or ethnicity; it is also not a direct measure of genealogical ancestry.*

(Supplementary Material | P.8)

2. We are now consistently using “British-European genetic ancestry” as our primary term in the main text, supplementary materials, and FAQ, and do not introduce other terms as synonyms. We have also consulted the existing literature, especially studies based on the UK Biobank, and found this term to be the least potentially harmful compared to other terms used, such as “White British ancestry.”

3. We have further clarified how we determine genetic ancestry and why we are subsetting our sample accordingly in the Supplementary Material:

7.1 Analyses

... Genetic ancestry was determined based on principal components (PC) analysis of the genetic data.

(Supplementary Material | P.20)

7.2 Sample inclusion criteria

... We focus on individuals of British-European genetic ancestry only in order to decrease the risk of confounding due to population stratification.

(Supplementary Material | P.21)

Furthermore, the Hastings Center Report had guidelines for both responsible conduct and responsible communication. The authors primarily seemed to have followed a number of the “responsible conduct” recommendations, but there are responsible communication recommendations from that report that could be followed in addition to the FAQ the authors already put together:

1. Developing a “key-points” box that includes how results should(n’t) be (mis)interpreted or (mis)used
2. Diverting misinterpretations or misuse via FAQs (which the authors have done), videos, and careful press release
3. Reporting effect sizes in the abstract and avoiding exaggerating them, including in graphs
4. Embedding caveats and context in graphs and tables
5. Defining and justifying the use of “populations”
6. Moving away from population language that is easily conflated with race or ethnicity

Answer: Thank you for these suggestions. We have implemented the recommendations for responsible communication according to the Hastings report and provide further details on how we have done so in our responses below.

I think in particular, developing a “key-points” box that includes how results should(n’t) be interpreted or used could enhance the accessibility of the FAQ.

Answer: We previously had the section ‘What are the big take-away findings?’ which was the fifth point of the original FAQs. We have now moved that section to

the top and renamed it “Key points”. We also slightly edited it and added more explanation to ensure it is accessible to a broader audience.

I also think that briefly describing how results should(n’t) be interpreted or used in Box 1 would be useful.

Answer: That is a very good idea and we have an entire section now in the FAQ and have emphasized this in Box 1 in various places as well.

I also think the authors could report effect sizes in the abstract.

Answer: We have now updated our abstract and included the effect sizes.

“Our GWAS focuses solely on a population of British-European ancestry, which means that results can only be generalized to that population.” There is evidence that participation bias has impacted the UKBB (e.g., Schoeler, T., Speed, D., Porcu, E., Pirastu, N., Pingault, J.-B., and Kutalik, Z. (2023). Participation bias in the UK Biobank distorts genetic associations and downstream analyses. *Nat. Hum. Behav.* 7, 1216–1227. 10.1038/s41562-023-01579-9). As a result, the UKBB database has several caveats, as participation bias creates the possibility that any genetic trends identified are not related to the trait but to the subset of individuals who enroll in the database. It therefore seems that there is some nuance required before being able to state that the results can be generalized to British-European ancestries. The authors do describe the limitations of UK BB later in the manuscript on page 16 (though they do not necessarily go so far as to state the possibility that any genetic trends identified are not related to the trait but to the subset of individuals who enroll in the database). However, given these limitations how appropriate is it to say the findings can be generalized to a population of British-European ancestry in Box 1 when readers have not been given the appropriate nuance?

Answer: Thank you for raising this point - it is an important nuance we should have clarified and included in both Box 1 and FAQs. We have done so accordingly by implementing the following changes noting the limitations introduced by participation bias in the UK Biobank (we have also included the references the reviewer mentions).

In the main text:

Our study is also not without limitations. The UK Biobank represents only 5.5% of the approached target population and over-represents individuals with lower

genetic risk of mental health problems, BMI, nonsmoking, higher education, and from less economically deprived areas.^{59,91-94} Consequently, participation bias affects the UK Biobank, limiting its generalizability and introducing the potential that observed genetic associations may be influenced by the characteristics of the subset of individuals who chose to participate in the UK Biobank.⁴⁰ We do show how our measures of occupational status differ from UK census data (SI, Section 6). We can expect environmental heterogeneity across different populations to challenge our findings. We, accordingly, use the NCDS sample, another UK dataset with different potential selection biases, to strengthen our analytic approach.

(Manuscript | P.16-17)

Box 1:

We also note that the dataset we use is affected by participation bias (see discussion, FAQ in Supp Mat).

(Manuscript | P.4)

FAQ:

Second, we draw most of our results from the UK Biobank, which is a selective population that has fewer health problems and a higher SES. Such a participation bias limits the generalizability and introduces the potential that observed genetic associations may be influenced by the characteristics of the subset of individuals who chose to participate in the UK Biobank. A recent Nature Genetics (2023) study by colleagues working in our own Leverhulme Centre for Demographic Science (LCDS) explain why this is an issue for genetic research, also described here. Third, due to data limitations, we recognize that we were unable to also include parent's polygenic scores to estimate genetic confounding effects, which we have shown can be problematic (Nature Genetics, 2022), as have our own LCDS researchers in other publications (Science, 2018).

(Supplementary Material | P.14)

FAQ.

I appreciated that the FAQ explicitly stated that it was intended for broader audience who are “new to the scientific terminology and methods.” I therefore read the FAQ with this in mind and have several suggestions as I felt it often was not written in language that would be accessible to a broader audience.

Answer: Thank you for your constructive feedback to improve our FAQ. With your comments in mind, we have now implemented various changes to ensure the clarity and transparency of our work, including adding a Glossary. We hope these revisions have significantly improved the FAQ for a broader audience.

First, the FAQ document could do with some re-organization. For instance, it introduces concepts like GWAS and DNA variants 4 pages in when the terms have already been used. It would make more sense to define key concepts first.

Answer: We are grateful for this suggestion, as it was indeed inconvenient for readers to find the definitions of terms at the end of the FAQ. We have referred to the Glossary at the beginning in the designated box.

Second, there are moments when the FAQ reads in more accessible language but then within the same sentence or set of sentence switches to more technical language. It is unclear to me how the authors decide which terms to define more accessibly and which to leave as-is. This is another reason why I think some definitions of key terms at the very start would be helpful (e.g., GWAS, polygenic score, SNP, genetic association, occupational status).

Answer: This is always a tension when trying to convey complicated and interdisciplinary scientific findings to a broader audience. This remains a challenge but we have now extended our Glossary and key terms and note the existence of the Glossary at the beginning. The online version of this FAQ will also have hyperlinks with the definitions, making it more rapidly accessible and readable.

Just as an example, on page 8 the authors discuss “individual genetic scores” called “polygenic scores.” I see this as a way to help make the term ‘polygenic score’ more digestible for someone who may not be familiar with the scientific terminology. But then the authors go on to say “which we used to control for genetic associations” and I am left wondering why they do not explain what a ‘genetic association’ is. I also wonder whether referring to a ‘polygenic score’ as an ‘individual genetic score’ might lead people to think this information is predictive at the level of the individual rather than as a population-level tool.

Answer: These are important considerations—thank you for raising them. We agree that we should have defined the term “genetic association” since it is quite technical. We have now included this definition in the glossary section.

Regarding the use of “individual genetic scores” as an alternative to “polygenic scores,” we acknowledge that this might introduce further challenges rather than clarifications. As the reviewer rightly pointed out (and we have also pointed out clearly in our FAQs), these are not individual-level analytic tools. Moreover, they are

genomic or polygenic in nature rather than genetic scores, which might lead readers to mistakenly think they are functional genes we have identified. This was not our goal.

To eliminate such confusion, we are now consistently using "polygenic score" throughout our FAQ and have defined it clearly within the glossary section. We believe that with both "genetic associations" and "polygenic score" clearly defined at the start of the document, there is no need to use alternative terms that could be challenging. We have amended the following paragraphs in our FAQ accordingly:

Genetic associations. *Refers to the relationship between single nucleotide polymorphisms (SNPs) and a particular outcome of interest. GWAS methodology tests for these associations across the genome. Significant associations identified in GWAS can then be used to create polygenic scores, where the effect sizes of selected SNPs are combined to predict the outcome.*

Polygenic score (PGS). *A single quantitative variable that summarizes genetic association to a phenotype by combining multiple genetic variants and their associated weights, derived from a GWAS. Polygenic scores for social outcomes are not tools to derive individual-level predictions but rather a population-level analytic tools.*

(Supplementary Material | P.8)

Third, I would urge the authors to take a look at the “Responsible Research Lifecycle” in a 2023 Gordon et al paper on the genetics of musicality:

<https://onlinelibrary.wiley.com/doi/abs/10.1111/nyas.14972> Part of reflecting on the ethical considerations of this study is not just to list those considerations but to spend some time thinking about how to respond to them. Box 1 provides a brief description of a fraught history, it also states what the study is not (i.e., not doing group comparisons), and then it tells people to go look at the FAQ. I understand that there are constraints on what can go in Box 1. In the spirit of the “responsible research lifecycle,” the FAQ could be a place to explicitly challenge determinist interpretations and address potential for misuse while providing guidelines for appropriate use (e.g., an FAQ question like: “did you find the genetic basis for occupational status” and “are people biologically predetermined to an occupational status” or to take a question from the Mignogna FAQ: “Should public officials, policy makers, insurers, or health care professionals use the results of this study to make decisions?”)

Answer: We now more explicitly challenge these determinist interpretations and potential for misuse in Box 1 and the FAQs. We have also added more

explicit questions and answers as above to clarify our position. We now have the relevant sections:

Are people biologically predetermined to have an occupational status?

Could genetic results alone be used at the individual level to predict someone's occupational status?

Should public officials, policy makers, insurers, or health care professionals use the polygenic score from this study to make decisions?

The FAQ says: "For socioeconomic status measures such as educational attainment, nearly 4,000 genetic variants have been associated with the outcomes in previous studies. Genetic variants associated with income have also been established. Previous research has also shown that the third measure of occupational status, is not only driven by socio-environmental factors, but it also potentially has a biological and genetic basis. Twin studies suggest a heritability of occupational status as between 0.30 to 0.40. The current study goes substantially beyond what we know about the genetics and biological factors associated with occupational status." I was struck by how little was discussed regarding what the socio-environmental factors known to impact occupational status. Earlier the FAQ says: "The majority of social stratification research has focused on the social determinants and social aspects of intergenerational transmission (i.e., from parent to offspring) of SES, often neglecting any role of biology or genetics," but again does not share what that literature has revealed. Given the fact that the study is presented as examining a complex interplay between genes and environments, I worried that not enough attention is being given to the environments side when describing the prior literature.

Answer: We deeply appreciate this point and have now added more information about previous non-genetic social stratification research. Given that some of us have worked extensively in this area examining social, economic and structural factors of social stratification, we inadvertently assumed broader knowledge and focused on the more novel genetic aspects.

I am struck by what I think is a very important point in the abstract of the manuscript that did not seem to be in the FAQ: 62% of the intergenerational transmission of occupational status can be ascribed to non-genetic inheritance such as family environment and possibly the effects of rare genetic variants." There are two very important points that are highlighted in the FAQ that are related:

"The intergenerational transmission of occupational status only partly explain children's occupational status. Occupational status is only partly associated with genetic variants, and partly with the family background; the rest is neither family nor genes but unique circumstances and factors that are still not measured. "

"Social environmental factors, such as the family that someone grows up in, has a strong impact

via what is called gene-environment correlations... gene-environment correlations likely account for approximately 25% of the discovered estimates.”

However, I could not find mention of this 62% anywhere in the FAQ and wondered why. It looks like it may be broken down into different components.

Answer: Thank you for pointing this out. We agree that it is beneficial to align the abstract with our FAQ presentation, especially the 'Key Points' section. We have now ensured that the same statistics reported in the abstract are embedded within the Key Points of our FAQ:

- *The **intergenerational correlation of occupational status between parents and their children is only partly explained by genetic factors, with 62% of the intergenerational correlation due to non-genetic factors** such as family environment and potentially rare genetic variants. The rest is neither family nor genes but unique circumstances and factors that are still not measured. Notably, this is not specific to occupations, but to most complex diseases, behavioral, and social outcomes.*

(Supplementary Material | P.5)

Response to the editor's comments

Thank you once again for your manuscript, entitled "Polygenic predictions of occupational status GWAS elucidate genetic and environmental interplay for status transmission, careers, and health", and for your patience during the peer review process.

Your Article has now been evaluated by 4 referees. You will see from their comments copied below that, although they find your work of [considerable] potential interest, they have raised quite substantial concerns. In light of these comments, we cannot accept the manuscript for publication, but would be interested in considering a revised version if you are willing and able to fully address reviewer and editorial concerns.

We hope you will find the referees' comments useful as you decide how to proceed. If you wish to submit a substantially revised manuscript, please bear in mind that we will be reluctant to approach the referees again in the absence of major revisions. We are committed to providing a fair and constructive peer-review process. Do not hesitate to contact us if there are specific requests from the reviewers that you believe are technically impossible or unlikely to yield a meaningful outcome.

As you will see, the reviewers suggest a number of additional analyses to increase the robustness of your results. We ask that you perform all of them and include results in your revision. Could you also provide information on how you accessed the NCDS data and the associated approval.

Answer: We are grateful for the thoughtful and constructive reviews of our manuscript and the Editor's openness to allow us to engage in major revisions and resubmit. The suggestions were extensive, but fair and resulted in a very major revision of the article. We conducted multiple new analyses that were suggested by the reviewers, with many included in a Supplementary Information. We are confident that given our extensive and serious revisions, we were able to address all of the points that were raised and hope you agree that the manuscript has now significantly improved. We also specify that our work was conducted under NCDS application number GDAC_2021_16_TROPF (Manuscript | P.17 and P.30) and apologize for this oversight in the original manuscript.

We have also implemented the changes to comply with editorial policies. Please note, we have slightly modified the title, so it does not exceed the length allowed.

Point-to-point response to the reviewers' comments

Reviewer #1

Remarks to the Author:

This paper investigates genetic associations with occupational status and gene-environment interplay for intergenerational status transmission. The research questions are interesting and the findings are important for multiple disciplines, including human genetics and social science research in general. I have several comments that hopefully useful to improve the current manuscript.

Answer: We thank the reviewer for the positive comments and appreciate the useful and constructive suggestions. To summarize our central changes, we conducted further investigations and substantially revised our analyses on polygenic predictions over the life course. We also provided detailed descriptions of our measures and implemented the changes described in greater details below. We hope that you agree this has now further strengthened our investigation and its presentation.

1. First, it is uncertain how some of the key variables were measured in the study (e.g., cognitive performance, scholastic motivation, externalizing behavior etc.). It'll be great if a table can be added to the supplemental materials to describe the definitions of the key variables used in the analysis.

Answer: Thank you for this suggestion. We have now incorporated detailed descriptions of the key variables, such as cognitive performance, scholastic motivation, and externalizing behavior, among others. We believe this addition will offer readers a clearer understanding of the metrics and methodologies employed in our research. These descriptions can now be found in “**15.1 Description of mediators**”, “**17.1 General health**,” and “**17.2 Mental health**” sections of the supplemental material (**Supplementary Information | P.32 | P.43 | P.46**).

2. Second, the results on three occupational status variables (S Figure 7) consistently show non-linear patterns of the genetic association with occupational status (i.e., decreases after age 33 but increases after age 42). I'd suggest the authors give more discussions on these findings. What are possible explanations for this pattern and to what extent it's attributable to measurement issues (e.g., measures were not available in a 10-year window between 33 and 42)?

Answer: We thank the reviewer for this excellent point. It is indeed true that there is a non-linear trend and we conducted a thorough investigation in order

to find a theory-driven explanation of this pattern. We provide a more detailed response here, but also have adjusted the manuscript and added an expanded section '12. Polygenic score prediction over the life course' in our supplementary material (**Supplementary Information | P. 30**).

Initially, our additional analyses did not reveal any explanations behind these trends. There were no differences between the sexes that would indicate that the decrease might be driven disproportionately by females, for example as a result of family formation. Similarly, there is no evidence for a difference in r^2 if we stratify the analysis by number of children. Furthermore, the distribution remains fairly consistent for all sweeps from age 33 to 55. The case counts gradually decrease, as one would expect due to attrition, but nothing that would sensibly explain a marked decrease to age 42 and then again, an increase later.

Subsequently, we conducted additional systematic checks on our data cleaning procedures. At this stage, we spotted certain anomalies in the NCDS occupation data, particularly, during the age 42 sweep. As one of our tests, we focused on the subset of individuals for which occupational change from age 33 to 42 was observed in the data, but who apparently had switched back to their previous occupation at age 46 and stayed there at age 50. A substantial set of such cases (519) emerged. In comparison, only 36 people had similar jobs at 33, 42 and 50, but switched at age 46. Even a cursory glance raised substantial doubt about the quality of age 42 data, as a significant number of these occupational changes were atypical, to say the least, e.g. from broker to nurse back to broker, from postal worker to farmer and back, or from farmer to travel agent and back, etc. which theoretically might be possible but rather exceptional.

However, there are two different datasets within the NCDS data release, both containing occupational information. We had used the official NCDS *occupation_coding* data, as they contained ready-made and over time standardized occupational information, including CAMSIS-scores we relied on. But in addition, the raw occupational information is also available as part of the activity calendar in the standard data release for each wave. This data significantly differed for the anomalous cases and using it to manually construct all occupational status metrics, we obtained a much-improved performance on sweep 6 (and to a lesser extent sweep 7 and 8). The nonlinearities were therefore an artifact based on erroneous data in the official NCDS release. We have informed the NCDS about this issue and hope that an official updated version can be released at some point in the near future as this dataset is used extensively to study socioeconomic stratification.

Given this, we changed our overall prediction for the NCDS in the main text, where we combine all five measurements. The same applies to all other secondary analyses based on the NCDS occupation data. The necessary

modifications have been implemented in the text, and none of the results, however, qualitatively change. There is one exception, which we discuss in-depth in our reply to the next point (3) since it directly relates to the criticism raised there.

Below, we present an updated Supplementary Figure 7, where we plotted incremental R-squares of polygenic score predictions of occupational status measures over the life course in the NCDS (**Supplementary Information | P.31**) and updated Figure 4 in the main text for out-of-sample prediction performance (**Manuscript | P.32**) and updated the text within our paper (**Manuscript | P.8**):

“The longitudinal data in the NCDS reveal changes in the PGS effects across the life course or career trajectories, respectively. First, we were able to examine PGS prediction of occupational status across the life course, at ages 33, 42, 46, 50, and 55(see SI 13).”

Supplementary Figure 7. Incremental R^2 of polygenic score predictions of occupational status over the life course, NCDS

Figure 4. Out of sample polygenic prediction performance within UK Biobank and NCDS. Incremental R^2 compared to a baseline model consisting of 10 principal components, sex, and age. Bars denote 95% confidence intervals. $N = 24,579$ for CAMSIS and $24,472$ for ISEI and SIOPS in the UK Biobank, for NCDS average performance over different ages ($N = 5,389; 5,312; 5,211; 4,902; 4,263$ for CAMSIS at age 33, 42, 46, 50 and 55, $N = 5,449; 5,293; 5,197; 4,892; 4,252$ for ISEI/SIOPS)

- Third, related to the second point, another interesting finding is from the trajectory analysis. The results show that occupational status of individuals at the lower end of the PGS distribution experienced an early increase but then stabilized (or even decreased), whereas those at the higher end had an opposite pattern. I'd like to see more investigations on this result. For example, who are those at the lower and higher ends of the PGS distribution and why they show divergent trajectories in their occupational status? That'll help to understand the results better.

Answer: Due to the data issues outlined earlier, we revisited our growth curve models. Subsequent analysis revealed that the observed nonlinear contraction was likely an artifact of the problematic data and could not be replicated in

subsequent runs. However, while constructing occupational status from the raw NCDS data, we identified an opportunity to utilize the activity calendar data within the NCDS. This enabled us to extract occupational status information for every occupation the participants undertook up to the age of 55.

Given the limited interpretability of growth curve models, we adjusted our approach. Capitalizing on this expansive dataset, we now present comprehensive descriptive trajectories spanning 30 years of career development. These trajectories illustrate the mean occupational status year-by-year, stratified by PGS, parental SES, and sex. Rather than focusing on fixed time points, our revised analysis emphasizes the duration since an individual's entry into the labor market.

This approach unveils pronounced disparities between sexes and parental educational backgrounds. It also underscores the richness of career trajectories, highlighting the considerable depth and nuance that would be overlooked if we were to simplify the analysis into a quadratic model based on just five time points.

We included a new Figure 6 which is below and reflected on these findings in the main text as well (**Manuscript | P.8**):

“By leveraging the NCDS activity calendar data, we delineated comprehensive career trajectories over thirty years, from the onset of participants’ professional lives. When stratified by PGS quintiles, parental SES, and sex, these trajectories revealed a significant interplay between polygenic scores and social factors (Figure 6). Individuals who started their careers in the lower end of occupational status scores but ranked high in the PGS consistently advanced in their careers over the years. Conversely, those who initially held higher occupational status but had lower PGSs exhibited a steady decline in their professional trajectories, as measured by occupational status scores (Supplementary Figure 12). Drawing parallels with gravitational theory,⁵¹ this might suggest that PGSs capture a set of factors that, in turn, act as a “gravitational pull”, guiding and navigating individuals in specific occupational environments and trajectories. While our focus is on CAMSIS, similar patterns were evident for SIOPS and ISEI, underscoring the consistency of our findings (Supplementary Figures 13-14). These results further highlight the importance of understanding how and why societal structures and factors correlate with genotypes and jointly predict career trajectories. ”

Figure 6. Mean Percentile of the occupational status (CAMSIS) distribution across the career stratified by sex, parental education and the CAMSIS PGS. $N = 201,939$ time points from 5,475 individuals. Parental education measured as Low = No Qualifications, Medium = Lower Secondary, High = Upper Secondary/Degree.

Furthermore, this updated data provides deeper insights into the influence of an individual's first job on PGS-based career trajectories. Specifically, individuals who begin their careers in the bottom 20% of occupational status but rank in the top 20% of the PGS distribution experience a swift rise in their mean occupational status over the years. Conversely, those who start in the top 20% of occupational status but are in the bottom 20% of the PGS distribution witness a consistent decline in their mean occupational status throughout their careers.

We included these analyses into our supplementary material section 16 (**Supplementary Information | P.37**) with novel figures as provided below. While shown for CAMSIS, these results hold also for SIOPS and ISEI as we document in the supplement.

Supplementary Figure 12. Mean Percentile of the occupational status (CAMSIS) distribution across the career stratified by career start and the CAMSIS PGS.

- Fourth, in terms of the social mechanisms linking genetics and occupational status, in addition to the factors investigated in the analysis, health status and delinquency at early life are two other possible mediators. The authors have shown that the association between PGS and general health is reduced when parental occupational status is included. Another possibility is that health status at early life mediates the association between genetics and occupational status. Moreover, there is evidence that delinquency and early-life criminal justice involvement mediate the genetic association with educational attainments. If appropriate measures are available in the data, the authors may explore those additional social mechanisms.

Answer: Thank you for these excellent points. Early-life criminal justice involvement and delinquency are behaviors that are captured by our externalizing factor, so it is already incorporated into the mediation model.

You can find detailed information on this in our new section “15.1 Description of mediators” (**Supplementary Information | P.32**) which we added as our response to the point (1) where we addressed the valid criticism raised and now provide a transparent information of our mediation models and measures.

Regarding health, we have included the subjective health assessment from late adolescence/early adulthood, which was obtained during sweep 4 (age 23), as it represents the earliest available general health indicator. The results related to this addition are discussed in the main text as following (**Manuscript | P.11**):

“In the NCDS data, we tested the mediating effects for the occupational status PGS of adolescent phenotypic measures of cognitive ability, externalizing behavior, internalizing behavior, scholastic motivation, occupational aspiration and subjective health (see Methods and SI 15). Depending on the career stage of the respondents indicated by NCDS waves, these variables explained 56-74% of the link between our PGSs and occupational status. As expected, cognitive ability was the main mediator, explaining 33-51% of the association depending on respondents’ age. Scholastic motivation explained between 8-11%, occupational aspiration 7-11%, and other non-cognitive traits up to 5%. The overall mediation by subjective health was minimal. Effect reductions are proportional when adjusting for parental SES in order to control for passive gene-environment correlation or indirect effects respectively (SI 11.3).”

5. Fifth, another key finding is that 38% of the intergenerational transmission of occupational status can be attributed to genetic inheritance and 62% is to non-genetic inheritance. Based on the description, these estimates seem to be based on the reduction of the parent-child correlation in occupational status after adjusting for the child’s PGS rescaled based on the SNP heritability. Yet only including the child’s PGS does not full take into account all possibilities of genetic confounding. Parental genetic factors can be linked to children’s occupational status via non-genetic mechanisms (i.e., genetic nurturing effects) and these effects are not captured by child’s PGS. Therefore, to estimate genetic confounding effects, both parents’ PGSs are better options as control variables. I’d encourage the authors to perform this analysis or at least to discuss the limitations if data (i.e., parents’ genotypes) are not available.

Answer: This an important point and we thank the reviewer for raising it. In response, we have provided a more detailed explanation here and have also adapted the text to include an expanded section on 'limitations' in the final discussion.

We agree with the reviewer regarding the importance including parental genotypes, as the child's PGS alone may not fully account for genetic confounding. However, we are currently constrained by data availability in

this regard and hope that future research can address this limitation with access to more comprehensive family genetic data. We have duly incorporated these considerations into our discussion section (**Manuscript | P.16**):

“The reduction of PGS prediction within families, also emphasizes the relevance of recent initiatives for discovery designs using family data and to further study the role of assortative mating for within-family effect reduction.⁵² It is particularly important since parental genetic factors could influence their children’s occupational status through non-genetic mechanisms and these effects might not be adequately captured when considering only the child’s PGS. Therefore, we recognize the importance of including both parents’ PGSs as control variables to estimate genetic confounding effects, but this was not possible using the current data. This underscores the need for additional research with multi-generational genetic and social survey data. Despite these limitations, the current study offers many new insights into the interplay between genetics and occupational status scores along with socioeconomic status.”

Reviewer #2

Remarks to the Author:

The authors report the results of a GWA analysis of occupational status (OS) measures using UKB data. The GWA analysis identified 106 associations of which 8 are novel. The OS measures yield a SNP heritability of 10-15% and polygenic scores derived from their GWA summary statistics predicted 5-8% of the OS variance in an independent sample.

Although several GWA analyses of SES variables, especially educational attainment, have been reported, this GWA report of occupational status per se follows up a previous GWA report of occupational status in UKB (Ko et al., 2022). The present paper analyzes three indices of occupational status and shows that they are highly correlated genetically and could be aggregated in a single index of occupational status (OS).

This is a thorough analysis of the UKB data, with replication of key findings in other data sets, as well as a within-family analysis using sib pairs in UKB.

Answer: We thank the reviewer for the positive comments and have made extensive revisions to our paper based on the points raised. We have now conducted a substantial revision, compared our estimates with Ko et al.'s study, and performed additional analyses involving UK Biobank spouses, as suggested.

6. I have only one issue, which is the relationship of the present analysis to the previous UKB GWA analysis of OS by Ko et al. The authors say Ko et al. used ‘skill-based occupational groups’ and say that ‘skill-based measures have been criticized’ and contrasted Ko et al.’s measure with their use of ‘353 categories’. However, this seems a bit unfair to Ko et al., who analysed a measure based on the UK Standard Occupational Classification (OSC) scheme which was devised by the UK Office for National Statistics and begins with the 353-unit groups mentioned by the current authors. Importantly, the OCS makes it possible to treat OS as a continuous variable, which provides more robust analyses than the analysis of categories. The OSC has been used in thousands of studies, so it is an important measure to analyse – and to analyse in relation to the three OS measures used in the present paper. One of the present study’s findings is that their three OS indices yielded very high genetic intercorrelations. So, I bet that the present paper’s composite measure of OS correlates very highly with the Ko et al. OSC measure. Could the authors calculate the phenotypic and genetic correlations between their OS index and the OSC index used by Ko et al.? Regardless of the overlap with Ko et al., as the authors point out, their OS index doubled the heritability using CAMSIS and increased SNP discovery by more than threefold. But their paper should be seen as an extension of the Ko et al. analysis.

Answer: We thank the reviewer for pointing this out, as we do not wish to sound unfair towards the initial study by Ko et al. We now immediately highlight in the introduction that our study is an extension of theirs and state that sociological measures are preferable, as first digits of SOC2000 coding analyzed in Ko et al. have limited analytic capacity to capture SES and are open to criticism based on this limitation (**Manuscript | P.3**):

“We extend previous work of a GWAS on broadly skill-based occupational groups using the UK Biobank, which identified 30 independent SNPs associated with 9 categories of the UK Standard Occupational Classification (SOC) and a SNP-heritability of 0.085.²⁷ Since occupation in the UK Biobank is richly measured using 353 categories, we go markedly beyond the existing GWAS by drawing from expert sociological theory and measurement of occupational stratification. Sociological measures are preferable since purely skill-based measures suffer from inconsistent operationalization and lack theoretical and substantive thinking about the underlying mechanisms of status attainment, ignoring, for example, social prestige and other status factors.¹”

We acknowledge Ko et al.’s study and the importance of the UK Standard Occupational Classification (SOC) scheme. Our intent was not to undermine their work but to highlight the differences in our approaches in understanding occupational status. We would like to clarify that while Ko et al. utilized the SOC scheme, they specifically looked at the first digit of SOC2000. Our reference to their usage of ‘skill-based occupational groups’ was in relation to this particular aspect of their analysis. Our intention was to be descriptive

rather than critical, and in hindsight, we realize that the phrasing may have come across as the latter. We understand the point raised regarding the continuous nature of the SOC. However, it's worth noting that interpreting the first digit of SOC2000 as a continuous measure is a unique decision. In our experience within the realm of occupational classification analyses in the social sciences, such an interpretation is uncommon. Ko et al.'s analytical choice to treat it as ordinal is indeed commendable and aligns more with traditional social science perspectives on the data.

Following the suggestion, we did investigate the correlations between our occupational status measures and the SOC index used by Ko et al. As anticipated, there is a substantial genetic correlation. The phenotypic correlation, while present, is of a lesser magnitude.

SIOPS	0.906	0.804	0.82
0.995	ISEI	0.824	0.799
0.987	0.99	CAMSIS	0.753
0.974	0.972	0.949	SOC2000 (first digit)

Response Figure 1. Phenotypical Correlation (Upper Triangular) vs. Genetic Correlation (Lower Triangular) of Occupational Status Measures and SOC2000 (first digit)

- The paper is well written and reports other interesting analyses that replicate and extend other publications. For example, the authors report that intergenerational

transmission of OS is more than half due to shared family environment.

Answer: Thank you for such an encouraging comment with regards to our findings.

8. Another finding that replicates findings in the literature is that the between-family association between polygenic scores and OS is halved when the association is analyzed within families. The authors suggest that assortative mating is responsible for much of this reduction, which indicates that so-called indirect effects need to take assortative mating into account. Could the authors test this hypothesis using spouses in UKB? What is the assortative mating coefficient for the authors' three indices of occupational status?

Answer: Thank you for this excellent suggestion. We tested this hypothesis accordingly and found further support that at least 25% of the effect reduction is attributed to assortative mating. We extended our supplementary material section 12.4 *The sibling design and assortative mating* where we provide the details of our additional analyses (**Supplementary Information | P.28**). We also reflected on it in our main text accordingly (**Manuscript | P.10**):

“By employing a method first proposed by Lee et al.,⁶ we demonstrate that, even in the absence of indirect effects, within-family effects are plausibly anticipated to be attenuated by 22-27% (SI 12). We find further support for attenuation by directly analyzing the spousal PGS correlation, which substantially exceed what could be expected from simple phenotypical assortment and provide a full description (see SI 12). Accordingly, it closes the observed gap between both estimates.”

We also provide spousal correlations below for convenience:

Outcome	cor	ci_lower	ci_upper	N
camsis	0.3305987	0.3187508	0.3423432	21,903
isei	0.2355167	0.2229088	0.2480458	21,696
siops	0.2239975	0.2113209	0.2365988	21,696

Supplementary Table 5. Spousal correlations of occupational status scores.

Outcome	cor	ci_lower	ci_upper	N
camsis_sbayesr	0.11544624	0.09509622	0.1356998	9,074
isei_sbayesr	0.08937277	0.06892360	0.1097469	9,074
siops_sbayesr	0.08797762	0.06752393	0.1083574	9,074

Supplementary Table 6. Spousal polygenic scores correlations of occupational status scores.

9. A novel and intriguing finding is the compression of genetic prediction in mid-career, although I wasn't sure what to make of this. Could the authors unpack this some more? I note that they did not include this finding in their FAQ ('What are the big take-away findings'), where they might have explained it in simpler terms.

Answer: This is an excellent point and we have now included a more straightforward explanation in the FAQs and actually rewrote them in general to ensure that they are clear. In the FAQs under 'What are the big take-away findings?', we now have (**Supplementary Information | P.9**):

- *We examined the genetic scores across 30-year career trajectories measured at age 33, 42, 46, 50 and 55 to reveal that societal structures correlate with genotypes and jointly predict career trajectories. Individuals who started in lower occupational status percentiles but ranked high in the PGS quintile for occupational status, consistency advanced their careers over those 30 years. Those who held higher occupational status jobs but had lower PGSs, exhibited a steady decline in their professional trajectories.*

10. The Discussion is well written and raises intriguing issues. The Discussion, Box 1 and FAQ in the Supplement will attenuate some of the blowback that might come from showing the importance of genetics in the traditional sociological domain of occupational status.

Answer: We thank the reviewer for such a positive comment and appreciation of our attention to how we communicate our approach and findings transparently in the Discussion and through the inclusion of Box 1 and FAQ.

11. One very minor quibble: The Discussion includes this sentence: 'As often noted, however, twin studies likely also overestimate heritability due to the violation of the assumption that identical and fraternal twins share environmental influences to the same extent.' However, this is unreferenced and, to my knowledge, many tests of the equal environment assumption using different designs show that the assumption holds up quite well.

Answer: We thank Reviewers 2 and 4 for raising a similar point here; we agree that empirical evidence is scarce and have now modified this paragraph to eliminate any possible confusion (**Manuscript | PP.14-15**):

“We note that the applied extrapolation assumes SNP-heritability levels but could still represent an underestimation since PGSs have a lower prediction compared to SNP-heritability. But, the latter is still smaller than the heritability estimated from twin models; hence, SNP-heritability as measured here remains conservative compared to previous studies.⁶¹ The discrepancy between SNP- and twin-heritability might be due to rare genetic variants, higher environmental homogeneity within families, and non-linear genetic effects.^{46,76}”

Reviewer #3

Remarks to the Author:

This is an exceptional well thought through and conducted study. It should be regarded as the gold standard as to what questions regarding SES can be addressed using genetic data as well as the best way to address them. My comments are minor but I hope contribute in some way to polishing a fine piece of scientific literature.

Answer: We want to express our sincere gratitude for such kind words! Your acknowledgment of our work is nothing but truly inspiring. We also appreciate your comments, which, though minor, undoubtedly contributed to refining our manuscript – we were able to perform additional tests to address your concerns and further clarified methods and statistical tools we used.

12. In the abstract and throughout the manuscript 106 genetic variants are described as being identified. Either loci or independent SNPs would be preferable here as there are far more than 106 associated variants. Furthermore, it is stated that eight of these variants has not been previously associated with differences in SES. Could the authors clarify if these eight loci harbour variants that have previously been associated with differences in SES?

Answer: We thank the reviewer for this thoughtful comment and have now made the necessary amendments to address these concerns. To provide clarity and precision, we have updated the terminology in the main text and the appendix, replacing “genetic variants” with “independent SNPs”.

With respect to the eight SNPs not previously associated with SES differences, we accordingly conducted a comprehensive assessment. Utilizing FUMA, we

identified all 1000 Genomes SNPs in linkage disequilibrium (with $R^2 > 0.6$ for European ancestry) related to the 106 independent SNPs. This yielded a total of 11,206 SNPs. Subsequent analysis, referencing the GWAS catalogue and the IEU OpenGWAS project, revealed 1,005,470 phenotypical associations that reached at least suggestive significance ($p < 5 \times 10^{-6}$). We then excluded any variants having a genome-wide significant association with traits related to education, income, or other socioeconomic outcomes. This filtering process culminated in the identification of 8 unique hits (rs12137794, rs17498867, rs10172968, rs7670291, rs26955, rs2279686, rs72744938, rs62058104).

Overall, none of these eight SNPs is in LD (at $R^2 > 0.6$) with a SNP that is present in 1KG and has been noted in the GWAS catalogue or IEU OpenGWAS to harbour associations with income, years of schooling, qualifications or similar socioeconomic status related traits at $p < 5 \times 10^{-6}$. These findings are in the Table B8 of the Supplementary Tables file (**Supplementary Tables | B8. PheWAS of Top Hits**).

13. Additionally, it is stated that FUMA was used to identify these 106 SNPs. However, in the results it is stated that the R^2 threshold was 0.1. The documentation of FUMA is poor here but FUMA uses two rounds of clumping. The first is conducted in exactly the same manner as PLINK and is used to define independent significant SNPs as well as the border of the loci that they are in. A second round of clumping (by default at a lower R^2) is then applied on these independent significant SNPs to identify what FUMA calls lead SNPs.

My question is was the second R^2 equal to the first? If so the independent significant SNPs are the same as the lead SNPs. Also why does the R^2 change between defining independent SNPs/loci and examining if these same SNPs are associated with other traits (In the PHEwas the $R^2 = 0.6$)?

Answer: We thank the reviewer for raising this point and for providing us with the opportunity to clarify the methods we used. To eliminate any confusion regarding the FUMA method and its application, we have now updated Section 6.3 of the Supplementary Information (**Supplementary Information | P.18**):

“7.3 Findings

After inflating the standard errors by the square root of their respective intercepts from LD Score regressions, our GWASs identified 106 independent SNPs for CAMSIS including 56 found also for ISEI and 51 for SIOPS based on clumping all genome-wide significant SNPs using an R^2 -threshold of 0.1 and a window-size of 1000kb, one of which (only significant for CAMSIS) was found on the X-chromosome (see Figure 2 in the main text for the Manhattan plot of the autosome). In exploratory sex-specific GWAS, no separate hits emerged, with genetic correlations being very close to and not significantly different from one.”

As the reviewer can see now, FUMA was not employed to identify the independent SNPs. Instead, for this task, we utilized the --clump command in PLINK. We set an R^2 -threshold of 0.1 and a window-size of 1000kb, targeting all genome-wide significant hits ($p < 5 \times 10^{-8}$). We then turned to FUMA, but solely with the objective of identifying all SNPs present in the 1000 Genomes dataset that are in linkage disequilibrium (LD) with our previously identified 106 independent SNPs, as detailed in our response to the comment above.

14. The use of MTAG seems somewhat questionable here. I do see the increase in the number of genome-wide significant loci but I don't understand how this has occurred. Specifically, the authors have near (genetically) identical traits (the r_G between them is >0.8) and the GWAS are performed on the same samples (If the education data set used is from Lee et al. there would be new participants, but it seems the data used are all from UKBB). Could the authors provide the mean χ^2 and effective N_s pre and post MTAG? It would help confirm if MTAG is of benefit.

Answer: These are important points, and we would like to clarify our approach.

In the case of the NCDS, we utilized the EA3 GWAS dataset, excluding data from 23andMe and NCDS itself. When conducting validation within the UKB dataset, we exclusively employed the UKB GWAS dataset and ensured the exclusion of siblings or adoptees for specific analyses.

The observed improvement in our results can be attributed to variations in sample size and heritability. The sample sizes for income, and particularly for educational attainment (EA), are substantially larger. Regarding heritability, EA exhibited the highest values, followed by CAMSIS. This hierarchical pattern of heritability is also reflected in the relative increases in effective sample size, which are largest for SIOPS/ISEI and smallest for EA, as displayed in the following table below.

Response Table 1. MTAG: χ^2 and sample sizes

Trait	GWAS_mean_chi2	MTAG_mean_chi2	GWAS_N	MTAG_eff_N
CAMSIS	1.738	2.508	273157	697176
ISEI	1.568	2.418	271769	817666
SIOPS	1.534	2.435	271769	863640
Income	1.639	1.972	353673	721973
Years of Education (EA3)	2.074	2.584	763541	886012

15. Also, regarding the mean χ^2 . I do see that the authors have used fast-GWA-GLMM (again minor but I believe the title of the paper is incorrect) to account for relatedness and reported the LDSC intercepts which all seem close enough to 1. However, by including the mean χ^2 it would be possible to observe the degree to which non-polygenic effects have contributed to the signal.

Answer: We apologize for any confusion and would like to clarify our methodology. We indeed employed fastGWA for our analyses; however, since all our phenotypes are continuous, we did not use fastGWA-GLMM, which is an extension tailored for binary phenotypes.

Regarding the title, if you are referring to our original title - '*Genome-wide association study of occupational status and prestige identifies 106 genetic variants and defines their role for intergenerational status transmission and the life course*,' we have made revisions. The title is now '*Polygenic prediction of occupational status GWAS elucidate genetic and environmental interplay for intergenerational transmission, careers, and health*.' If your concern revolves around the term 'polygenic,' we are open to further adjustments for even greater clarity.

In response to the concern raised about the mean χ^2 , we agree with the assessment. Below, we provide the values for the mean χ^2 . The observed inflation, which ranges between 5% and 7.5%, suggests that potential confounders such as population stratification have a limited impact. This finding further supports our interpretation that the predominant portion of the observed signal is polygenic. We accordingly added a new section to our supplementary materials (**Supplementary Information | P.20**):

“9. Population stratification test

We applied LD Score intercept method⁸ to assess whether population stratification influenced our findings or potentially resulted in false positives. In doing so, we used the LDSC software⁹ to calculate LD Score regressions for each occupational score separately. LD Scores were computed utilizing genotypic data from individuals of European ancestry within the 1000 Genomes Project, specifically focusing on HapMap3 SNPs. Inclusion in the LD Score regression analysis was limited to HapMap3 SNPs with a minor allele frequency (MAF) exceeding 0.01 only.

Supplementary Table 2 below demonstrates the results of our analyses. It is noticeable that LD Score intercepts deviate statistically significantly from 1, although the deviation is not substantial. The χ^2 statistics fall within the range

of 1.53 to 1.74 for each occupational score. Consequently, we find evidence supporting the notion that some of the identified SNPs are linked to our phenotypes, and approximately 5.2% to 7.4% of the observed inflation in χ^2 can be attributed to potential biases arising from factors such as population stratification and other confounders. Nevertheless, this influence appears limited, and our overall findings indicate evidence that the predominant portion of the observed signal is attributable to polygenic factors.

Outcome	Mean χ^2	Intercept (SE)	Inflation
CAMSIS	1.738	1.1193 (0.0142)	0.0740622
ISEI	1.568	1.0993 (0.0134)	0.0616461
SIOPS	1.534	1.0845 (0.0123)	0.0524581

Supplementary Table 2. *Chi2 statistics for occupational status scores*

16. In the discussion it is stated “Certain genetic traits may predispose individuals to achieve higher levels in these areas through biological pleiotropy (Hill et al. (2019))”. I think you mean vertical pleiotropy here. Hill et al. (2019) write “more biologically proximal/distal phenotypes” but the schematic in Figure 1 of Hill et al. (2019) clearly describes vertical (mediated) pleiotropy.

Answer: We thank the reviewer for this excellent suggestion, and we corrected it accordingly. We, indeed, rather referred here to the instances of vertical pleiotropy. The sentence is changed to the following (**Manuscript | P.13**):

“Certain genetic traits may predispose individuals to achieve higher levels in particular areas through a mechanism known as vertical pleiotropy (i.e., mediated pleiotropy).²³”

Reviewer #4

Remarks to the Author:

This study will be of great interest to behavioural geneticists and social scientists in general, and I am sure it will become highly-cited. The discovery of 106 variants (8 of which are novel), significantly expands our knowledge of genetic influence on behaviour, and the development of (relatively) highly-predictive polygenic scores will find many applications in future studies. The outstanding feature of the study is its meticulous phenotyping, using reliable measures that have been traditionally confined to sociological work. Consequently, the paper is also a great example of

interdisciplinary work, combining behavioural genetics and sociology. The detailed phenotyping is also the critical improvement compared to the previous GWAS of social status (Ko et al., 2022), an evidence-based advance of scale and rigour, as defined by the standards of Nature Human Behaviour. A few comments:

Answer: We are extremely grateful to read these very positive comments and appreciation of our study. The acknowledgment of our meticulous phenotyping and the incorporation of reliable measures, which are traditionally confined to sociological work, is particularly gratifying. We also thank the reviewer for their feedback. We have now revised our manuscript in accordance with the comments provided, including the validation of LDSC heritability using the BOLT-GREML method and enhancing the text of our manuscript.

17. The second sentence in Box 1 implies that the linking of biology to intelligence, criminality and status, is invalid. Given the recent advances in sociogenomics (including the current study), this position appears tenuous and is being contradicted by the rest of the paper. I suggest re-writing the sentence in a more neutral manner.

Answer: This is an important suggestion. We have now rewritten Box 1 and the accompanying FAQs in a more neutral and less tenuous manner as a better reflection of the field. Given the historical sensitivities around these topics and fact that it can be quite divisive, we are keen to ensure that our approach is clear and that results are not misunderstood or misappropriated.

18. It is interesting that all three measures of occupational status showed similar heritabilities. Have the authors considered validating the LDSC heritability estimates using another method, such as GREML?

Answer: This is an excellent point. Indeed, while it might initially seem that the heritabilities are closely aligned, a deeper analysis reveals important nuances.

The observation that using CAMSIS instead of ISEI/SIOPS can lead to an increase in SNP-heritability by almost 50% is particularly notable. This sizable difference underscores the significance of the measure chosen to gauge occupational status, while the high genetic correlations across the measures suggest that we are indeed tapping into a similar genetic component. However, the differences in heritability across the measures indicate that they capture this genetic component with varying degrees of precision.

To further validate our findings, we turned to BOLT-REML for heritability calculations. The results from BOLT-REML echoed our initial findings, albeit

with a slight increase in heritability across all traits. We reported it accordingly in the supplemental material (**Supplementary Information | P.19**):

“In addition and as a robustness check, SNP-heritability was also estimated using BOLT-GREML on the GWAS samples. The results confirm and even exceed the LDSC estimates.

Outcome	h²	SE
CAMSIS	0.152535	0.002779
Years of Education	0.162157	0.002790
Household Income	0.094426	0.002843
ISEI	0.119530	0.002696
SIOPS	0.114053	0.002680

Supplementary Table 1. Results for BOLT-GREML SNP-heritability”

- 19.** The reported incremental R² of 7.7% is that of the MTAG predictor. Is it fair to report this as the main result of the PGS analysis, given that MTAG also makes use of summary statistic from previous GWAS (education and income)? At the very least, it should be made clear from early in the paper, that this estimate comes from MTAG using education and income.

Answer: We thank the reviewer for this thoughtful comment. We agree that the results should be better clarified and have now changed the paragraph to the following (**Manuscript | P.8**):

“MTAG-based out-of-sample predictions, which incorporate occupational status measures with household income and educational attainment, were nearly identical in both UK data sets, with an incremental R² of 0.097 (SE=0.0035) in NCDS across all observations (0.075, SE=0.00287 in the UK Biobank) for CAMSIS, 0.065 (SE 0.0032; 0.054, SE 0.00250) for ISEI and 0.067 (SE=0.0031; 0.053, SE = 0.00248) for SIOPS (Figure 4). As expected, polygenic scores based on PRSice2 and SBayesR weights have smaller but comparable incremental R² values in both UK data sets across all measures of occupational scores (Figure 4).”

20. The authors use a clever method by Lee et al. to indicate that the reduction in PGS within-family prediction can be entirely due to assortative mating. While this may well be true, work by Demange et al. (2022) shows that adoptee prediction should not be attenuated by population stratification or assortative mating (see supplementary note 6 of the former study). In the current study, the adoptee design attenuation is very close to that of the sibling design, suggesting a role for indirect parental effects (I would urge the authors to use this term instead of “genetic nurture”, which is no longer favoured), rather than AM or stratification. This would make sense, given that the phenotype in question (occupation) is probably greatly influenced by family background. The Demange study was on cognitive and non-cognitive skills, and it showed the opposite result, namely much less attenuation in the adoption design compared to the other designs (suggesting that indirect effects were mostly due to AM/stratification). Perhaps it would be useful to compare and contrast these two findings.

Answer: This is a very good suggestion, and we thank you for the reference to the work by Demange et al. (2022). We agree that the term "indirect parental effects" may be more suitable than "genetic nurture" and made this adjustment accordingly in our manuscript. However, we disagree with the assessment that our results are incompatible with the findings from Demange et al. – quite the opposite, which we have now clarified in the manuscript. In our case, the effect reduction in the sibling design is almost double of what we see in the adoption design, therefore clearly indicating that a substantial amount of attenuation is due to AM/stratification. We apologize if this finding was not communicated clearly enough. A misunderstanding might result from the other result we present in this section. While we did not have information on parental polygenic scores available to complete Demange et al.'s trifecta, in a third design we instead included parental occupational status in the estimation of the effect of polygenic scores on occupational status. *This* model shows an effect reduction very similar to the adoption design, which is sensible, as both induce a conceptually similar control for passive gene-environment correlation by keeping the effect of parental phenotypes constant.

Overall, your suggestion to compare our findings with those of the Demange et al (2022) study is indeed valuable. We incorporated this comparison into our results section accordingly and extended our discussion of adoption design (**Manuscript | P.9**):

“We re-ran our GWAS for occupational status, while excluding the set of 3,414 respondents of European ancestry in the UK Biobank that stated that they were adopted and for which occupational information was available. Results from both designs are remarkably similar with the parental SES control design reducing the incremental PGS R^2 to 0.044, 0.031, 0.028 for CAMSIS, SIOPS, and ISEI, with an effect attrition of 22% for all three measures, and the adoptee prediction resulting in an R^2 of 0.043, 0.031, 0.027 for CAMSIS, ISEI, SIOPS respectively with an effect reduction of 23%, 22%, and 27%. Notably, our results concur with Demange et al. (2022),⁵³ where the extent of attenuation

for cognitive and non-cognitive skills was considerably smaller in an adoption compared to a sibling design.”

21. In the GenomicSEM analysis, it is shown that much of the genetic effect on social status is mediated by other factors, including risk tolerance. It is stated in the supplement that risk tolerance, perhaps surprisingly, has a positive effect on social status. I believe this is worth reporting in the main paper.

Answer: We thank the reviewer for this comment and for the opportunity to reflect on this in the manuscript. It is however worthwhile to stress that the positive effect of risk tolerance is positive in a model which already includes ADHD (as a proxy for behavioral disinhibition), which is genetically correlated with risk tolerance. In this sense, the more negative aspects of risk tolerance, relating to impulsivity and externalization are therefore to a certain extent already controlled for. We have now extended the results section and added the following into our main text (**Manuscript | P.10**):

“Among these mediators, the associations are generally similar for all three measures of occupational status. Of these, the strongest effects are observed for cognitive performance. However, when we introduce ADHD and openness to experience into the models, these associations are slightly reduced. The importance of ADHD is increased by the introduction of risk tolerance. In contrast to ADHD and neuroticism, risk tolerance positively correlates with the SES factor, when controlling for the other potential mediators (see Supplementary Tables 3-5).”

22. The measure that is defined by interaction networks (CAMIS) is the one with the strongest genetic signal. According to the authors: “A potential reason for this observation may be genetic selection into interaction networks”. I would rather argue that these interaction networks eventually become genetic networks, since by definition they include spouses. R.A. Fisher operationally defined social class as a spousal network (see a recent PNAS commentary by James Lee which provides the full quote). The authors should discuss this process more.

Answer: We thank the reviewer for pointing us to R. A. Fishers definition of social class also describing the processes motivating his statement. We expand on this in our paragraph now as follows (**Manuscript | PP.12-13**):

“As Fisher stated, referring to past historical periods and particular contexts: “[P]revailing opinion, mutual interest, and the opportunities for social intercourse, have proved themselves sufficient, in all civilized societies, to lay on

the great majority of marriages the restriction that the parties shall be of approximately equal social class.”⁷⁰ Evidence for genetic assortative mating has been demonstrated for political views⁷¹ and educational attainment,^{72,73} supporting strong phenotypic evidence of assortative mating by SES, race/ethnicity and religion, also showing this has evolved with demographic change.⁷⁴ The heritability of CAMSIS might partly capture effects of assortative mating on the phenotype of the individual. However, high genetic correlations between CAMSIS, SIOPS, and ISEI may point to the benefits of a more granular and exact measure of the same latent phenotype in CAMSIS.”

- 23.** The sentence “Second, when examining the genetics related to the intergenerational transmission of SES, mostly heritability studies on educational attainment assumed that genetic influences are stable in absolute terms and environmentally driven inequality reduces with lower intergenerational correlations” is poorly written and hard to comprehend. Please rephrase.

Answer: We thank the reviewer for pointing this out and have now improved the sentence as follows (**Manuscript | P.14**):

“Second, there are further important considerations related to the intergenerational transmission of SES. It has been a common assumption in heritability studies of educational attainment that genetic influences are stable in absolute terms, while environmentally driven inequalities tend to reduce with lower intergenerational correlations.”⁸

- 24.** “As often noted, however, twin studies likely also overestimate heritability due to the violation of the assumption that identical and fraternal twins share environmental influences to the same extent.” I do not think there is enough evidence to render the violation of the equal environments assumption “likely”. Also, one of the authors has a preprint on how heritability may also be underestimated in twin studies (Wolfram & Morris). Please either remove or rewrite the sentence.

Answer: We thank Reviewers 2 and 4 for raising a similar point here; we agree that empirical evidence is scarce and have now modified this paragraph to eliminate any possible confusion (**Manuscript | PP.14-15**):

“We note that the applied extrapolation assumes SNP-heritability levels but could still represent an underestimation since PGSs have a lower prediction compared to SNP-heritability. But, the latter is still smaller than the heritability estimated from twin models; hence, SNP-heritability as measured here remains conservative compared to previous studies.”⁶¹ The discrepancy between SNP-

and twin-heritability might be due to rare genetic variants, higher environmental homogeneity within families, and non-linear genetic effects.^{46,83}”

25. Minor point, there is a reference to “reproductive behaviour” in page 8 of the supplement, which should probably be replaced by “occupational status”.

Answer: We thank the reviewer for spotting the mistake that we corrected accordingly.